# TabSDS: a Lightweight, Fully Non-Parametric, and Model Free Approach for Generating Synthetic Tabular Data

**Elias Chaibub Neto** [1]

## Abstract

The development of deep generative models for tabular data is currently a very active research area in machine learning. These models, however, tend to be computationally heavy and require careful tuning of multiple model parameters. In this paper, we propose TabSDS - a lightweight, non-parametric, and model free alternative to tabular deep generative models which leverages rank and data shuffling transformations for generating synthetic data which closely approximates the joint probability distribution of the real data. We evaluate TabSDS against multiple baselines implemented in the Synthcity Python library across several datasets. TabSDS showed very competitive performance against all baselines (including TabDDPM - a strong baseline model for tabular data generation). Importantly, the execution time of TabSDS is orders of magnitude faster than the deep generative baselines, and also considerably faster than other computationally efficient baselines such as adversarial random forests.

## 1. Introduction

Easy access to data is fundamental for machine learning research. Real-world data, however, often contains sensitive information, which precludes its unrestricted sharing across the research community. As a result, synthetic data has become an increasingly popular alternative for training ML models. The goal is to create training datasets that closely resemble real-world data, allowing models trained on synthetic data to perform effectively on real data.

However, generating high-quality synthetic data that accurately reflects the complexities of real-world data remains a significant challenge for tabular datasets, which often con-

tain heterogeneous relationships among distinct data types, and usually tend to be considerably smaller than the large scale datasets used to train computer vision and natural language processing tasks.

The generation of synthetic tabular data has been a research topic in the statistical disclosure control field for over three decades now (Drechsler and Haensch, 2023; Reiter, 2023). More recently, the machine learning community has also turned its attention to tabular data, and several sophisticated deep generative modeling frameworks (including generative adversarial networks, variational autoencoders, diffusion models, normalizing flows, and large language models) have also been adapted for the generation of tabular data (see Bond-Taylor et al. 2021; Lu et al., 2023; Fang et al., 2024 for a few recent reviews).

These deep learning based approaches, however, tend to be computationally intensive and require extensive tuning of multiple model parameters. This caveat has motivated the development of more lightweight alternatives such as adversarial random forests (Watson et al., 2023).

In this paper, we propose a lightweight alternative to deep generative models as well. Building over sequential joint probability preserving data shuffling (SJPPDS), a recently proposed perturbative method for performing statistical disclosure control of numeric data (Chaibub Neto, 2024), we develop a completely non-parametric and model free approach for synthetic tabular data generation.

At a high level, our approach leverages rank and data shuffling transformations for the generation of synthetic data. It represents a multi-step procedure involving the generation of (uncorrelated) synthetic marginal distributions which are then rank-matched to shuffled versions of the real data. The synthetic marginal distributions are generated using a novel non-parametric approach which (in the asymptotic case) draws samples from the same distribution as the real data (see Theorem 1). The rank-matching step essentially induces the association structure of the real data into the synthetic marginal data. The result is the generation of the synthetic data which closely approximates the joint probability distribution of the real data. Our main contributions are the following.

[1]Sage Bionetworks, Seattle, Washington, United States of America. Correspondence to: Elias Chaibub Neto <elias.chaibub.neto@sagebase.org>.

*Proceedings of the 42nd International Conference on Machine Learning*, Vancouver, Canada. PMLR 267, 2025. Copyright 2025 by the author(s).

**First**, we introduce the synthetic tabular sequential joint probability preserving data shuffling approach, which we denote (for short) as TabSDS (**Tab**ular **S**ynthetic **D**ata **S**huffling) - a principled fully non-parametric and model free method for the generation of mixed synthetic data containing numeric and categorical variables. TabSDS extends the SJPPDS data perturbation approach in two important ways. First, it accounts for categorical variables (while SJPPDS is only applicable to numeric data). Second, it extends SJPPDS from a data perturbation approach (where the data is shuffled but no new values are generated) into a fully synthetic data method. Quite importantly, TabSDS is a truly lightweight approach, being considerably faster than adversarial random forests (the current state-of-the-art for computational efficiency among the strongest baseline methods for tabular data) and is orders of magnitude faster than deep generative methods.

**Second**, we show that contrary to deep generative models which require extensive tuning of multiple parameters, TabSDS depends more critically on a single tuning parameter which controls the amount of data shuffling. In addition to being easier to tune, this allows for a very precise control of the trade-off between data utility/fidelity and data privacy.

**Third**, we benchmark TabSDS against alternative generators available in the Synthcity library (Qian et al., 2023) across multiple datasets, and show that it consistently shows very competitive performance against the alternative approaches (including TabDDPM - one of the current state-of-the-art for data quality).

## 2. Related Work

Synthetic tabular data generation typically employs either machine learning or statistical models to produce artificial datasets that replicate the structure and statistical characteristics of real-world data.

Traditional synthetic data approaches based on statistical models include information preserving statistical obfuscation (IPSO) methods (Burridge, 2003; Cano and Torra, 2009; Langsrud, 2019), bayesian-network methods (Young, 2009), and fully conditional specification (FCS) methods (Drechsler, 2011). Non-parametric FCS methods based on CART models (Reiter, 2005) are considered state-of-the-art synthesizers in the statistical disclosure control (SDC) field (Drechsler and Reiter, 2011; Nowok, 2016).

In addition to synthetic data, the SDC literature is rich with data perturbation methodology (which attempts to preserve privacy by perturbing the data, rather than synthesizing new datasets from scratch). Traditional perturbative methods include microaggregation (Domingo-Ferrer et al., 2002), noise addition (Brand, 2004), rank-swapping (Moore, 1996), and data shuffling (Muralidhar and Sarathy, 2006). More

recently, Chaibub Neto (2024) proposed SJPPDS, a more principled data shuffling approach for numeric data, which tends to outperform CART in terms of data fidelity at the expense of a small increase in privacy risk. The TabSDS approach, proposed in this paper, extends SJPPDS (which is reviewed in the Background section) from a data perturbation method for numeric data into a fully blown synthetic data approach for mixed tabular data.

A synthetic data method related to TabSDS has been recently proposed by Domingo-Ferrer et al. (2025). Similarly to our contribution, their work is also based on the permutation of data ranks. But, contrary to our approach, their method is considerably more difficult to apply in practice as it: (i) requires the identification of the marginal distributions of the data using predefined parametric distributions (e.g., Gaussian, Gamma, etc); and (ii) requires the use of a more complex ontology-based semantic ranking approach (Domingo-Ferrer et al., 2013) for handling categorical variables. Another they difference is that their method relies on different data shuffling algorithms, rather than the SJPPDS method adopted here.

The number of tabular data generators in the ML literature is fairly large (see see Bond-Taylor et al. 2021; Lu et al., 2023; Fang et al., 2024 for reviews). In this paper, we compare TabSDS against a diverse and representative selection of synthetic data generators available in Synthcity. The selected models included: bayesian networks (Young, 2009), a traditional statistical method, which uses graphical models to represent probabilistic relationships; conditional tabular generative adversarial networks (CTGAN) (Xu et al., 2019), a deep learning-based approach that leverages conditional GANs to model complex non-linear dependencies in tabular data; tabular variational autoencoders (TVAE) (Xu et al., 2019), a specialized form of variational autoencoder designed specifically for tabular data; adversarial random forests (ARF) (Watson et al., 2023), which employs a recursive process where trees gradually capture the data's structural properties through alternating phases of generation and discrimination; and tabular denoising diffusion probabilistic models (DDPM or TabDDPM) (Kotelnikov et al., 2023), which uses a diffusion-based model specifically developed for tabular data and represents a very strong baseline model in terms of data quality. (DDPM has been highlighted as one of the strongest baselines in recent benchmarking studies (Kindji et al., 2024; Du and Li, 2025).)

While several recently proposed generators have improved upon DDPM (Jolicoeur-Martineau et al., 2024; Zhang et al., 2024; Shi et al. 2025; Cresswell and Kim, 2024), none of these methods are available in Synthcity. Furthermore, the evaluations presented in these recent works consistently place DDPM among the top generators across numerous evaluation metrics (with DDPM often ranking in second

place, only behind the proposed method). Finally, it is important to point out that all these additional models still require extensive optimization and are computationally more expensive than TabSDS.

## 3. Background - the SJPPDS Approach

Recently, Chaibub Neto (2024) proposed the sequential joint probability preserving data shuffling (SJPPDS) approach, as a more principled data perturbation method for statistical disclosure control of numeric microdata. Traditional data perturbation approaches (e.g., microaggregation) tend to be "ad-hoc" in the sense that while they preserve some particular aspects of the data, they end up modifying others.

The basic idea behind the SJPPDS approach is to approximate the joint distribution of the real (numeric) data via sequential applications of restricted permutations to the data (where the restricted permutations are guided by the joint distribution of a discretized version of the data). Importantly, because the approach only shuffles the data, the marginal distributions of the perturbed data are identical to the marginal distributions of the real data. The main drawbacks of SJPPDS are that it is only applicable to numeric data, and is unable to generate new data values (it only shuffles the data).

To explain how restricted permutations generate shuffled data which still retains the statistical association structure of the real data, we closely follow the illustrative example provided in Section 3 of Chaibub Neto (2024). Consider a toy dataset composed of two correlated variables, $X_1$ and $X_2$, as shown in rows 1 and 2 of Table 1 and in Figure 1a.

*Table 1.* Restricted permutation toy example

| $X_1$ | 8.8 | 9.5 | 9.3 | 9.7 | 10.9 | 9.6 | 10.2 | 11.4 | 12.1 |
|---|---|---|---|---|---|---|---|---|---|
| $X_2$ | 9.6 | 10.1 | 10.5 | 10.8 | 11.1 | 11.4 | 11.6 | 11.9 | 12.2 |
| $C_2$ | a | a | a | b | b | b | c | c | c |
| $X_1^\star$ | 9.3 | 8.8 | 9.5 | 10.9 | 9.7 | 9.6 | 12.1 | 10.2 | 11.4 |

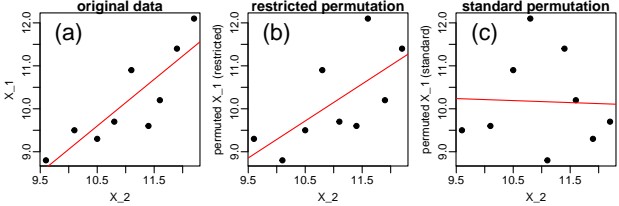

*Figure 1.* Restricted permutation example.

To generate a restricted permutation of the $X_1$ data relative to the $X_2$ variable we proceed as follows. First, we discretize the $X_2$ data into $n_c$ categories to obtain the categorical version of $X_2$, represented by $C_2$ in Table 1. (The discretization is done by binning the range of $X_2$ into $n_c = 3$ equally sized buckets, which are assigned categorical levels "a", "b", and "c".) A restricted permutation of $X_1$ relative to $X_2$ is then generated by separately permuting the values of

the $X_1$ data within each level of the $C_2$ variable, as shown in the $X_1^\star$ row of Table 1. Contrary to the standard permutations which completely destroys the association between $X_1$ and $X_2$ (Figure 1c), the restricted permutation procedure preserves the association to a good extent (Figure 1b). Note that the number of categories/levels, $n_c$, can be used to tune the amount of shuffling of the $X_1$ variable, with larger $n_c$ values leading to less overall shuffling and better association preservation between $X_1^\star$ and $X_2$.

The SJPPDS algorithm corresponds, essentially, to repeated applications of restricted permutations to the first $p - 1$ columns of a numerical dataset relative to its last column, over sequential rearrangements of the order of the columns. For instance, for a dataset containing $p = 3$ variables the algorithm performs a restricted permutation of $\{X_1, X_2\}$ relative to $X_3$, then of $\{X_2, X_3\}$ relative to $X_1$, and finally of $\{X_3, X_1\}$ relative to $X_2$. For completeness, the SJPPDS procedure is presented in Algorithm 6 in the Appendix. (See also Chaibub Neto 2024 for more details.)

## 4. The TabSDS Approach

TabSDS extends SJPPDS to categorical and synthetic data by leveraging rank transformations.

As described in Algorithm 1, the TabSDS approach calls 3 distinct algorithms depending on whether the data contains: (i) only numeric variables (lines 2 and 3 of Algorithm 1); only categorical variables (lines 5 and 6); or (iii) both numeric and categorical variables (lines 8 and 9).

---

**Algorithm 1** TabSDS($\mathbf{X}, i_n, i_c, n_c, p$)

1: **Input:** data matrix $\mathbf{X}$; indexes of the numeric variables $i_n$; indexes of the categorical variables $i_c$; number of levels $n_c$; sampling proportion $p$
2: **if** $\mathbf{X}$ contains only numeric data **then**
3:      $\mathbf{X}_s \leftarrow$ SyntheticSJPPDS($\mathbf{X}, n_c, p$)
4: **end if**
5: **if** $\mathbf{X}$ contains only categorical data **then**
6:      $\mathbf{X}_s \leftarrow$ CategoricalSJPPDS($\mathbf{X}, n_c$)
7: **end if**
8: **if** $\mathbf{X}$ contains numerical and categorical data **then**
9:      $\mathbf{X}_s \leftarrow$ MixedSyntheticSJPPDS($\mathbf{X}, i_n, i_c, n_c, p$)
10: **end if**
11: **Output:** synthetic data $\mathbf{X}_s$

---

In the next subsections we go over each of these algorithms.

### 4.1. The SyntheticSJPPDS Approach

The SyntheticSJPPDS approach (Algorithm 2) leverages rank and data shuffling transformations for the generation of synthetic data. At a high level, it corresponds to a two step procedure which first generates (uncorrelated) synthetic data

from the same marginal distributions as the original data, followed by a rank-matching operation which essentially induces the association structure of the real data into the synthetic marginal data. Next we describe the algorithm in more detail.

---

**Algorithm 2** SyntheticSJPPDS($\mathbf{X}, n_c, p$)

---

1: **Input:** numeric data matrix $\mathbf{X}$; number of levels $n_c$; sampling proportion $p$
2: $\mathbf{M}_s \leftarrow$ GenerateSyntheticMarginalsIOS($\mathbf{X}, p$) {create synthetic marginal distributions (Algorithm 8)}
3: $\mathbf{X}^\star \leftarrow$ SJPPDS($\mathbf{X}, n_c$) {perform sequential joint probability preserving data shuffling on $\mathbf{X}$ (Algorithm 6)}
4: $\mathbf{R}^\star \leftarrow$ DataRankMatrix($\mathbf{X}^\star$) {create matrix of ranks from the shuffled data (ties are broken at random)}
5: $\mathbf{X}_s \leftarrow$ MatchRanks($\mathbf{M}_s, \mathbf{R}^\star$) {rank-match the synthetic marginal data to the shuffled rank matrix (Algorithm 10)}
6: **Output:** synthetic data $\mathbf{X}_s$

---

The first step (line 2) is to generate the matrix $\mathbf{M}_s$ containing the (uncorrelated) synthetic versions of the marginal distributions of each of the variables in $\mathbf{X}$. The exact procedure is described in Algorithms 8 and 9 in the Appendix, but the core process is to separately sample from the marginal distribution of each variable using the interpolated order statistics procedure implemented in Algorithm 3.

---

**Algorithm 3** IOSSampler($\mathbf{x}_1, \mathbf{x}_2$)

---

1: **Input:** subsampled data vectors $\mathbf{x}_1$ and $\mathbf{x}_2$ of length $m$
2: $\mathbf{s}_1 \leftarrow$ Sort($\mathbf{x}_1$) {sort the values of $\mathbf{x}_1$}
3: $\mathbf{s}_2 \leftarrow$ Sort($\mathbf{x}_2$) {sort the values of $\mathbf{x}_2$}
4: $\tilde{\mathbf{x}} \leftarrow []$ {create empty vector}
5: **for** $i = 1$ **to** $m$ **do**
6: $\quad a \leftarrow$ Min($\mathbf{s}_1[i], \mathbf{s}_2[i]$) {find the min between $\mathbf{s}_1[i]$ and $\mathbf{s}_2[i]$}
7: $\quad b \leftarrow$ Max($\mathbf{s}_1[i], \mathbf{s}_2[i]$) {find the max between $\mathbf{s}_1[i]$ and $\mathbf{s}_2[i]$}
8: $\quad \tilde{\mathbf{x}}[i] \leftarrow$ Uniform($a, b$) {sample value in the [a, b] interval}
9: **end for**
10: **Output:** vector of interpolated samples $\tilde{\mathbf{x}}$

---

The basic idea is the following. For each variable, Algorithm 3: (i) takes as input 2 random subsamples ($\mathbf{x}_1$ and $\mathbf{x}_2$) of size $m$ from the original data (line 1); (ii) sort the 2 subsamples, so that the $i$th element of each subsample correponds to the $i$th order statistic of the subsample (lines 2 and 3); and (iii) for each index of the sorted subsamples, the algorithm generates synthetic data by randomly sampling a value within the range of these two values (lines 6 to 8).

Since Algorithm 3 generates only $m$ synthetic data-points, in order to generate $n$ samples, Algorithm 9 (in the Appendix) repeatedly calls Algorithm 3 until it generates $n$ (or more) synthetic data-points (and, if the total number is larger than $n$, it randomly sample $n$ of these points).

Note that by randomly interpolating the values of order statistics in two random subsamples of the real data, Algorithm 3 generates synthetic data which approximates the marginal distribution as the real data. Figures 6 and 7 in the Appendix provide illustrative examples. In the asymptotic case, where the subsample size $m$ grows to infinity, the following result (proved in Appendix C) holds.

**Theorem 1.** *Let* $\mathbf{x}_1 = (x_{1,1}, \ldots, x_{1,m})^t$ *and* $\mathbf{x}_2 = (x_{2,1}, \ldots, x_{2,m})^t$ *represent random subsamples of size $m$ from a random variable $X$ distributed as $X \sim F_X$. Then the distribution of the synthetic data, $\tilde{\mathbf{x}}$, generated by Algorithm 3 converges almost surely to $F_X$ as $m \to \infty$.*

While Theorem 1 only proves that Algorithm 3 draws synthetic samples from the same marginal distribution as the real data in the asymptotic case, in practice the procedure produces very good approximations to the distribution of $X$ in the finite sample setting as well. To see why, note that, by construction, for any value of $t$, we have that the empirical cumulative distribution function (ecdf) of the interpolated synthetic sample, $\hat{F}_m^{\tilde{X}}$, is bounded by $\min\{\hat{F}_{m,1}(t), \hat{F}_{m,2}(t)\} \leq \hat{F}_m^{\tilde{X}}(t) \leq \max\{\hat{F}_{m,1}(t), \hat{F}_{m,2}(t)\}$, where $\hat{F}_{m,1}$ and $\hat{F}_{m,2}$ represent the ecdfs of the $\mathbf{x}_1$ and $\mathbf{x}_2$ subsamples (as shown in Figure 8 in the Appendix). Hence, if both $\hat{F}_{m,1}$ and $\hat{F}_{m,2}$ provide good approximations to $F_X$, it also follows that $\hat{F}_m^{\tilde{X}}$ will also provide a good approximation.

While this random interpolation approach generates synthetic data whose marginal distributions closely approximate the original data, it does not preserve any of the statistical associations between the variables. To fix this issue, the second step (implemented in lines 3 to 5 of Algorithm 2) is to induce an association structure in the marginal synthetic data which closely approximates the association structure of the real data (without changing the marginal distributions of the synthetic data). The result is a synthetic dataset which approximates well the joint probability distribution of the real data. More precisely, in line 3, Algorithm 2 applies SJPPDS to the real data matrix, $\mathbf{X}$, to obtain a perturbed (partially shuffled) data matrix $\mathbf{X}^\star$ which approximates the joint probability distribution of $\mathbf{X}$.

In line 4, Algorithm 2 computes the matrix of ranks from the shuffled data, $\mathbf{R}^\star$ (where each column of $\mathbf{R}^\star$ contains the ranks of each of variable (column) of the shuffled data).

Finally, in line 5 the algorithm performs a rank-matching operation which essentially replaces the shuffled rank values by synthetic marginal values with identical ranks. This last step essentially rearranges the row positions of each variable in the synthetic marginal data in a way that recovers the statistical association structure of the shuffled ranks (which, by their turn, approximate the association structure of the real data). This rank-matching operation was origi-

nally proposed by Domingo-Ferrer and Muralidhar (2016) in a different context. For completeness we describe it in Algorithm 10 in the Appendix.

## 4.2. The CategoricalSJPPDS Approach

The CategoricalSJPPDS approach (Algorithm 4) extends the SJPPDS to categorical data. At a high level, the basic procedure is to: (i) transform the categorical data to ranks; (ii) perform SJPPDS on the ranks; and (iii) transform back the shuffled ranks to categorical data.

---

**Algorithm 4** CategoricalSJPPDS($\mathbf{X}, n_c$)

1: **Input:** categorical data matrix $\mathbf{X}$; number of levels $n_c$
2: $\{map_\mathbf{R}, \mathbf{R}\} \leftarrow$ CategoricalToNumeric($\mathbf{X}$) {call Alg. 11}
3: $\mathbf{R}^\star \leftarrow$ SJPPDS($\mathbf{R}, n_c$) {call Algorithm 6 in the Appendix}
4: $\mathbf{X}_s \leftarrow$ NumericToCategorical($\mathbf{R}^\star, map_\mathbf{R}$) {call Alg. 12}
5: **Output:** synthetic data $\mathbf{X}_s$

---

Step (i) is performed in line 2 of the algorithm, which: (a) creates a mapping (denoted $map_\mathbf{R}$) between the values (levels) of each categorical variable in the dataset $\mathbf{X}$ and an (arbitrary) numeric rank variable; and (b) encodes the categorical data into the numeric rank data matrix $\mathbf{R}$ according to the mapping $map_\mathbf{R}$. (This procedure will be explained in more detail below.) Step (ii) is performed in line 3 of the algorithm, which applies SJPPDS to the encoded numerical rank data. Finally, step (iii) is performed in line 4 which transforms back the shuffled numeric rank data to categorical data using again the mapping created in line 2.

A detailed description of the numeric rank encoding procedure in line 2 of Algorithm 4 is presented in Algorithm 11 in the Appendix. But the basic idea is to encode the categorical variables using (arbitrary) numeric rank assignments (with ties broken at random). Table 2 provides an illustrative toy example.

Let $X_1 = (A, A, A, B, B, C, C, C, C)^t$ represent the first column of a categorical data matrix $\mathbf{X}$. This categorical variable has three levels $A$, $B$, and $C$ with counts $n_A = 3$, $n_B = 2$, and $n_C = 4$. To generate the numerical rank encoding $R_1$ presented in Table 2, the algorithm transforms the categorical values according to the mapping, $map_{R_1}$, in Table 2. Note that this mapping corresponds to an arbitrary ranking of the categorical levels according to the arbitrary order $A < B < C$. That is, starting with class $A$, the mapping assigns ranks 1, 2, and 3 (in random order) to the three tied elements $A$ in $X_1$. (Note that by assigning the ranks in random order the mapping is effectively using the "at random" method to break ties among identical values of the categorical variable.) Continuing with class $B$, the mapping assigns ranks 4 and 5 (in random order) to the two tied elements $B$ in $X_1$. Finally, for class $C$, the mapping assigns ranks 6, 7, 8, and 9 (in random order) to the four

*Table 2.* A toy example of the rank-based numeric encoding of categorical data performed by Algorithm 11 (in the Appendix).

| $map_{R_1}$ | $A = \{1, 2, 3\},$ | | $B = \{4, 5\},$ | | $C = \{6, 7, 8, 9\}$ | | | |
|---|---|---|---|---|---|---|---|---|
| $X_1$ | $A$ | $A$ | $A$ | $B$ | $B$ | $C$ | $C$ | $C$ | $C$ |
| $R_1$ | 3 | 1 | 2 | 5 | 4 | 7 | 6 | 9 | 8 |

*Table 3.* Rank-based numeric encoding for the alternative ordering, $C < A < B$, of the levels of the categorical variable $X_1$.

| $map_{R'_1}$ | $C = \{1, 2, 3, 4\},$ | | $A = \{5, 6, 7\},$ | | $B = \{8, 9\}$ | | | |
|---|---|---|---|---|---|---|---|---|
| $X_1$ | $A$ | $A$ | $A$ | $B$ | $B$ | $C$ | $C$ | $C$ | $C$ |
| $R'_1$ | 7 | 5 | 6 | 9 | 8 | 2 | 4 | 3 | 1 |

tied elements $C$ in $X_1$. Algorithm 11 essentially applies this encoding procedure to all columns of $\mathbf{X}$ to produce the numerical rank encoding matrix $\mathbf{R}$.

Once $\mathbf{R}$ is created, Algorithm 4 performs SJPPDS on $\mathbf{R}$ to generate a shuffled version of the encoded data (with a similar joint probability distribution as $\mathbf{R}$), which is then transformed back to (shuffled) categorical data using Algorithm 12 (presented in the Appendix).

Quite importantly, note that this numeric encoding procedure is necessarily arbitrary, since the levels of categorical variables have no intrinsic order, and distinct orderings generate distinct numerical rank encodings. (This is illustrated in Table 3, for the distinct ordering $C < A < B$, where $R'_1$ in Table 3 is different from $R_1$ in Table 2.)

This arbitrariness, however, does not affect the performance of the CategoricalSJPPDS approach because the same mapping used to transform $\mathbf{X}$ to $\mathbf{R}$ is also used to transform $\mathbf{R}^\star$ back to $\mathbf{X}_s$. Appendix F provides an illustration of this point.

## 4.3. The MixedSyntheticSJPPDS Approach

For mixed datasets, Algorithm 5 describes how to properly account for the joint statistical associations between numeric and categorical variables. (Note that separate applications of SyntheticSJPPDS to the numerical variables and of CategoricalSJPPDS to categorical variables preserves the associations among the numeric variables themselves and the associations among the categorical variables themselves, but destroys any associations between numerical and categorical variables.)

To also preserve the association structure between numeric and categorical variables Algorithm 5 first computes the numerical rank encoding matrix of the categorical data (line 6), before it applies SJPPDS to the concatenated data of numerical variables and numerical rank encodings (lines 7 and 8). By shuffling the concatenated data, the algorithm ensures the preservation of the joint probability distribution

of all variables (numeric and categorical). After this is done, the algorithm transforms back the shuffled ranks of the numerical variables to synthetic numerical data (lines 9, 10, and 11), and the shuffled rank encodings to shuffled categorical data (line 12).

---

**Algorithm 5** MixedSyntheticSJPPDS($\mathbf{X}$, $i_n$, $i_c$, $n_c$, $p$)

1: **Input:** data matrix $\mathbf{X}$; indexes of the numeric variables $i_n$; indexes of the categorical variables $i_c$; number of levels $n_c$; sampling proportion $p$
2: $\mathbf{X}_n \leftarrow \mathbf{X}[, i_n]$ {numerical dataset}
3: $\mathbf{X}_c \leftarrow \mathbf{X}[, i_c]$ {categorical dataset}
4: $nnms \leftarrow \text{ColumnNames}(\mathbf{X}_n)$ {names of numeric variables}
5: $cnms \leftarrow \text{ColumnNames}(\mathbf{X}_c)$ {names of categorical variables}
6: $\{map_{\mathbf{R}_c}, \mathbf{R}_c\} \leftarrow \text{CategoricalToNumeric}(\mathbf{X}_c)$ {obtain mapping and the numerical rank encoding for the categorical data}
7: $\mathbf{W} \leftarrow \text{Concatenate}(\mathbf{X}_n, \mathbf{R}_c)$ {concatenate the numeric data and numerical rank encoding of the categorical data}
8: $\mathbf{W}^\star \leftarrow \text{SJPPDS}(\mathbf{W}, n_c)$ {perform SJPPDS on the concatenated numeric data}
9: $\mathbf{M}_s \leftarrow \text{GenerateSyntheticMarginalsIOS}(\mathbf{X}_n, p)$ {create synthetic marginal distributions for the numeric variables}
10: $\mathbf{R}_n^\star \leftarrow \text{DataRankMatrix}(\mathbf{W}^\star[, nnms])$ {create matrix of ranks from the shuffled numeric variables}
11: $\mathbf{W}_n \leftarrow \text{MatchRanks}(\mathbf{M}_s, \mathbf{R}_n^\star)$ {rank-match the synthetic marginal data to the shuffled rank data from the numeric variables}
12: $\mathbf{W}_c \leftarrow \text{NumericToCategorical}(\mathbf{W}^\star[, cnms], map_{\mathbf{R}_c})$ {transform the shuffled numeric rank encoding data from the categorical variables back to categorical data}
13: $\mathbf{X}_s \leftarrow \text{Concatenate}(\mathbf{W}_n, \mathbf{W}_c)$ {concatenate the numeric and categorical matrices}
14: $\mathbf{X}_s \leftarrow \mathbf{X}_s[, \text{ColumnNames}(\mathbf{X})]$ {reorder the columns to match the order in the original data}
15: **Output:** synthetic data $\mathbf{X}_s$

---

# 5. Experiments

In our experiments, we compared synthetic data generators across 12 datasets (see Table 4) including 10 mixed datasets (evaluated in Hansen et al., 2023), 1 exclusively categorical dataset (Mushroom), and 1 exclusively numerical dataset (California housing). Further details about these datasets are provided in Appendix G. All datasets were randomly split into training and test sets of approximately same sizes. We performed two sets of experiments.

## 5.1. Experiment Set 1

In the first set of experiments, we compared TabSDS against 5 general purpose generators available in Synthcity, namely: tabular denoising diffusion probabilistic models (DDPM); adversarial random forests (ARF); tabular variational autoencoders (TVAE); conditional tabular generative adversarial networks (CTGAN); and bayesian networks (BayesNet).

*Table 4.* Datasets used in the experiments. (abbr: abbreviated name; # samples: number of samples; # num: number of numeric variables; # cat: number of categorical variables; task: learning task (regression or classification)) Number of variables corresponds to the number of features plus the target.

| abbr | name | # samples | # num | # cat | task |
|------|------|-----------|-------|-------|------|
| AB | Abalone | 4177 | 8 | 1 | regr. |
| AD | Adult | 45222 | 2 | 13 | class. |
| BM | Bank marketing | 10578 | 7 | 1 | class. |
| CH | California housing | 20640 | 9 | 0 | regr. |
| CR | Credit | 16714 | 10 | 1 | class. |
| DI | Diabetes 130US | 71090 | 7 | 1 | class. |
| EL | Electricity | 38478 | 7 | 1 | class. |
| EM | Eye movements | 7608 | 20 | 1 | class. |
| HO | House 16H | 13488 | 16 | 1 | class. |
| MT | Magic telescope | 13376 | 10 | 1 | class. |
| MU | Mushroom | 8124 | 0 | 21 | class. |
| PO | Pol | 10082 | 26 | 1 | class. |

These selected baselines correspond to a diverse and representative sample of tabular data generators in the literature, including strong baselines such as DDPM and ARF. (See additional comments regarding baseline selection in Appendix G.1.) Details about hyperparameter optimization for the baseline models are described in Appendix G.2.

The generator's performances were evaluated with respect to fidelity, utility, and privacy using Synthcity's `Benchmarks` function. Fidelity was measured using the `xgb` detection test metric which reports the AUROC for a XGBoost classifier trained to distinguish real and synthetic data. Utility was evaluated using ML efficiency based on a XGBoost learner trained on the synthetic data and evaluated on the real test data. Privacy was evaluated using the Domias MIA metric (van Breugel et al., 2023). All benchmark experiments were replicated 10 times.

For datasets containing mostly numeric variables, we also evaluated data privacy by comparing the distributions of the distance to closest record (DCR) between synthetic and real samples in terms of Euclidean distance.

## 5.2. Experiment Set 2

In the second set of experiments, we evaluated privacy risks with respect to DCR and two alternative metrics (implemented in R): the sorted version of the Distance Based Record Linkage (SDBRL) metric, which measures robustness to re-identification attacks; and the sorted version of the Standard Deviation Interval Distance (SSDID) metric, which measures robustness to attribute disclosure attacks. (See Appendix G.3 for details). Fidelity was again measured using the detection test. We considered the same baseline models as before, in addition to SMOTE (Chawla, 2002).

## 5.3. TabSDS Tuning Parameter Selection

TabSDS relies on only two tuning parameters. The first is the $p$ parameter in Algorithm 8 which controls the quality

of the synthetic marginal distributions by controlling the subsample size, $m$, used in Algorithm 3. Overall, we found that lower values of $p$ can lead to accentuated decreases in data fidelity without comparable increases in data privacy. (See Appendix G.9 for these evaluations.) For this reason we adopt $p = 0.5$ (the maximal value) in our evaluations. The second, and more critical parameter, is the number of levels/categories, $n_c$, used by the restricted permutations. This parameter controls the amount of shuffling applied to the data and allows for a more precise control of the trade-off between data fidelity/utility and data privacy.

The $n_c$ parameter can take any value between 1 and $n$ (the number of samples in the real dataset). Setting $n_c = 1$ leads to maximum shuffling, as it allows for unrestricted permutations of the data and completely removes the association between the variables. Larger values of $n_c$ constrain the amount of shuffling, leading to higher fidelity and decreased privacy. In the limit case when $n_c = n$, the data is not shuffled. (See Figure 11 in the Appendix for an illustration of the effect of $n_c$ on the synthetic data quality.) In our experiments, we evaluate TabSDS over a grid of 24 values, $n_c = \{5, 10, 15, \ldots 50, 60, 70, \ldots 100, 200, 300, \ldots, 1000\}$.

To illustrate the tradeoff between fidelity, utility, and privacy, Figures 2 and 3 report the detection test metric (panel a), ML efficiency (panel b), and domias MIA (panels c) for synthetic versions of the CH and EL datasets generated by the TabSDS approach over the grid of $n_c$ values above.

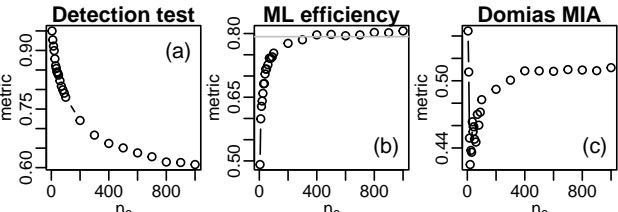

*Figure 2.* Benchmark metrics over the $n_c$ grid for the CH dataset.

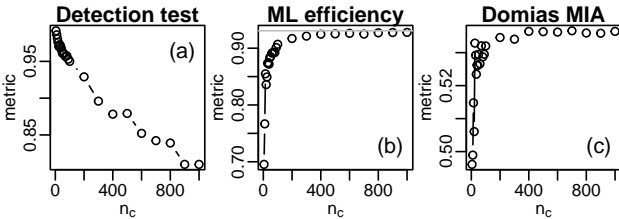

*Figure 3.* Benchmark metrics over the $n_c$ grid for the EL dataset.

To further illustrate the effect of $n_c$ on privacy, Figure 4 shows the distributions of the distance to the closest record (DCR) between the synthetic and real training samples (white boxplots) for datasets CH and EL. For comparison, it also shows the DCR distribution between the training and test set data (grey boxplots). Note that the grey boxplots provide an estimate of the typical DCR values we would expect to see from an ideal generator capable of drawing

independent samples from the same distribution as the training data (as our training and test sets have identical sizes). The red horizontal line corresponds to the median of the test set DCR distribution. DCR distributions whose medians are located below the red line show reduced privacy when compared to an ideal generator, while distributions whose medians are above the red line show increased privacy. For both datasets, we observe that low $n_c$ values tend to generate synthetic data which is more private than a truly i.i.d. dataset, but that larger $n_c$ values lead to synthetic datasets that are closer (in terms of DCR) to the training data. Interestingly, some datasets show a much faster decrease in DCR than others (compare panels b and a).

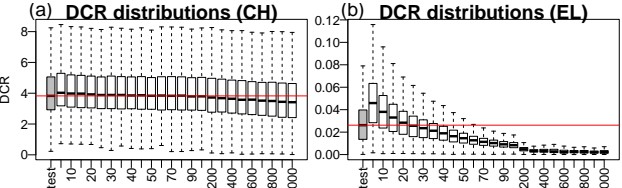

*Figure 4.* DCR distributions for the CH (a) and EL (b) datasets.

These results clearly illustrate that the quality of the synthetic data improves with increasing values of $n_c$ (as illustrated by the increasing ML efficiencies in Figures 2b and 3b, and decreasing ability to discriminate between real and synthetic data in Figures 2a and 3a). However, increases in $n_c$ also lead to lower privacy protection, as illustrated by the higher accuracies of the membership inference attacks in Figures 2c and 3c, and by DCR distributions concentrating closer to 0 in Figure 4.

This ability to exert a precise control over the trade-off between data utility/fidelity and data privacy is a very attractive feature of TabSDS. In practice, an analyst can inspect this trade-off over a grid of $n_c$ values across several utility, fidelity, and privacy metrics to select an adequate value for the intended application. For our comparisons, we selected $n_c$ as follows. For datasets containing mostly numerical variables we used the DCR distributions and selected the largest $n_c$ value in our grid which generates a DCR distribution with a median value close to the median of the test set DCR distribution. (In the examples in Figure 4, we select $n_c$ equal to 200 and 20.) For datasets containing mostly categorical variables we selected the largest $n_c$ which balanced well the tradeoff between the detection test, ML efficiency, and Domias MIA metrics. In Appendix G.5, we describe the tuning parameter selection for all 12 datasets evaluated in this work.

## 5.4. Results

### 5.4.1. EXPERIMENT SET 1

Tables 5, 6, and 7 report the benchmark results for the fidelity, utility, and privacy metrics, respectively. In each

table, the top 2 generators were highlighted in bold. Overall, TabSDS showed very competitive performance against strong baselines such as DDPM and ARF. In terms of fidelity and utility, in most datasets, it either ranked first or second, intercalating these top positions with DDPM and ARF. In terms of privacy, none of the generators stood-up as systematically outperforming the others.

*Table 5.* Fidelity comparisons as measured by the detection test metric. The table reports average AUROC and standard deviations (in parenthesis). Lower values imply better performance.

| dataset | TabSDS | DDPM | ARF | TVAE | CTGAN | BayesNet |
|---|---|---|---|---|---|---|
| AB | **0.837** | **0.878** | 0.981 | 0.995 | 0.994 | 1.000 |
|  | (0.021) | (0.007) | (0.004) | (0.002) | (0.002) | (0.000) |
| AD | **0.69** | **0.782** | 0.877 | 0.997 | 0.998 | 1.000 |
|  | (0.004) | (0.007) | (0.006) | (0.001) | (0.001) | (0.000) |
| BM | **0.693** | **0.697** | 0.996 | 0.999 | 0.999 | 0.998 |
|  | (0.043) | (0.005) | (0.001) | (0.000) | (0.000) | (0.000) |
| CH | **0.721** | **0.647** | 0.965 | 0.976 | 0.971 | 0.993 |
|  | (0.028) | (0.004) | (0.002) | (0.001) | (0.002) | (0.001) |
| CR | **0.852** | **0.769** | 0.997 | 1.000 | 1.000 | 0.999 |
|  | (0.006) | (0.021) | (0.000) | (0.000) | (0.000) | (0.000) |
| DI | **0.627** | **0.589** | 0.990 | 1.000 | 1.000 | 0.999 |
|  | (0.023) | (0.004) | (0.000) | (0.000) | (0.000) | (0.000) |
| EL | **0.976** | **0.802** | 0.980 | 0.998 | 0.999 | 0.999 |
|  | (0.001) | (0.003) | (0.001) | (0.000) | (0.000) | (0.000) |
| EM | **0.961** | **1.000** | 1.000 | 1.000 | 1.000 | 1.000 |
|  | (0.006) | (0.000) | (0.000) | (0.000) | (0.000) | (0.000) |
| HO | **0.752** | **0.748** | 0.942 | 0.987 | 0.989 | 0.998 |
|  | (0.006) | (0.009) | (0.002) | (0.001) | (0.002) | (0.000) |
| MT | **0.720** | **0.631** | 0.817 | 0.948 | 0.958 | 0.996 |
|  | (0.004) | (0.005) | (0.007) | (0.004) | (0.008) | (0.000) |
| MU | **0.664** | 0.954 | **0.700** | 0.887 | 0.872 | 0.908 |
|  | (0.005) | (0.039) | (0.008) | (0.021) | (0.060) | (0.005) |
| PO | **0.955** | **0.980** | 0.997 | 1.000 | 1.000 | 0.999 |
|  | (0.006) | (0.003) | (0.000) | (0.000) | (0.000) | (0.000) |

*Table 6.* Utility comparisons as measured by ML efficiency. The table reports average AUROC for classification tasks and average $R^2$ for regression tasks (with standard deviations reported in parenthesis). Higher values imply better performance.

| dataset | TabSDS | DDPM | ARF | TVAE | CTGAN | BayesNet |
|---|---|---|---|---|---|---|
| AB | **0.511** | **0.524** | 0.466 | 0.347 | 0.363 | -0.003 |
|  | (0.028) | (0.019) | (0.028) | (0.050) | (0.049) | (0.064) |
| AD | **0.885** | 0.877 | **0.882** | 0.845 | 0.851 | 0.863 |
|  | (0.003) | (0.003) | (0.003) | (0.006) | (0.008) | (0.004) |
| BM | **0.849** | **0.839** | 0.836 | 0.793 | 0.790 | 0.558 |
|  | (0.006) | (0.003) | (0.006) | (0.016) | (0.018) | (0.038) |
| CH | **0.777** | **0.769** | 0.658 | 0.583 | 0.595 | -0.156 |
|  | (0.005) | (0.004) | (0.012) | (0.014) | (0.012) | (0.021) |
| CR | 0.817 | **0.831** | **0.838** | 0.787 | 0.783 | 0.779 |
|  | (0.005) | (0.006) | (0.005) | (0.018) | (0.018) | (0.012) |
| DI | 0.611 | **0.621** | **0.625** | 0.561 | 0.579 | 0.616 |
|  | (0.005) | (0.003) | (0.004) | (0.023) | (0.031) | (0.005) |
| EL | 0.836 | **0.889** | **0.889** | 0.821 | 0.806 | 0.657 |
|  | (0.003) | (0.003) | (0.005) | (0.010) | (0.023) | (0.024) |
| EM | **0.600** | 0.485 | **0.533** | 0.540 | 0.523 | 0.530 |
|  | (0.019) | (0.034) | (0.032) | (0.027) | (0.027) | (0.019) |
| HO | **0.949** | **0.947** | 0.914 | 0.922 | 0.892 | 0.488 |
|  | (0.001) | (0.002) | (0.006) | (0.004) | (0.011) | (0.043) |
| MT | **0.912** | **0.918** | 0.885 | 0.862 | 0.843 | 0.497 |
|  | (0.004) | (0.004) | (0.009) | (0.010) | (0.013) | (0.039) |
| MU | **1.000** | 0.621 | **1.000** | 0.998 | 0.989 | 0.993 |
|  | (0.000) | (0.138) | (0.000) | (0.002) | (0.004) | (0.001) |
| PO | **0.99** | 0.911 | **0.968** | 0.931 | 0.848 | 0.905 |
|  | (0.002) | (0.007) | (0.003) | (0.018) | (0.056) | (0.007) |

Table 8 reports the combined runtime for model training and data generation (in seconds) of each generator across all datasets. (See Appendix G.7 for runtime experiment details.) In all datasets, the runtime for TabSDS was orders of magnitude faster than the deep generative models, and

*Table 7.* Privacy comparisons as measured by domias MIA. The table reports average AUROC and standard deviations (in parenthesis). Lower values imply better performance.

| dataset | TabSDS | DDPM | ARF | TVAE | CTGAN | BayesNet |
|---|---|---|---|---|---|---|
| AB | 0.531 | 0.541 | **0.523** | 0.555 | 0.555 | **0.503** |
|  | (0.017) | (0.019) | (0.023) | (0.046) | (0.046) | (0.038) |
| AD | 0.766 | 0.703 | 0.724 | 0.716 | **0.698** | **0.635** |
|  | (0.009) | (0.01) | (0.013) | (0.021) | (0.024) | (0.015) |
| BM | 0.377 | **0.332** | 0.361 | 0.376 | **0.360** | 0.400 |
|  | (0.003) | (0.007) | (0.006) | (0.009) | (0.023) | (0.010) |
| CH | 0.492 | 0.512 | 0.575 | **0.472** | **0.442** | 0.640 |
|  | (0.004) | (0.011) | (0.015) | (0.05) | (0.022) | (0.021) |
| CR | **0.345** | 0.477 | **0.334** | 0.399 | 0.362 | 0.380 |
|  | (0.006) | (0.083) | (0.01) | (0.035) | (0.030) | (0.011) |
| DI | 0.51 | 0.491 | 0.473 | **0.454** | 0.491 | **0.468** |
|  | (0.004) | (0.005) | (0.013) | (0.01) | (0.028) | (0.009) |
| EL | **0.506** | 0.521 | 0.536 | 0.519 | 0.527 | **0.431** |
|  | (0.003) | (0.005) | (0.01) | (0.013) | (0.031) | (0.016) |
| EM | - | - | - | - | - | - |
| HO | - | - | - | - | - | - |
| MT | 0.375 | **0.342** | **0.344** | 0.387 | 0.406 | 0.426 |
|  | (0.005) | (0.007) | (0.013) | (0.016) | (0.016) | (0.011) |
| MU | 0.646 | 0.612 | 0.612 | **0.603** | **0.572** | 0.636 |
|  | (0.018) | (0.129) | (0.029) | (0.046) | (0.057) | (0.027) |
| PO | 0.772 | 0.789 | **0.744** | **0.749** | 0.762 | 0.762 |
|  | (0.008) | (0.009) | (0.009) | (0.014) | (0.007) | (0.009) |

*Table 8.* Average runtime in seconds.

| data | samples | TabSDS | BayesNet | ARF | DDPM | TVAE | CTGAN |
|---|---|---|---|---|---|---|---|
| AB | 2088 | **1.33** | 2.86 | 24.52 | 282.69 | 444.13 | 172.78 |
| AD | 22611 | **5.43** | 9.50 | 304.17 | 699.89 | 1655.63 | 1046.77 |
| BM | 5289 | **2.99** | 3.87 | 17.38 | 93.21 | 314.36 | 277.35 |
| CH | 10320 | **7.76** | 21.99 | 82.89 | 980.76 | 261.92 | 997.33 |
| CR | 8357 | 9.23 | **6.43** | 35.15 | 195.42 | 543.87 | 504.83 |
| DI | 35545 | 22.98 | **14.06** | 116.92 | 616.87 | 1887.39 | 1148.02 |
| EL | 19237 | **11.17** | 11.77 | 86.42 | 337.81 | 1006.72 | 782.99 |
| EM | 3804 | **6.40** | 13.47 | 84.35 | 97.10 | 291.14 | 321.38 |
| HO | 6744 | **14.95** | 55.77 | 131.26 | 121.44 | 407.68 | 357.32 |
| MT | 6688 | **5.43** | 32.05 | 42.51 | 123.36 | 382.79 | 389.47 |
| MU | 4062 | **1.22** | 2.27 | 51.68 | 2203.03 | 327.70 | 726.17 |
| PO | 5041 | **10.31** | 13.90 | 202.36 | 94.95 | 367.63 | 428.78 |

also considerably faster than ARF. (In most datasets it even outperformed the BayesNet generator.)

### 5.4.2. EXPERIMENT SET 2

Figure 5 reports privacy vs fidelity tradeoff curves comparing the DCR, SDBRL, and SSSID privacy metrics against the detection test AUC fidelity metric. (Due to space limitations the figure only report results for the Abalone (AB) and Bank marketing (BM) datasets. The complete results for all datasets are presented in Figures 24 and 25 in Appendix G.6.) In addition to the baselines in the previous experiment these results also include comparisons against the SMOTE baseline (Chawla et al., 2002) based on $k = 5$ and $k = 20$ neighbors. Panels a and b shows the tradeoff between DCR vs detection AUC. The red vertical line represents the DCR score comparing the training and test sets and provides an estimate of the DCR value we would expect to see for an ideal generator able to draw i.i.d. data from the same distribution as the training data. The black curve represents the TabSDS results across the entire $n_c$ parameter grid, while the red dot corresponds to the TabSDS performance based on the $n_c$ value selected according to the DCR criterium (i.e., the DCR value closest to the test set DCR). Panels

c and d show the tradeoff plots comparing SDBRL vs detection AUC, while panels e and f report the tradeoff plots between SSDID vs detection AUC.

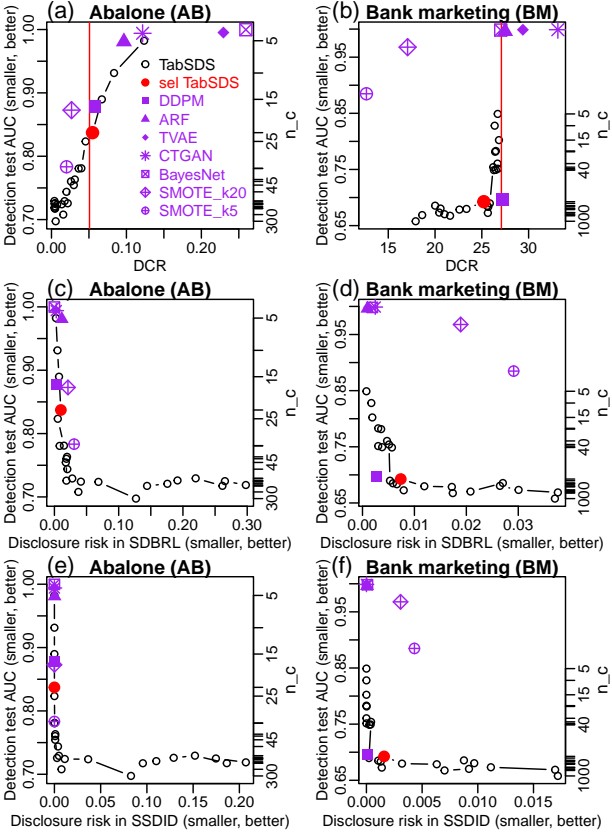

*Figure 5.* Privacy/utility tradeoff comparisons for the Abalone (AB) and Bank marketing (BM) datasets. See legend on panel a.

Overall, in these two datasets, DDPM (purple filled square), SMOTE (purple crossed diamond and purple crossed circle), and TabSDS (red filled circle) tended to outperform the other methods in terms of fidelity (note the lower AUC scores for the detection test in the y-axis). In terms of privacy, SMOTE tended to be worse than the other methods with respect to DCR (note the lower DCR values in panels a and b). In terms of SDBRL and SSDID, all methods tended to generate low risk scores with SMOTE tending to perform slightly worse than the other methods.

Appendix G.6 reports a more detailed evaluation across 10 datasets. Overall, similar results hold for the other datasets. In addition to the baselines investigated here, the results in the Appendix include additional comparisons against two differential privacy based generators, as well as, against a simple additive noise perturbation method.

Finally, Appendix G.8 presents qualitative comparisons of how well these different generators mimic the marginal distributions and statistical associations in the real data. (These last evaluations suggest that TabSDS can, overall,

generate more realistic data than alternative generators.)

# 6. Limitations

An important limitation of TabSDS is that, being a model free approach, it cannot be adapted to the generation of fair synthetic data. (State-of-the-art approaches such as DECAF (van Breugel et al., 2021) require modeling the synthetic data generation process according to a causal model representation of the data generation process). The main application of TabSDS is to facilitate data sharing, where synthetic versions of real data can be more readily shared. (This application is what motivated its development.)

A minor caveat of the proposed IOSSampler method (Algorithm 3) for generating marginal distributions is that for discontinuous multimodal distributions, where training values cluster near separate modes with no training examples in between, the method may generate a small fraction of intermediate values outside the training data range (see Figure 69 for an illustration). In these situations, a post-processing step could be easily added to remove values outside the range of the training data.

# 7. Conclusions and Final Remarks

We introduce a simple, lightweight, fully non-parametric, and model free approach for the generation of synthetic tabular data. The work extends data perturbation methodology from the statistical disclosure control field, and provides very competitive performance against strong baselines in the ML field, at a fraction of the runtime.

While the interpolated order statistics approach proposed in this paper performed well in our experiments, an interesting future research avenue is to evaluate alternative approaches for the generation of the synthetic marginal distributions. (Note that any method for generating uncorrelated synthetic marginals can be directly plugged into line 2 of Algorithm 2. Since the generation of marginal synthetic data should, in principle, be easier than the generation of joint distributions, this could lead to the development of hybrid approaches, where the uncorrelated marginals are generated with parametric models, while the association structure is recovered in a non-parametric way.) Another interesting future research direction is the incorporation of differential privacy into the TabSDS approach.

Finally, note that while the TabSDS approach treats any numerical variables as real valued, for any integer valued variables in a dataset, we recommend the use of a post-processing step for rounding the data of these variables to the nearest integer value.

R and Python implementations of TabSDS are available at `https://github.com/echaibub/TabSDS`.

## Impact Statement

This paper presents work whose goal is to advance the field of Machine Learning. There are many potential societal consequences of our work, none which we feel must be specifically highlighted here.

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

## A. SJPPDS Algorithms

For completeness, we present below the SJPPDS algorithm. (See Chaibub Neto (2024) for a detailed description of the algorithm.) While the original paper proposes two versions of joint probability preserving data shuffling, namely, the "full" and "simplified" data shuffles, in this paper we only consider the simplified version, as it is considerably faster than the full one without incurring any loss in performance. (Chaibub Neto (2024) recommends the use of the simplified version in practical applications.)

---

**Algorithm 6** Sequential joint probability preserving data shuffling (SJPPDS)

1: **Input:** microdata, $\boldsymbol{X}$; number of classes/levels, $n_c$
2: Find the number of attributes, $p$, of $\boldsymbol{X}$, i.e., $p \leftarrow$ NumberOfColumns($\boldsymbol{X}$)
3: Create the categorical version of $\boldsymbol{X}$, denoted $\boldsymbol{C}$, by discretizing each column of $\boldsymbol{X}$ into $n_c$ categories: $\boldsymbol{C} \leftarrow$ CategorizeData($\boldsymbol{X}, n_c$)
4: Generate the masked data, $\boldsymbol{X}^\star$, by using the JPPDS algorithm: $\boldsymbol{X}^\star \leftarrow$ JointProbabilityPreservingDataShuffling($\boldsymbol{X}, \boldsymbol{C}$)
5: **for** $i = 1$ **to** $p - 1$ **do**
6:     Change the order of the columns of $\boldsymbol{X}^\star$, so that the first column is placed last, i.e., $\boldsymbol{X}^\star \leftarrow \boldsymbol{X}^\star[, c(2 : p, 1)]$
7:     Update the categorical version of the masked microdata, i.e., $\boldsymbol{C}^\star \leftarrow$ CategorizeData($\boldsymbol{X}^\star, n_c$)
8:     Update the masked microdata, i.e., $\boldsymbol{X}^\star \leftarrow$ JointProbabilityPreservingDataShuffling($\boldsymbol{X}^\star, \boldsymbol{C}^\star$)
9: **end for**
10: Restore column order to the order in the original dataset, i.e., $\boldsymbol{X}^\star \leftarrow \boldsymbol{X}^\star[, c(2 : p, 1)]$
11: **Output:** return the masked dataset $\boldsymbol{X}^\star$

---

**Algorithm 7** Joint probability preserving data shuffling - simplified version (JPPDS-s)

1: **Input:** microdata, $\boldsymbol{X}$; categorical version of the microdata, $\boldsymbol{C}$
2: Find the number of attributes, $p$, of $\boldsymbol{X}$, i.e., $p \leftarrow$ NumberOfColumns($\boldsymbol{X}$)
3: Find the number of records, $n$, of $\boldsymbol{X}$, i.e., $n \leftarrow$ NumberOfRows($\boldsymbol{X}$)
4: Initialize the matrix with shuffled data with the original microdata, i.e., $\boldsymbol{X}_s \leftarrow \boldsymbol{X}$
5: Get the last column of $\boldsymbol{C}$, i.e., $C_p \leftarrow \boldsymbol{C}[, p]$
6: Get the unique values of $C_p$, i.e., $C_{pu} \leftarrow$ Unique($C_p$)
7: Get the length of the $C_{pu}$ vector, i.e., $n_u \leftarrow$ Length($C_{pu}$)
8: **for** $j = 1$ **to** $n_u$ **do**
9:     Find the indexes of $C_p$ that have the value in $C_{pu}[j]$, i.e., $idx \leftarrow$ Which($C_p == C_{pu}[j]$)
10:     Obtain a randomly shuffled version of the indexes in $idx$, i.e., $idx_s \leftarrow$ Shuffle($idx$)
11:     For the first $p - 1$ columns, replace the data in the rows of $\boldsymbol{X}_s$ indexed by $idx$ by the data in rows $idx_s$ of $\boldsymbol{X}$, i.e., $\boldsymbol{X}_s[idx, 1 : (p - 1)] \leftarrow \boldsymbol{X}[idx_s, 1 : (p - 1)]$
12: **end for**
13: Randomly shuffle the rows of $\boldsymbol{X}_s$, i.e., $\boldsymbol{X}_s \leftarrow \boldsymbol{X}_s[\text{Shuffle}(1 : n), ]$
14: **Output:** return the shuffled dataset $\boldsymbol{X}_s$

---

## B. Additional Algorithms for Generating Synthetic Marginals via Interpolated Order Statistics

Here, we describe in detail the generation of the synthetic marginal distributions. Algorithm 8 is basically a wrapper function for separately applying Algorithm 9 to each one of the variables (columns) of the data matrix $\mathbf{X}$.

Algorithm 9 and Algorithm 3 (in the main text) describe the interpolation approach (based on order statistics) used to generate synthetic data from the marginal distributions of each variable. For each variable, the basic idea is to: (i) take 2 random subsamples of size $m$ of the variable data (as described in lines 8 to 10 of Algorithm 9); (ii) sort the 2 subsamples as described in lines 2 and 3 of Algorithm 3 (so that the $i$th element of each subsample correponds to the $i$th order statistic of the subsample); and (iii) for each index of the sorted subsamples, the algorithm generates synthetic data by randomly sampling a value within the range of these two values (as described in lines 5 to 9 of Algorithm 3).

Since Algorithm 3 generates only $m$ synthetic data-points, in order to generate $n$ samples, Algorithm 9 repeatedly calls Algorithm 3 until it generates $n$ (or more) synthetic data-points. (The number of calls is given by $\lceil n/m \rceil$ as described in line 5 of Algorithm 9). If the total number of synthetic samples generated over the $\lceil n/m \rceil$ calls is larger than $n$, the algorithm

randomly sample $n$ of these points (as described in line 14).

---

**Algorithm 8** GenerateSyntheticMarginalsIOS($\mathbf{X}$, $p$)

---

1: **Input:** data matrix $\mathbf{X}$; sampling proportion $p$
2: $n \leftarrow$ NumberOfRows($\mathbf{X}$) {find number of rows}
3: $k \leftarrow$ NumberOfCols($\mathbf{X}$) {find number of columns}
4: $\mathbf{M}_s \leftarrow [,]$ {create empty matrix}
5: **for** $j = 1$ **to** $k$ **do**
6:    $\mathbf{M}_s[,j] \leftarrow$ InterpolatedOrderStatsSampling($\mathbf{X}[,j]$) {generate interpolated synthetic marginal data for the $j$th variable (Algorithm 9)}
7: **end for**
8: **Output:** matrix of synthetic marginals $\mathbf{M}_s$

---

**Algorithm 9** InterpolatedOrderStatsSampling($\mathbf{x}$, $p$)

---

1: **Input:** data vector $\mathbf{x}$; sampling proportion $p$
2: $n \leftarrow$ Length($\mathbf{x}$) {get length of $\mathbf{x}$}
3: $seq_n \leftarrow$ Sequence($1, n$) {get a sequence of integers from 1 to $n$}
4: $m \leftarrow$ Floor($n * p$) {round down $n * p$ to the nearest integer}
5: $ndraws \leftarrow$ Ceiling($n/m$) {round up $n/m$ to the nearest integer to get the number of draws}
6: $\mathbf{Y} \leftarrow [,]$ {create empty matrix}
7: **for** $i = 1$ **to** $ndraws$ **do**
8:    $idx_1 \leftarrow$ SampleWithoutReplacement($seq_n, m$) {sample $m$ indexes out of the $seq_n$ sequence}
9:    $idx_r \leftarrow$ SetDifference($seq_n, idx_1$) {compute the set difference to obtain the remaining indexes}
10:    $idx_2 \leftarrow$ SampleWithoutReplacement($idx_r, m$) {sample $m$ out of the remaining indexes}
11:    $\mathbf{Y}[i,] \leftarrow$ IOSSampler($\mathbf{x}_1[idx_1], \mathbf{x}_2[idx_2]$) {get interpolated sample}
12: **end for**
13: $\mathbf{y} \leftarrow$ Vector($\mathbf{Y}$) {vectorize the $\mathbf{Y}$ matrix}
14: $\mathbf{y} \leftarrow$ SampleWithoutReplacement($\mathbf{y}, n$) {take a random sample of $n$ elements from the $\mathbf{y}$ vector}
15: **Output:** vector of interpolated samples $\mathbf{y}$

---

By randomly interpolating between the values of each order statistic this approach generates synthetic data which approximates the marginal distribution of the original variable data. Intuitively, the quality of the approximation depends on the size of the subsamples, $m$, with higher values leading to better approximations. As described in line 4 of Algorithm 9, $m$ is computed as $m = \lfloor np \rfloor$, where $n$ is the number of data points in the dataset and $\frac{1}{n} \leq p \leq 0.5$.

To illustrate how the subsample size affects the quality of the synthetic data, in Figures 6 and 7 we apply Algorithm 9 (and Algorithm 3) to the same data using $p = 0.5$ and $p = 0.25$.

In these examples we use a random sample of size $n = 100$ from the latitude variable in the California housing (CH) data, as our real data. The blue histograms in Figures 6c and 7e show the real data distribution.

By adopting $p = 0.5$, we have that $m = 50$, and Algorithm 9 needs to call Algorithm 3 two times in order to generate 100 synthetic datapoints. Figures 6a and b show the synthetic data (red points) generated by these two calls. In these panels, $X_{(i)}$ represent the sorted real data, $\tilde{X}_{(i)}$ the sorted synthetic data, and $Y_{(i)}$ and $Z_{(i)}$ represent the sorted subsamples of size $m = 50$ randomly drawn from $X$. The grey dotted lines connect the $i$th order statistics in subsamples $Y$ and $Z$ (which corresponds to the $\mathbf{x}_1$ and $\mathbf{x}_2$ vectors in the notation of Algorithm 3). The red histogram in Figure 6c shows the distribution of the synthetic data (computed by pooling together the red dots in panels a and b). Finally, Figure 6d shows the quantile-quantile plot comparing the real and synthetic data distributions.

By adopting $p = 0.25$, we have that $m = 25$, and Algorithm 3 needs to be called four times to generate 100 synthetic datapoints. Figures 7a to d show the synthetic data generated by these four calls. Figure 7e shows the histograms of the synthetic data and real data, and Figure 7f shows the respective quantile-quantile plot.

Comparisons of the synthetic data histograms in Figures 6c and 7e, as well as, the quantile-quantile plots in Figures 6d and 7f, clearly show that lower values of $p$ (or, smaller subsample sizes) lead, as expected, to noisier synthetic data versions of the real data.

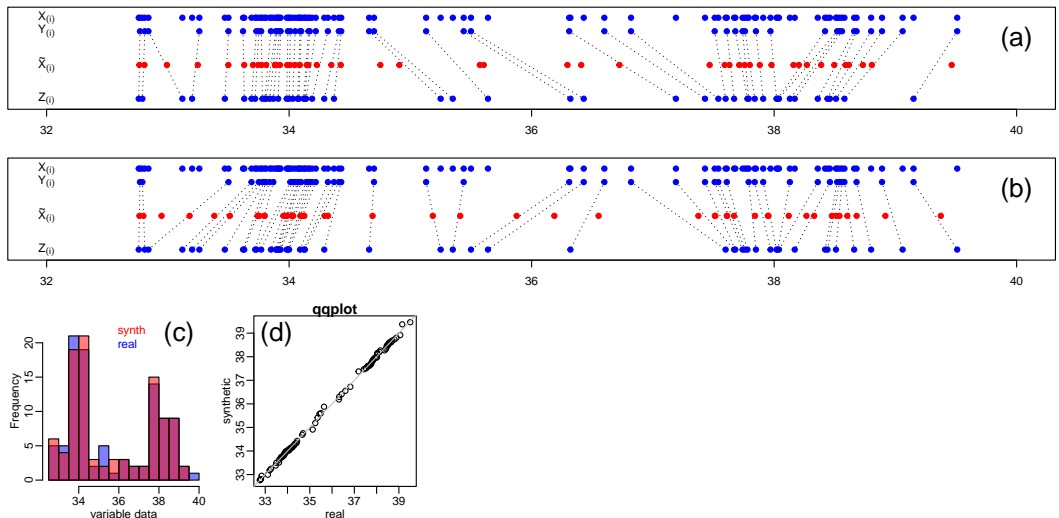

*Figure 6.* Illustrative application of Algorithm 9 with $p = 0.5$.

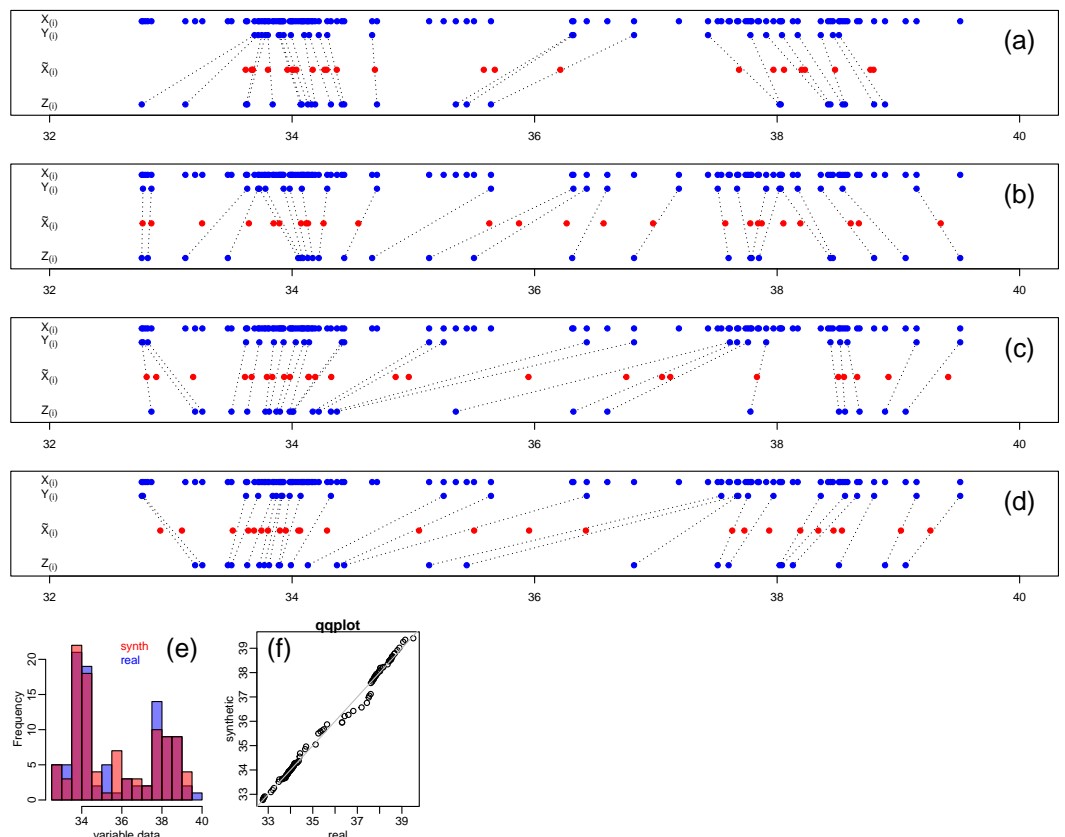

*Figure 7.* Illustrative application of Algorithm 9 with $p = 0.25$.

# C. Proof of Theorem 1

Consider a numeric random variable $X$ distributed as $X \sim F_X$. Let $Y$ and $Z$ represent two subsamples of size $m$ from $X$. Let,

$$\hat{F}_m^Y(t) = \frac{1}{m} \sum_{i=1}^{m} \mathbb{1}\{Y_i \leq t\} \,, \qquad \hat{F}_m^Z(t) = \frac{1}{m} \sum_{i=1}^{m} \mathbb{1}\{Z_i \leq t\} \,, \tag{1}$$

represent, respectively, the empirical cumulative distribution functions of the $Y$ and $Z$ subsamples, where $\mathbb{1}$ represents the indicator function. Let,

$$
\begin{aligned}
Y_{(1)} &< Y_{(2)} < Y_{(3)} < \ldots < Y_{(m-1)} < Y_{(m)} \,, \\
Z_{(1)} &< Z_{(2)} < Z_{(3)} < \ldots < Z_{(m-1)} < Z_{(m)} \,,
\end{aligned}
\tag{2}
$$

represent the sorted subsamples, where $Y_{(i)}$ and $Z_{(i)}$ represent the $i$th elements of the sorted subsamples (i.e., the $i$th order statistics of the subsamples).

Since the interpolated order statistics samples generated by Algorithm 3 are draw from,

$$\tilde{X}_{(i)} \sim \text{Uniform}\Big( \min\{Y_{(i)}, Z_{(i)}\} \,, \max\{Y_{(i)}, Z_{(i)}\} \Big) \,, \tag{3}$$

we have that, by construction,

$$\min\{Y_{(i)}, Z_{(i)}\} \leq \tilde{X}_{(i)} \leq \max\{Y_{(i)}, Z_{(i)}\} \,. \tag{4}$$

Taken together the constraints in equations (2) and (4) imply that for all $t$, the empirical cumulative distribution function of the interpolated samples, $\hat{F}_m^{\tilde{X}}(t)$, satisfy the relation,

$$\min\{\hat{F}_m^Y(t), \hat{F}_m^Z(t)\} \leq \hat{F}_m^{\tilde{X}}(t) \leq \max\{\hat{F}_m^Y(t), \hat{F}_m^Z(t)\} \,. \tag{5}$$

See Figure 8 for an illustrative example.

It follows directly from Glivenko-Cantelli theorem that both $\hat{F}_m^Y$ and $\hat{F}_m^Z$ converge almost surely to $F_X$, as $m$ converges to infinity. Now, reexpressing,

$$\min\{\hat{F}_m^Y(t), \hat{F}_m^Z(t)\} = \frac{\hat{F}_m^Y(t) + \hat{F}_m^Z(t) - |\hat{F}_m^Y(t) - \hat{F}_m^Z(t)|}{2} \,, \tag{6}$$

$$\max\{\hat{F}_m^Y(t), \hat{F}_m^Z(t)\} = \frac{\hat{F}_m^Y(t) + \hat{F}_m^Z(t) + |\hat{F}_m^Y(t) - \hat{F}_m^Z(t)|}{2} \,, \tag{7}$$

and taking the limit over $m$, it follows that both $\min\{\hat{F}_m^Y(t), \hat{F}_m^Z(t)\}$ and $\max\{\hat{F}_m^Y(t), \hat{F}_m^Z(t)\}$ converge to $F_X$. Hence, from (5) it follows that $\hat{F}_m^{\tilde{X}}$ is bounded from above and from below by $F_X$ and, therefore, also converges almost surely to $F_X$.

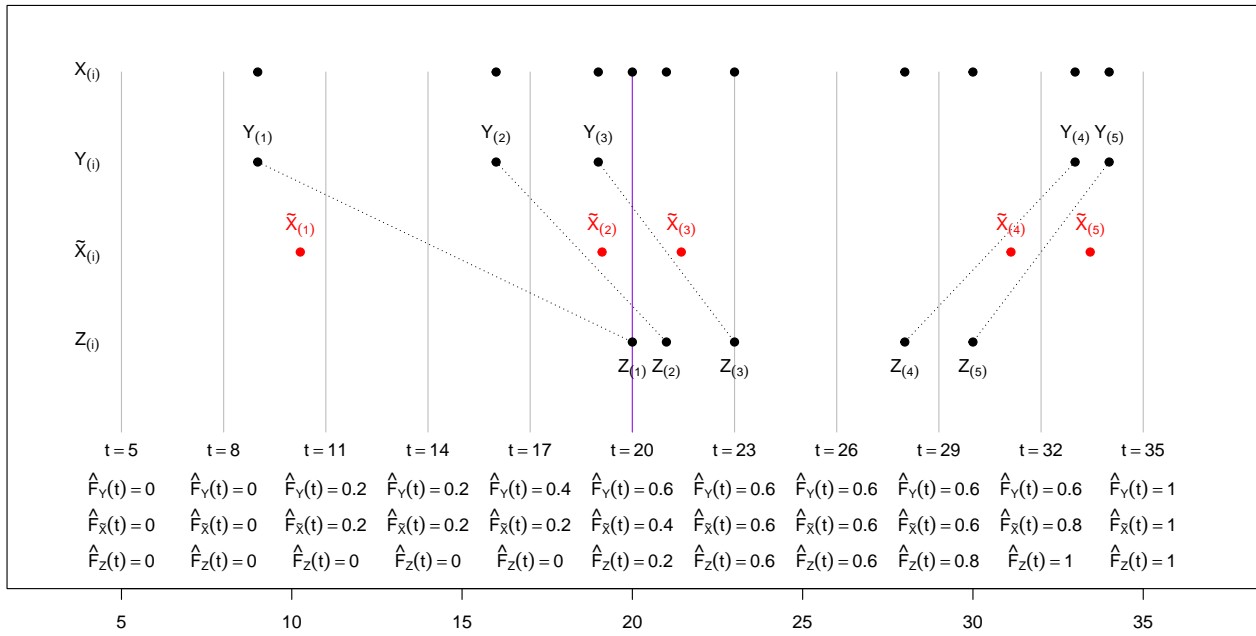

*Figure 8.* Consider, without loss of generality, a sorted variable $X_{(i)} = (9, 16, 19, 20, 21, 23, 28, 30, 33, 34)$ (shown by the black dots at the top row of the figure). Let $Y_{(i)}$ and $Z_{(i)}$ represent two sorted random subsamples of size 5 from $X$, and $\tilde{X}_{(i)}$ the synthetic data generated by randomly drawing values from $\text{Uniform}(\min\{Y_{(i)}, Z_{(i)}\}, \max\{Y_{(i)}, Z_{(i)}\})$, for $i = 1, \ldots, 5$. For any value of $t$, the values of the empirical cumulative distribution functions of $Y$, $\tilde{X}$, and $Z$ correspond, graphically, to the proportion of dots at or to the left of the vertical line set at $t$. For instance, for $t = 20$ (the purple vertical line), there are 3 points to the left of the line for variable $Y$, 2 points to the left for variable $\tilde{X}$, and 1 point at the line for variable $Z$, implying that $\hat{F}_Y(20) = 0.6$, $\hat{F}_{\tilde{X}}(20) = 0.4$, and $\hat{F}_Z(20) = 0.2$. Inspection of the figure shows that the relation $\min\{\hat{F}_Y(t), \hat{F}_Z(t)\} \leq \hat{F}_{\tilde{X}}(t) \leq \max\{\hat{F}_Y(t), \hat{F}_Z(t)\}$ holds for any value of $t$ in the real line.

## D. Rank-Matching Algorithm

---
**Algorithm 10** MatchRanks($\mathbf{M}_s$, $\mathbf{R}$)
---
1: **Input:** matrix of synthetic marginal data, $\mathbf{M}_s$; matrix of ranks from the original data, $\mathbf{R}$
2: $\mathbf{X}_s \leftarrow [,]$ {create empty matrix}
3: $n \leftarrow \text{NumberOfRows}(\mathbf{R})$ {find number of rows}
4: $k \leftarrow \text{NumberOfCols}(\mathbf{R})$ {find number of columns}
5: **for** $j = 1$ **to** $k$ **do**
6:     $\mathbf{r}_j \leftarrow \text{Rank}(\mathbf{M}_s[, j])$ {get ranks of the $j$th column of $\mathbf{M}_s$ (ties are broken at random)}
7:     **for** $i = 1$ **to** $n$ **do**
8:         $r_{ij} \leftarrow \mathbf{R}[i, j]$ {get the rank of the $i$th element of the $j$th variable in the original data}
9:         $idx \leftarrow \text{Which}(\mathbf{r}_j == r_{ij})$ {find which element of the vector $\mathbf{r}_j$ has the same rank as $r_{ij}$}
10:         $\mathbf{X}_s[i, j] \leftarrow \mathbf{M}_s[idx, j]$ {assign the corresponding element of $\mathbf{M}_s$ to the rank matched matrix}
11:     **end for**
12: **end for**
13: **Output:** rank matched matrix $\mathbf{X}_s$
---

## E. Additional Algorithms for the CategoricalSJPPDS Approach

Algorithm 11 describes how to create a mapping between categorical variables and their (arbitrary) numeric rankings (which can then be used to create a numerical rank encoding of the categorical data). For the sake clarity, we will walk through the algorithm using the toy example in the main text, where we assume that the $i$th column of $\mathbf{X}$ is given by $X_i = (A, A, A, B, B, C, C, C, C)^t$.

---

**Algorithm 11** CategoricalToNumeric($\mathbf{X}$)

---

1: **Input:** categorical data matrix $\mathbf{X}$
2: $numvar \leftarrow$ NumberOfColumns($\mathbf{X}$)
3: $map_{\mathbf{R}} \leftarrow \{\}$ {create empty map}
4: $\mathbf{R} \leftarrow [,]$ {create empty matrix}
5: **for** $i = 1$ **to** $numvar$ **do**
6: $\quad levels \leftarrow$ ExtractLevels($\mathbf{X}[, i]$) {obtain levels of variable $i$}
7: $\quad tb \leftarrow$ Table($\mathbf{X}[, i]$) {obtain level counts of variable $i$}
8: $\quad cumcounts \leftarrow$ CumulativeSum($tb$) {compute cumulative counts from the table counts}
9: $\quad numlevels \leftarrow$ Length($levels$) {obtain number of levels}
10: $\quad tmp \leftarrow [,]$ {create empty matrix to store the mapping for variable $i$}
11: $\quad$ **for** $j = 1$ **to** $numlevels$ **do**
12: $\quad\quad idx \leftarrow$ Which($\mathbf{X}[, i] == levels[j]$) {obtain the indexes of the records for which variable $i$ equals level $j$}
13: $\quad\quad lowerbound \leftarrow cumcounts[j] + 1$ {compute the lower bound for the numerical encoding of level $j$ for variable $i$}
14: $\quad\quad upperbound \leftarrow cumcounts[j + 1]$ {compute the upper bound for the numerical encoding of level $j$ for variable $i$}
15: $\quad\quad nrseq \leftarrow$ Sequence($lowerbound, upperbound$) {create a sequence of numeric rank values starting at lower-bound and ending at upper-bound}
16: $\quad\quad \mathbf{R}[idx, i] \leftarrow$ RandomPermutation($nrseq$) {assign randomly shuffled numeric rank values to the positions of variable $i$ corresponding to level $j$}
17: $\quad\quad tmp[j,] \leftarrow [levels[j], lowerbound, upperbound]$ {store the mapping for level $j$}
18: $\quad$ **end for**
19: $\quad map_{\mathbf{R}}[i] \leftarrow tmp$ {store the mapping for variable $i$}
20: **end for**
21: **Output:** mapping $map_{\mathbf{R}}$; numerical rank encoding $\mathbf{R}$

---

From line 6 we have that $X_i$ has $levels = [A, B, C]$. From line 7, the counts of each of these levels are $n_A = 3$, $n_B = 2$, and $n_C = 4$. From line 8, the cumulative counts are $cumcounts = [0, 3, 5, 9]$.

For $j = 1$, we have: from line 12 that the indexes of the rows containing level $A$ are $idx = [1, 2, 3]$; from line 13 that the lower-bound is 1; from line 14 that the upper-bound is 3; from line 15 that the sequence of numeric rank values are given by $[1, 2, 3]$. (Hence, the mapping between level $A$ and the numeric rankings is given by $A = \{1, 2, 3\}$.) From line 16 we see that the algorithm first randomly shuffles the numeric rank values before assigning them to the positions containing the $A$ level. This ensures that the algorithm is effectively using the "at random" method to break ties among identical values. In line 17 the algorithm is storing the mapping for the $A$ level in the table format [level, lower-bound, upper-bound], namely,

| level | lower bound | upper bound |
|-------|-------------|-------------|
| A | 1 | 3 |

For $j = 2$, we have that: the indexes of the rows containing level $B$ are $idx = [4, 5]$ (line 12); the lower-bound is 4 (line 13); the upper-bound is 5 (line 14); the sequence of numeric rank values is given by $[4, 5]$, so that the mapping between level $B$ and the numeric rankings is given by $B = \{4, 5\}$ (line 15),

| level | lower bound | upper bound |
|-------|-------------|-------------|
| A | 1 | 3 |
| B | 4 | 5 |

For $j = 3$, we have that: the indexes of the rows containing level $C$ are $idx = [6, 7, 8, 9]$ (line 12); the lower-bound is 6 (line 13); the upper-bound is 9 (line 14); the sequence of numeric rank values is given by $[6, 7, 8, 9]$, so that the mapping between level $C$ and the numeric rankings is given by $B = \{6, 7, 8, 9\}$ (line 15),

| level | lower bound | upper bound |
|-------|-------------|-------------|
| A | 1 | 3 |
| B | 4 | 5 |
| C | 6 | 9 |

$$(8)$$

Note that the above table (8) corresponds to the $i$th element of $map_{\mathbf{R}}$.

Algorithm 12 performs the reverse transformation and given a numerical rank encoding matrix, $\mathbf{R}$, and the same mapping created in Algorithm 11, it transforms back the numeric rank encodings to the corresponding categorical values. Essentially, for each categorical variable (column of $\mathbf{R}$), the algorithm first reads in the levels of the variable stored in the map (line 6), and then for each of these levels, the algorithm finds the indexes of the elements within the lower and upper rank bounds of the variable level (line 9), and then assign that level to the matrix storing the categorical data (line 10).

---

**Algorithm 12** NumericToCategorical($\mathbf{R}$, $map_{\mathbf{R}}$)

---

  1: **Input:** encoded numeric rank data matrix $\mathbf{R}$; mapping $map_{\mathbf{R}}$
  2: $numvar \leftarrow$ NumberOfColumns($\mathbf{R}$)
  3: $\mathbf{X}_s \leftarrow [,]$ {create empty matrix}
  4: **for** $i = 1$ **to** $numvar$ **do**
  5:     $map_i \leftarrow map_{\mathbf{R}}[i]$ {grab map of variable $i$}
  6:     $levels \leftarrow$ ExtractLevels($map_i$) {extract levels of var. $i$}
  7:     $numlevels \leftarrow$ Length($levels$) {obtain number of levels}
  8:     **for** $j = 1$ **to** $numlevels$ **do**
  9:        $idx \leftarrow$ Which($map_i[lowerbound] \leq \mathbf{R}[,i] \leq map_i[upperbound]$) {obtain the indexes of the records}
 10:        $\mathbf{X}_s[idx, i] \leftarrow levels[j]$ {assign level $j$}
 11:     **end for**
 12: **end for**
 13: **Output:** categorical data $\mathbf{X}_s$

---

## F. Arbitrary Numeric Rank Encodings Have no Impact in the CategoricalSJPPDS Algorithm

As described in the main text, numeric rank encodings generated by Algorithm 11 are necessarily arbitrary, since the levels of categorical variables have no intrinsic order, and distinct orderings generate distinct numerical rank encodings. This arbitrariness, however, does not affect the performance of the CategoricalSJPPDS approach because the same mapping used to transform $\mathbf{X}$ to $\mathbf{R}$ is also used to transform $\mathbf{R}^\star$ back to $\mathbf{X}_s$.

To illustrate this point we generate synthetic data from a subset of 7 randomly selected variables from the Mushroom (MU) dataset using 2 distinct numeric rank encoding maps, $map_{\mathbf{R}_a}$ and $map_{\mathbf{R}_b}$. The alternative rank encodings were obtained by modifying Algorithm 11 so that the order of variable levels obtained in line 6 is randomly shuffled.

For the rank encoding **a**, the modified algorithm generates a numeric rank encoding matrix $\mathbf{R}_a$, whose Pearson correlation matrix is shown in Figure 9a. For the alternative rank encoding **b**, the modified algorithm generates a numeric rank encoding matrix $\mathbf{R}_b$, whose Pearson correlation matrix is shown in Figure 9d.

Application of SJPPDS to $\mathbf{R}_a$ and $\mathbf{R}_b$ generate, respectively, the shuffled numeric rank encoding matrices $\mathbf{R}_a^\star$ and $\mathbf{R}_b^\star$, whose correlation matrices are shown in Figures 9b and e, respectively.

Note that while the correlation structure of the numerical encoding matrices $\mathbf{R}_a$ and $\mathbf{R}_b$ are quite distinct (compare Figure 9a to 9d), the SJPPDS approach generates shuffled versions $\mathbf{R}_a^\star$ and $\mathbf{R}_b^\star$ which closely approximate the association structures of the respective $\mathbf{R}_a$ and $\mathbf{R}_b$ matrices (compare Figure 9a to 9b and Figure 9d to 9e).

Application of Algorithm 12 to $\mathbf{R}_a^\star$ and $\mathbf{R}_b^\star$ generates synthetic categorical datasets $\mathbf{X}_s^a$ and $\mathbf{X}_s^b$ whose pairwise association matrices (computed using the Cramer-V statistic) are shown, respectively, in Figures 9c to 9f. (For comparison, Figure 10 shows the Cramer-V matrix computed on the real data, illustrating that both numeric rank encodings generate similar statistical associations to the real categorical data.)

This example, illustrates that the choice of the numerical rank encoding used to transform the categorical data is immaterial.

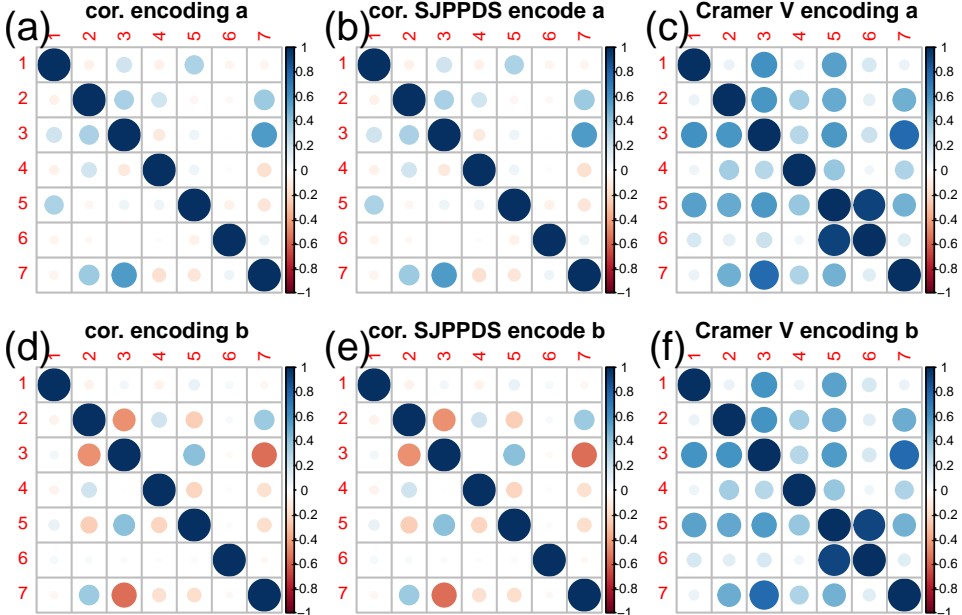

*Figure 9.* Performance of the CategoricalSJPPDS approach under alternative numerical rank encodings.

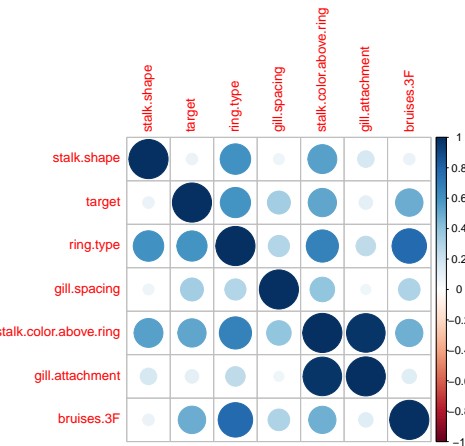

*Figure 10.* Pairwise association structure (as measure by the Cramer V statistic) computed on the real data for the 7 randomly selected variables from the Mushroom (MU) dataset. The association structure from the synthetic data in Figure 9 c and f closely matches the association structure in the real data.

## G. Extended Experimental Results Section

Experiments were based on 12 datasets described briefly in Table 4 in the main text. These datasets included 10 mixed datasets, 1 exclusively categorical dataset (Mushroom), and 1 exclusively numerical dataset (California housing).

The Abalone (AB) and Adult (AD) datasets were fetched from openML using the sklearn.datasets module by calling,

```
from sklearn.datasets import fetch_openml

fetch_openml(name="abalone", version=1, as_frame=True) # Abalone (AB)

fetch_openml(name="adult", as_frame=True, version=1) # Adult (AD)
```

For the Adult dataset we removed rows containing missing values and recoded some of the variables (see code for details) as described in Hansen et al. (2023).

The California housing (CH) dataset was fetched using the command,

```
from sklearn.datasets import fetch_california_housing
fetch_california_housing(as_frame=True) # California housing (CH)
```

The Mushroom (MU) data was fetched using the `openml` package by calling,

```
openml.datasets.get_dataset(24) # Mushroom (MU)
```

The Mushroom data was processed to remove two columns prior to the analysis ("stalk-root" which contained too many missing values, and "veil-type" which assumed the same value across all samples).

The remaining datasets were previously evaluated in Hansen et al. (2023). (The only two datasets from Hansen et al. which were not included in our comparisons were the Default and Heloc datasets which generated errors when evaluated with Synthcity's `Benchmarks` function.) The datasets were fetched using the `openml` package by calling,

```
openml.datasets.get_dataset(44126) # Bank marketing (BM)
openml.datasets.get_dataset(44089) # Credit (CR)
openml.datasets.get_dataset(45022) # Diabetes 130US (DI)
openml.datasets.get_dataset(44120) # Electricity (EL)
openml.datasets.get_dataset(44130) # Eye movements (EM)
openml.datasets.get_dataset(44123) # House 16H (HO)
openml.datasets.get_dataset(44125) # Magic telescope (MT)
openml.datasets.get_dataset(44122) # Pol (PO)
```

### G.1. Additional Comments on Baseline Model Selection

As pointed in the main text, we restricted our comparisons to generators implemented in Synthcity. Some of the generators available in Synthcity were, nonetheless, excluded from our evaluations due to computational constrains. Preliminary checks showed it was unfeasible to include GReaT (Borisov et al., 2023), due to its heavy computational demands (we run our experiments in CPUs). (Also, recent work by Zhang et al. (2024) has shown that GReaT tends to be outperformed by DDPM, which is included in our comparisons.) We also excluded normalizing flows from our comparisons, as preliminary evaluations also showed it was computationally more expensive than the other deep generative models. Finally, while the bayesian network generators does not represent a strong baseline, it was included in our comparisons as an example of a computationally efficient generator.

### G.2. Hyperparameter Optimization for the Baseline Models

For the Abalone (AB), California housing (CH), and Mushroom (MU) datasets, the hyperparameter optimization for the DDPM, ARF, TVAE, and CTGAN baselines was conducted in Synthcity using Optuna (Akiba et al., 2019) (over 20 trials for each model). The tuning process was guided by the minimization of the AUROC of a XGBoost classifier trained to discriminate synthetic and real data. Tables 9, 10 and 11 report the optimized hyperparameter values for these three datasets.

The remaining datasets were evaluated in Hansen et al. (2023) and we adopted the same hyperparameter values employed in their paper for the DDPM, TVAE, and CTGAN models. These hyperparameters are reported in Supplementary Table 3 in Hansen et al. (which also used Optuna guided by AUROC minimization of a XGBoost classifier in their optimization). (The adoption of these published hyperparameter values allowed for considerable computation savings, given that the Optuna optimization was by far the most time consuming step in the experiments.) For the ARF baseline (which was not evaluated by Hansen et al.) we adopted the default values for the ARF plugin in Synthcity, which are close to the values used by Watson et al. 2023 in their experiments. Table 12 report the adopted hyperparameter values for these remaining datasets.

For the BayesNet baseline we adopted hill-climbing for structure search and BIC score for structure learning for all datasets. (Which corresponds to the values adopted by Hansen et al., 2023.)

*Table 9.* Hyperparameters used for the generative models trained on the Abalone (AB) dataset.

| Model | Parameters |
|---|---|
| DDPM | n_iter: 7605
lr: 0.002991978123076162
batch_size: 970
num_timesteps: 407
is_classification: False |
| ARF | num_trees: 80
delta: 0
max_iters: 2
early_stop: False
min_node_size: 2 |
| TVAE | n_iter: 400
lr: 0.001
decoder_n_layers_hidden: 5
weight_decay: 0.0001
batch_size: 128
n_units_embedding: 200
decoder_n_units_hidden: 150
decoder_nonlin: tanh
decoder_dropout: 0.19964446358158816
encoder_n_layers_hidden: 4
encoder_n_units_hidden: 100
encoder_nonlin: relu
encoder_dropout: 0.0820245231222064 |
| CTGAN | n_iter: 700
generator_n_layers_hidden: 1
generator_n_units_hidden: 100
generator_nonlin: elu
generator_dropout: 0.13836424598477665
discriminator_n_layers_hidden: 2
discriminator_n_units_hidden: 100
discriminator_nonlin: tanh
discriminator_n_iter: 5
discriminator_dropout: 0.023861565936528797
lr: 0.001
weight_decay: 0.0001
batch_size: 200
encoder_max_clusters: 8 |

*Table 10.* Hyperparameters used for the generative models trained on the California housing (CH) dataset.

| Model | Parameters |
|---|---|
| DDPM | n_iter: 8300
lr: 0.009824330156648882
batch_size: 3177
num_timesteps: 200
is_classification: False |
| ARF | num_trees: 70
delta: 0
max_iters: 2
early_stop: True
min_node_size: 6 |
| TVAE | n_iter: 200
lr: 0.001
decoder_n_layers_hidden: 4
weight_decay: 0.001
batch_size: 512
n_units_embedding: 150
decoder_n_units_hidden: 300
decoder_nonlin: leaky_relu
decoder_dropout: 0.13648576055463643
encoder_n_layers_hidden: 2
encoder_n_units_hidden: 400
encoder_nonlin: tanh
encoder_dropout: 0.02705334756273372 |
| CTGAN | n_iter: 600
generator_n_layers_hidden: 1
generator_n_units_hidden: 150
generator_nonlin: relu
generator_dropout: 0.16863490048383495
discriminator_n_layers_hidden: 3
discriminator_n_units_hidden: 150
discriminator_nonlin: relu
discriminator_n_iter: 4
discriminator_dropout: 0.06303278452420555
lr: 0.0002
weight_decay: 0.001
batch_size: 200
encoder_max_clusters: 5 |

*Table 11.* Hyperparameters used for the generative models trained on the Mushroom (MU) dataset.

| Model | Parameters |
|---|---|
| DDPM | n_iter: 5127 
 lr: 0.00884671824119367 
 batch_size: 4093 
 num_timesteps: 853 
 is_classification: True |
| ARF | num_trees: 100 
 delta: 0 
 max_iters: 1 
 early_stop: False 
 min_node_size: 2 |
| TVAE | n_iter: 200 
 lr: 0.001 
 decoder_n_layers_hidden: 5 
 weight_decay: 0.001 
 batch_size: 512 
 n_units_embedding: 150 
 decoder_n_units_hidden: 150 
 decoder_nonlin: relu 
 decoder_dropout: 0.1171471896118231 
 encoder_n_layers_hidden: 4 
 encoder_n_units_hidden: 300 
 encoder_nonlin: tanh 
 encoder_dropout: 0.16007215982462047 |
| CTGAN | n_iter: 700 
 generator_n_layers_hidden: 4 
 generator_n_units_hidden: 50 
 generator_nonlin: leaky_relu 
 generator_dropout: 0.020973543252274986 
 discriminator_n_layers_hidden: 3 
 discriminator_n_units_hidden: 150 
 discriminator_nonlin: tanh 
 discriminator_n_iter: 4 
 discriminator_dropout: 0.1644064126493125 
 lr: 0.001 
 weight_decay: 0.001 
 batch_size: 500 
 encoder_max_clusters: 13 |

*Table 12.* Hyperparameters used for the generative models trained on the remaining datasets, namely, Adult (AD), Bank marketing (BM), Credit (CR), Diabetes 130US (DI), Electricity (EL), Eye movements (EM), House 16H (HO), Magic telescope (MT), Mushroom (MU), and Pol (PO).

| Model | Parameters |
|---|---|
| DDPM | n_iter: 1051 
 lr: 0.0009375080542687667 
 batch_size: 2929 
 num_timesteps: 998 
 is_classification: True |
| ARF | num_trees: 30 
 delta: 0 
 max_iters: 10 
 early_stop: True 
 min_node_size: 5 |
| TVAE | n_iter: 300 
 lr: 0.0002 
 decoder_n_layers_hidden: 4 
 weight_decay: 0.001 
 batch_size: 256 
 n_units_embedding: 200 
 decoder_n_units_hidden: 300 
 decoder_nonlin: elu 
 decoder_dropout: 0.194325119117226 
 encoder_n_layers_hidden: 1 
 encoder_n_units_hidden: 450 
 encoder_nonlin: leaky_relu 
 encoder_dropout: 0.04288563703094718 |
| CTGAN | n_iter: 1000 
 generator_n_layers_hidden: 2 
 generator_n_units_hidden: 50 
 generator_nonlin: tanh 
 generator_dropout: 0.0575 
 discriminator_n_layers_hidden: 4 
 discriminator_n_units_hidden: 150 
 discriminator_nonlin: relu 
 discriminator_n_iter: 1 
 discriminator_dropout: 0.1 
 lr: 0.001 
 weight_decay: 0.001 
 batch_size: 200 
 encoder_max_clusters: 10 |

### G.3. The DBLR and SDID Metrics and their Sorted Versions, SDBLR and SSDID

In our evaluations of experiment set 2 we adopted sorted versions of the following standard disclosure risk metrics in the statistical disclosure control field:

1. The Distance-Based Record Linkage (DBRL) metric (Pagliuca and Seri, 1999; Domingo-Ferrer and Torra, 2001) is a widely used approach for assessing re-identification disclosure risk in Statistical Disclosure Control (SDC). The method calculates the Euclidean distance between each record in the masked dataset and all records in the original dataset. A masked record is considered "linked" if its closest match in the original dataset is its true, corresponding record. The DBRL metric is then defined as the proportion of masked records that are correctly linked to their original counterparts.

2. The standard deviation interval distance (SDID) metric (MateoSanz, Sebe, and Domingo-Ferrer, 2004) is a popular metric for measuring attribute disclosure risk. It corresponds to the proportion of original records inside a standard deviation interval whose center is the corresponding masked record (where the interval width is computed in terms of a percentage $p$ of the standard deviation of the variable). A record in the original dataset is considered to be inside the standard deviation interval of masked record $i$ if, for all variables $j$, it is inside the respective stand. dev. interval.

Strictly speaking, these two metrics are only indicated to evaluate masking methods for which there exists a mapping between the original and masked values (such as when the masked data is generated by adding noise to the original data). For synthetic data (for which such mapping is unavailable) it is still, nonetheless, possible to find an approximate mapping using an analogous strategy as the ones adopted by Domingo-Ferrer et al. (2020) and Chaibub Neto (2024). Basically, an approximate mapping between the original and synthetic datasets is obtained by sorting the rows of both the original and synthetic datasets according to the values of a given attribute (column) of the data, and then perform the metric computation on the sorted datasets. (A justification for this procedure is provided in section 3 of Domingo-Ferrer et al., 2020.) In our experiments we sorted the data according to the variable with largest number of unique values. These sorted versions of the DBRL and SDID metrics are denoted here as the SDBRL and SSDID metrics.

### G.4. Illustration of the Influence of the $n_c$ Parameter on the Quality of the Synthetic Data Generated by TabSDS

Figure 11 shows scatterplots of the Latitude versus Longitude variables in the California housing (CH) dataset (alongside the boundaries if the California state in black). In each panel, a random sample of 1000 points is shown in blue for the real data and in red for the synthetic data generated by TabSDS. The blue and red lines represent the least squares fits to the real and synthetic data, respectively.

Panel a shows that for $n_c = 1$, TabSDS is unable to preserve any of the association observed in the real data. (The horizontal red line indicates that the Latitude and Longitude variables are essentially uncorrelated.) However, as $n_c$ increases, TabSDS generates increasingly more realistic data, and already generates fairly realistic data for $n_c = 30$.

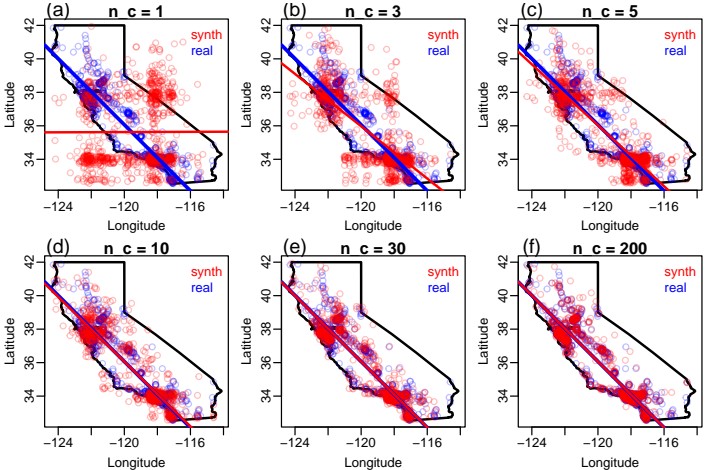

*Figure 11.* Influence of the $n_c$ parameter on the quality of the synthetic data generated by TabSDS.

### G.5. Tuning Parameter Selection for TabSDS

As described in the main text, the $n_c$ tuning parameter from the TabSDS approach was selected from a grid of 24 values. The adopted selection criterion was the following. For datasets containing mostly categorical variables (namely, Adult and Mushroom), we selected the largest $n_c$ value which balanced well the tradeoff between detection test, ML efficiency, and domias MIA. For the remaining datasets (which contain mostly numerical variables) we compared the DCR distributions from the synthetic data generated with increasing $n_c$ values against the DCR distribution of the test-set, and selected the largest $n_c$ value which generates a DCR distribution with a median value close to the median of the test set DCR distribution.

Figures 12 to 23 report the following:

- Panel a shows the detection test metric (based on a XGBoost classifier).

- Panel b shows the ML efficiency metric (based on a XGBoost learner). The horizontal grey line reports the ground truth value (i.e., the predicted performance on the (real) test set from a learner trained on the (real) training set data).

- Panel c reports the domias MIA metric.

- Panel d shows the DCR distributions between training and synthetic samples (white boxplots) and between training and test sets (grey boxplots).

Detection test, ML efficiency, and the domias MIA metrics were computed using Synthecity's `Benchmarks` function. For the domias MIA metric we only used the "KDE" and "prior" density estimation methods as the BNFA was computationally too expensive. For most datasets, the `Benchmarks` function only outputted the domias MIA metrics based on the prior knowledge method (and for the House 16H and Eye movement datasets, it failed to generate a MIA estimate). Additionally, for some datasets it generated MIA scores consistently lower than 0.5. Use of the considerably more expensive BNFA estimator in some preliminary tests did not resolve these issues. These observations suggest that computation of the domias MIA metric can be challenging in some datasets, and results should be interpreted with caution. In all Figures panels a, b, and c report the average values across 10 replications.

Except for the Adult and Mushroom datasets (Figures 13 and 22), the $n_c$ values were selected by inspecting panel d of the plots. In all plots the red dots highlight the metric values for the selected $n_c$ parameter, and Table 13 summarizes the results.

For the Adult dataset we selected $n_c = 20$, which produced a detection test lower than 0.7, ML efficiencies above 0.88, and domias MIA already above 0.76. For the Mushroom dataset we select $n_c = 40$, which produced a detection test lower than 0.7, ML efficiencies above 0.99, and domias MIA around 0.64.

*Table 13.* Selected $n_c$ values.

| DATASET | AB | AD | BM | CH | CR | DI | EL | EM | HO | MT | MU | PO |
|---|---|---|---|---|---|---|---|---|---|---|---|---|
| SELECTED $n_c$ | 20 | 20 | 100 | 200 | 1000 | 35 | 20 | 20 | 1000 | 25 | 40 | 15 |

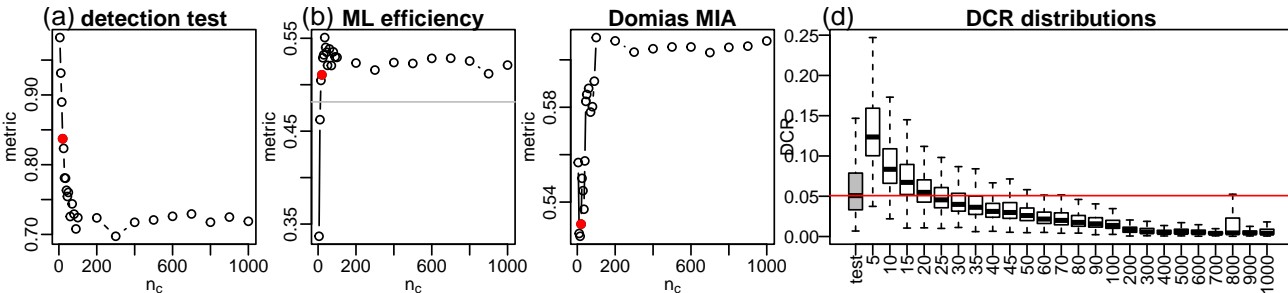

*Figure 12.* Tuning parameter selection for the Abalone (AB) dataset. Selected $n_c = 20$.

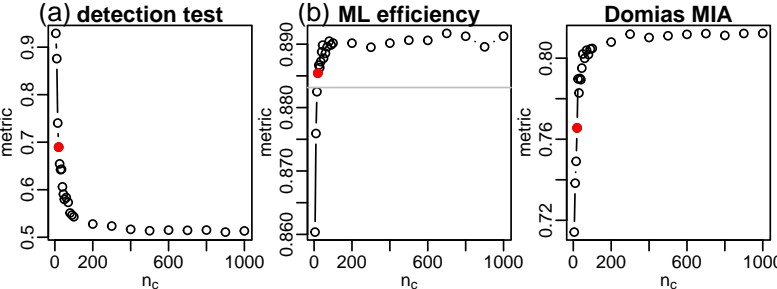

*Figure 13.* Tuning parameter selection for the Adult (AD) dataset. Selected $n_c = 20$.

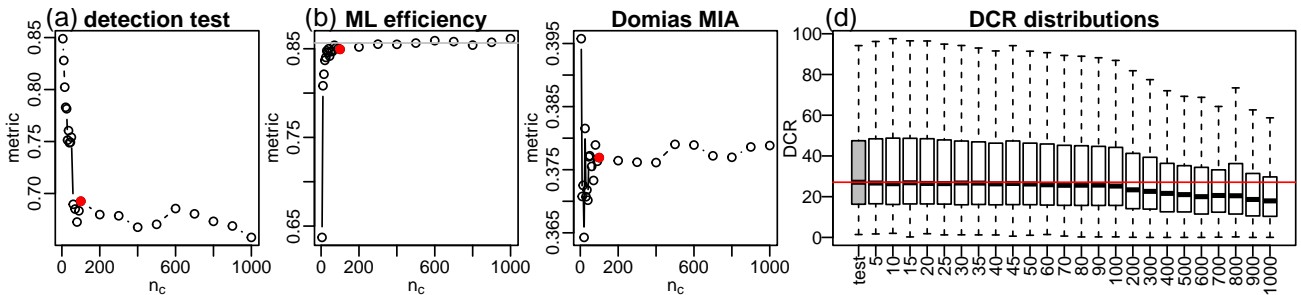

*Figure 14.* Tuning parameter selection for the Bank marketing (BM) dataset. Selected $n_c = 100$.

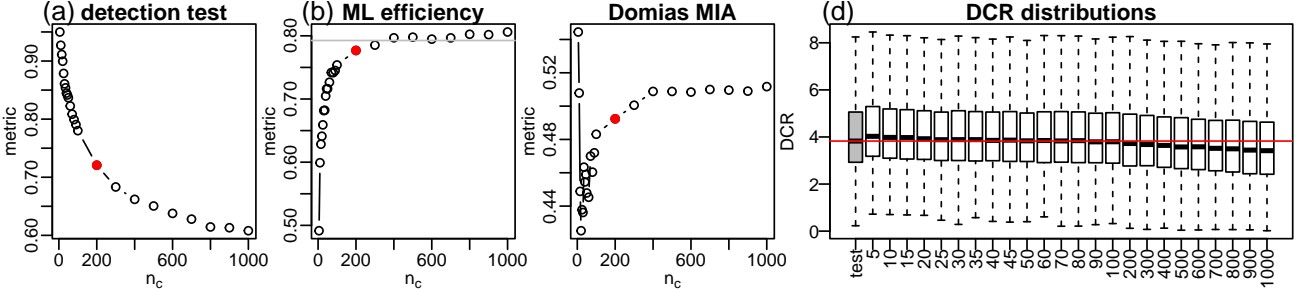

*Figure 15.* Tuning parameter selection for the California housing (CH) dataset. Selected $n_c = 200$.

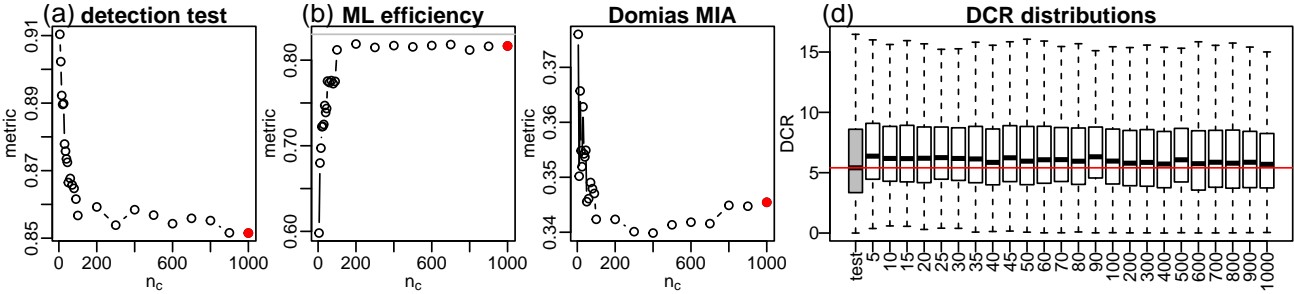

*Figure 16.* Tuning parameter selection for the Credit (CR) dataset. Selected $n_c = 1000$.

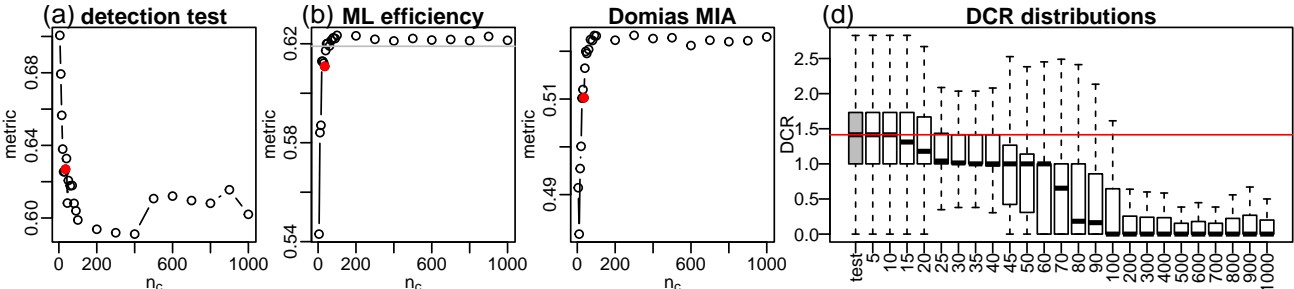

*Figure 17.* Tuning parameter selection for the Diabetes 130US (DI) dataset. Selected $n_c = 35$.

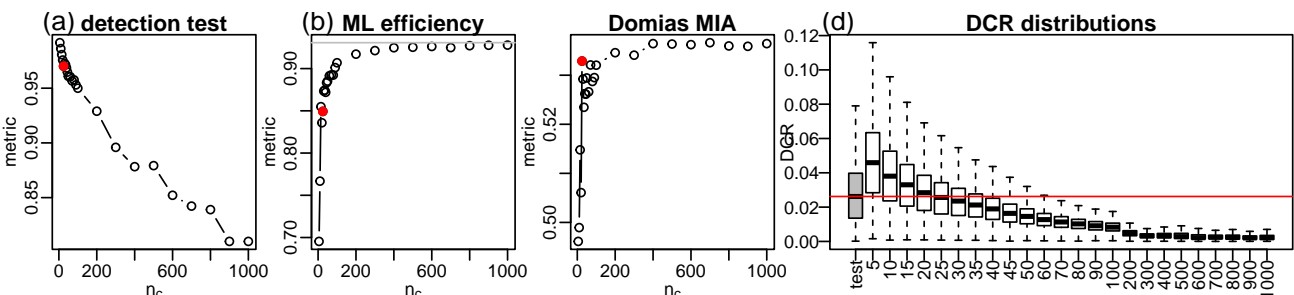

*Figure 18.* Tuning parameter selection for the Electricity (EL) dataset. Selected $n_c = 20$.

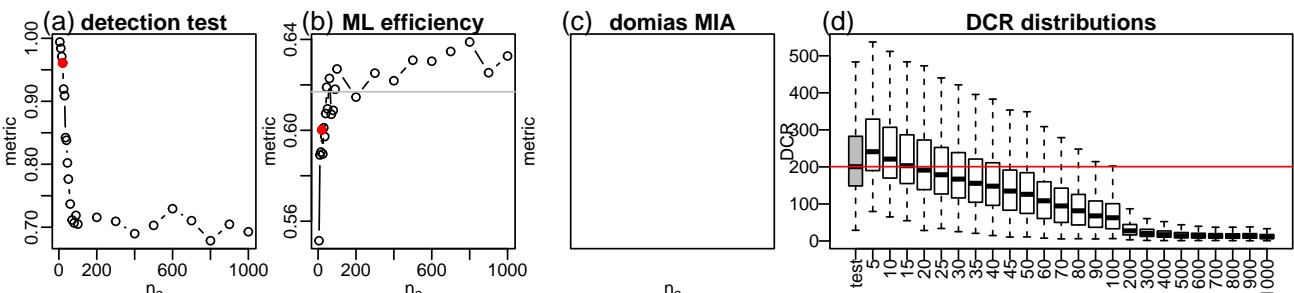

*Figure 19.* Tuning parameter selection for the Eye movements (EM) dataset. Selected $n_c = 20$.

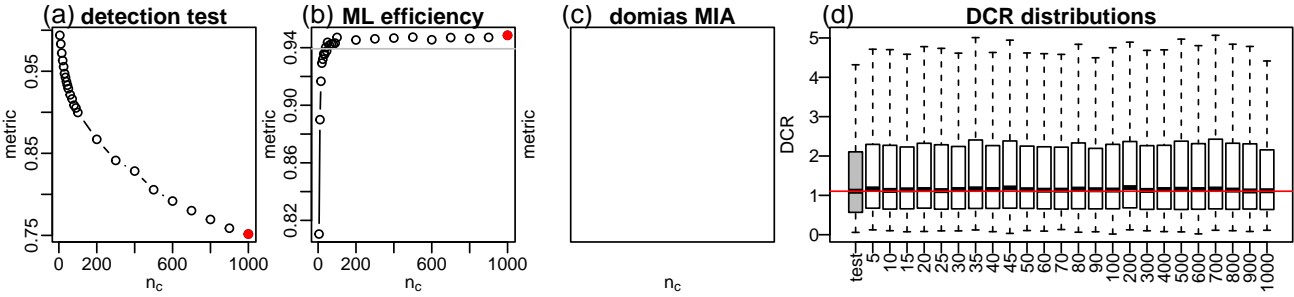

*Figure 20.* Tuning parameter selection for the House 16H (HO) dataset. Selected $n_c = 1000$.

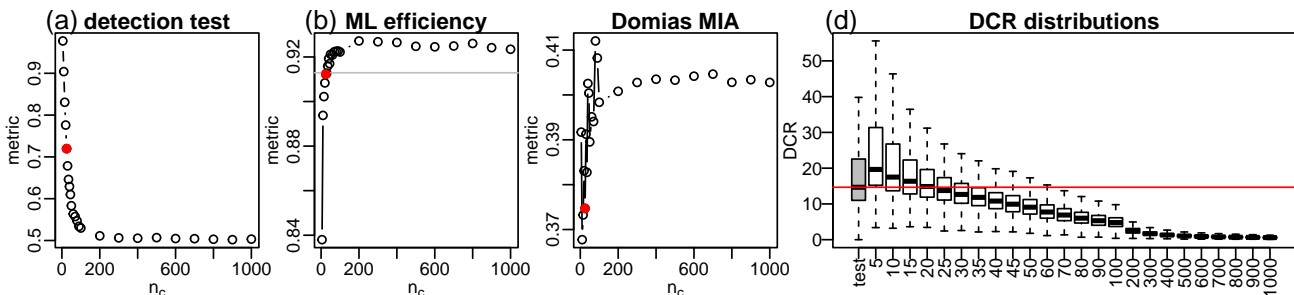

*Figure 21.* Tuning parameter selection for the Magic telescope (MT) dataset. Selected $n_c = 25$.

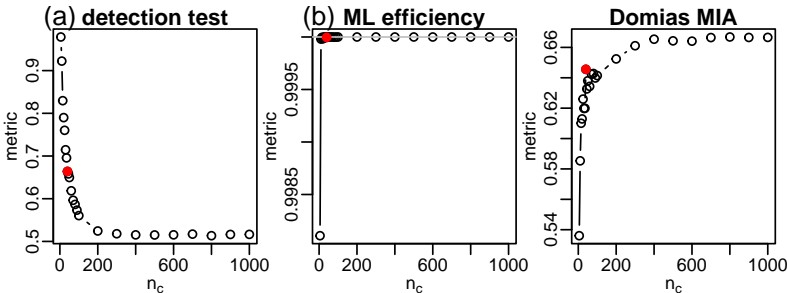

*Figure 22.* Tuning parameter selection for the Mushroom (MU) dataset. Selected $n_c = 40$.

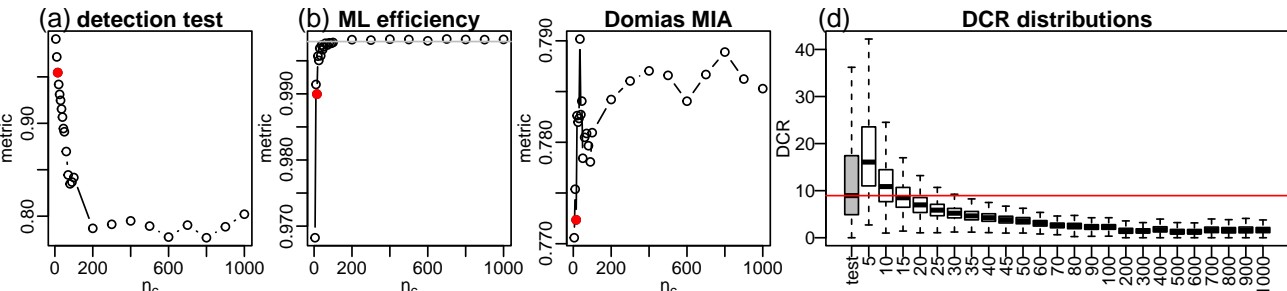

*Figure 23.* Tuning parameter selection for the Pol (PO) dataset. Selected $n_c = 15$.

**G.6. Extended Results for Experiment Set 2**

Here, we present extended results for experiment set 2. In addition to reporting the comparisons for the 10 datasets (which contain mostly numerical variables), these extended results also present comparisons against new baselines including: (i) SMOTE (Chawla et al., 2002), based on 5 and 20 nearest neighbors; (ii) two differential privacy (DP) methods, ADSGAN (Yoon et al., 2020) and PATEGAN (Jordon et al., 2019); and (iii) a simple additive noise perturbation method (Brand, 2004).

Noise addition was evaluated over a grid of 13 increasing levels of additive noise ranging from 1% to 50% of the standard deviation of each variable. The results for the DP-based models were based on default hyperparameter choices (and should, therefore, be taken with a grain of salt). SMOTE was evaluated using two number of neighbors tuning parameter values, $k = 5$ and $k = 20$ (and implemented for numerical variables alone).

Figures 24 and 25 report the comparisons between TabSDS and all the remaining baselines, except for the additive noise perturbation method (whose comparisons are presented in Figures 26 and 27). All figures report the tradeoff between data privacy (measured by DCR, SDBRL, and SSDID) and data fidelity (measured by the detection test AUC score).

The left panels (a, d, g, j, and m) on Figures 24 and 25 show DCR vs AUC plots. The red line represents the DCR score comparing the training and test sets and provides an estimate of the DCR value we would expect to see for an ideal generator able to draw i.i.d. data from the same distribution as the training data. The red dot represents the selected value of $n_c$ based on the DCR criterium (i.e., the DCR value closest to the test set DCR). The middle (b, e, h, k, and n) and right (c, f, i, l, and o) panels show the tradeoff plots comparing SDBRL vs AUC and SSDID vs AUC, respectively.

Overall, DDPM, TabSDS, and SMOTE tended to outperform the other methods in terms of fidelity (measured by the detection test AUC). These 3 methods tended to show somewhat balanced performances with none of the methods consistently outperforming the others. However, in terms of privacy, SMOTE tended to be considerably worse than DDPM and TabSDS with respect to DCR. It also tended to be worse than DDPM (and of TabSDS to a lesser extent) in terms of DBRL and SDID.

ADSGAN, PATEGAN, TVAE, CTGAN, and BayesNet tended to trade high data privacy by low data fidelity. In all datasets, these methods showed AUCs close to 1, low SDBRL and SSDID, and high DCR. (These high DCR values are likely a consequence that these models fail to approximate well the distribution of the training data.) ARF tended to do slightly better than these models in a few datasets.

Figures 26 and 27 present the tradeoff curves comparing TabSDS (black) vs additive noise (blue). The additive noise approach showed high AUC values across all noise levels across most datasets and is not competitive against TabSDS.

In order to evaluate the impact of generating new values for numerical variables to improve privacy protection, we also compare TabSDS against a simplified version of the algorithm (denoted TabSJPPDS), which discards the new value generation step. Figures 28 and 29 compare TabSDS against TabSJPPDS, and illustrate the additional privacy protection achieved by generating new values. (Note the higher DCR and lower SSDID scores. The differences were less clear cut for SDBRL).

The results described above (and in the main text) were based on DCR values computed on the original data scales. Categorical variables were one-hot-encoded prior to DCR computation. (Note this should not cause issues given that the datasets contained only a few binary categorical variables. Datasets containing only or mostly categorical variables such as Mushroom and Adult datasets were not evaluated given that noise addition can only be applied to numeric data.)

One potential caveat when variables have different scales is that distance-based metrics (such as DCR) might be dominated by the wider range variables. To evaluate this potential issue, we also performed comparisons based on DCR values computed on scaled data (Figures 30 and 31). These results show the same qualitative conclusions as before. The main quantitative difference was that the selected $n_c$ values tended to be lower, leading to a decrease in fidelity and increase in privacy of TabSDS relative to the previous results.

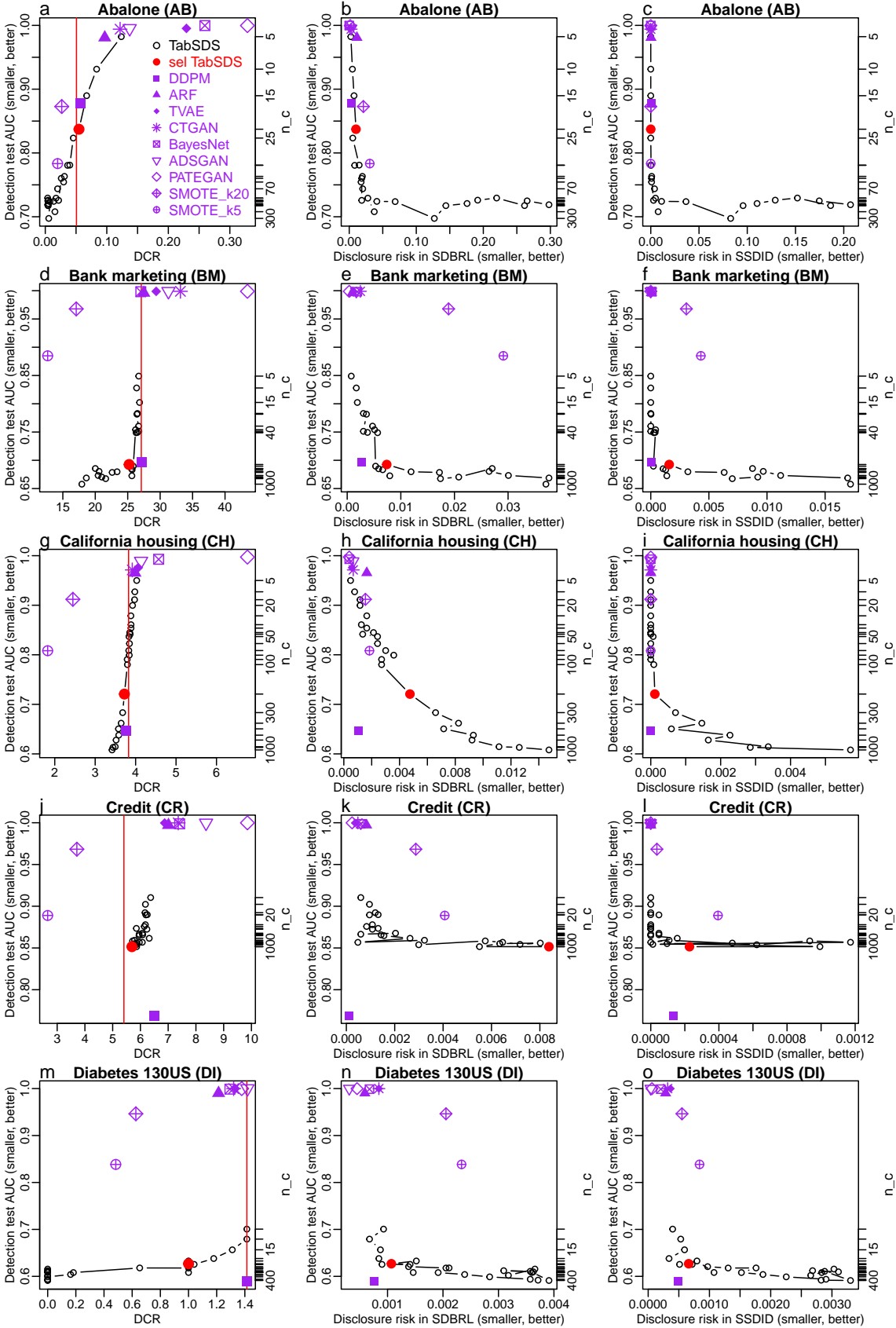

*Figure 24.* Privacy/utility tradeoff comparisons. See legend on panel a.

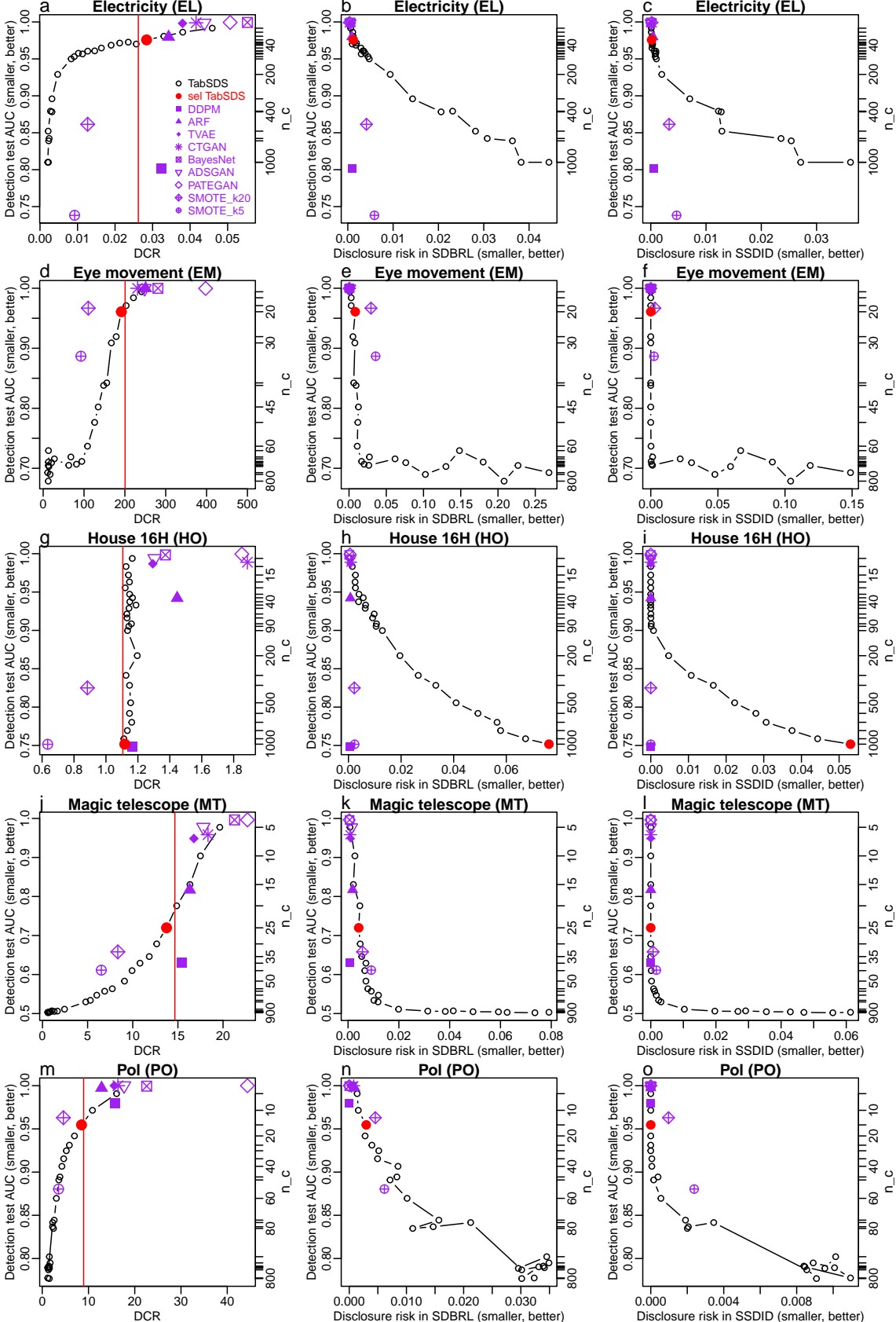

*Figure 25.* Privacy/utility tradeoff comparisons. See legend on panel a.

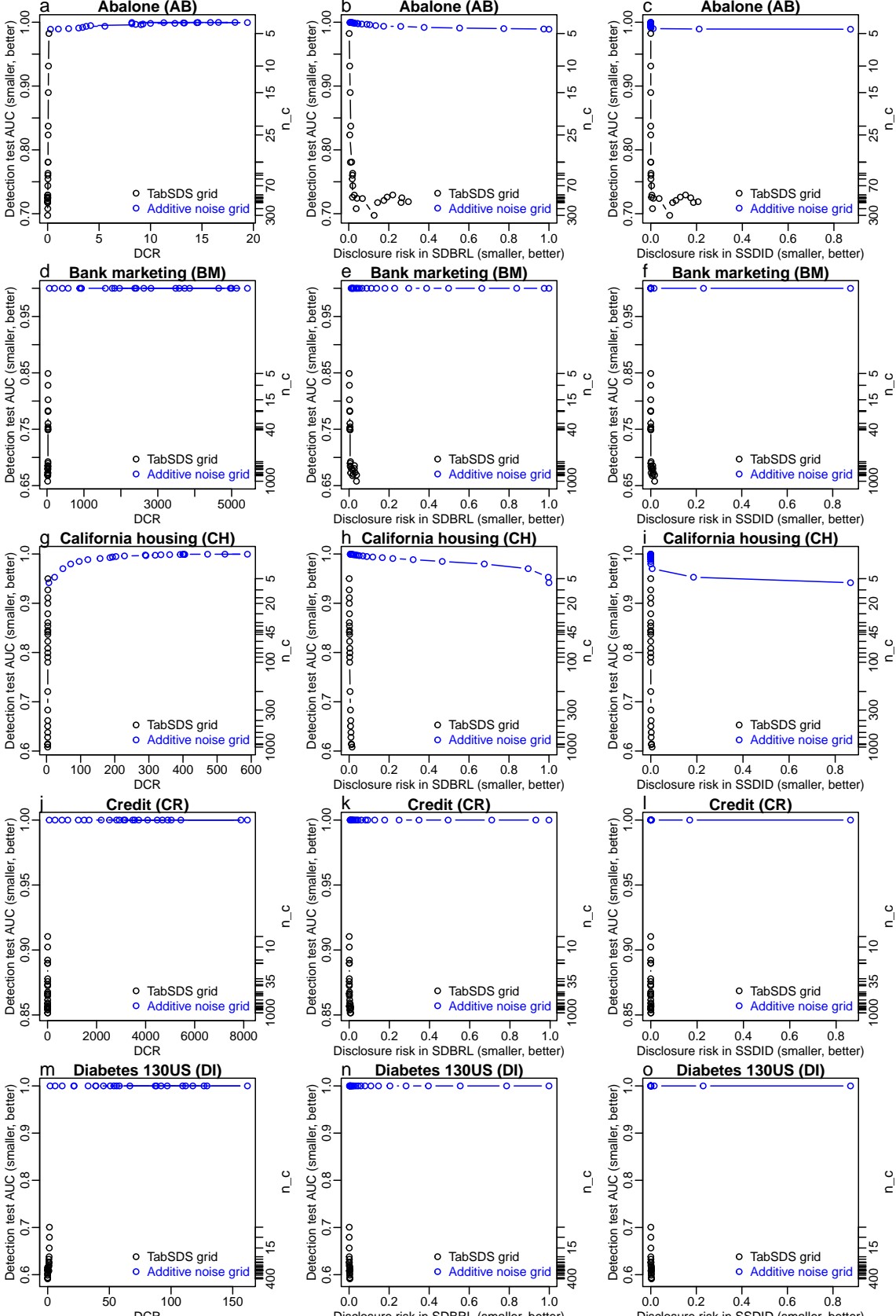

*Figure 26.* Privacy/utility tradeoff comparisons of TabSDS vs additive noise.

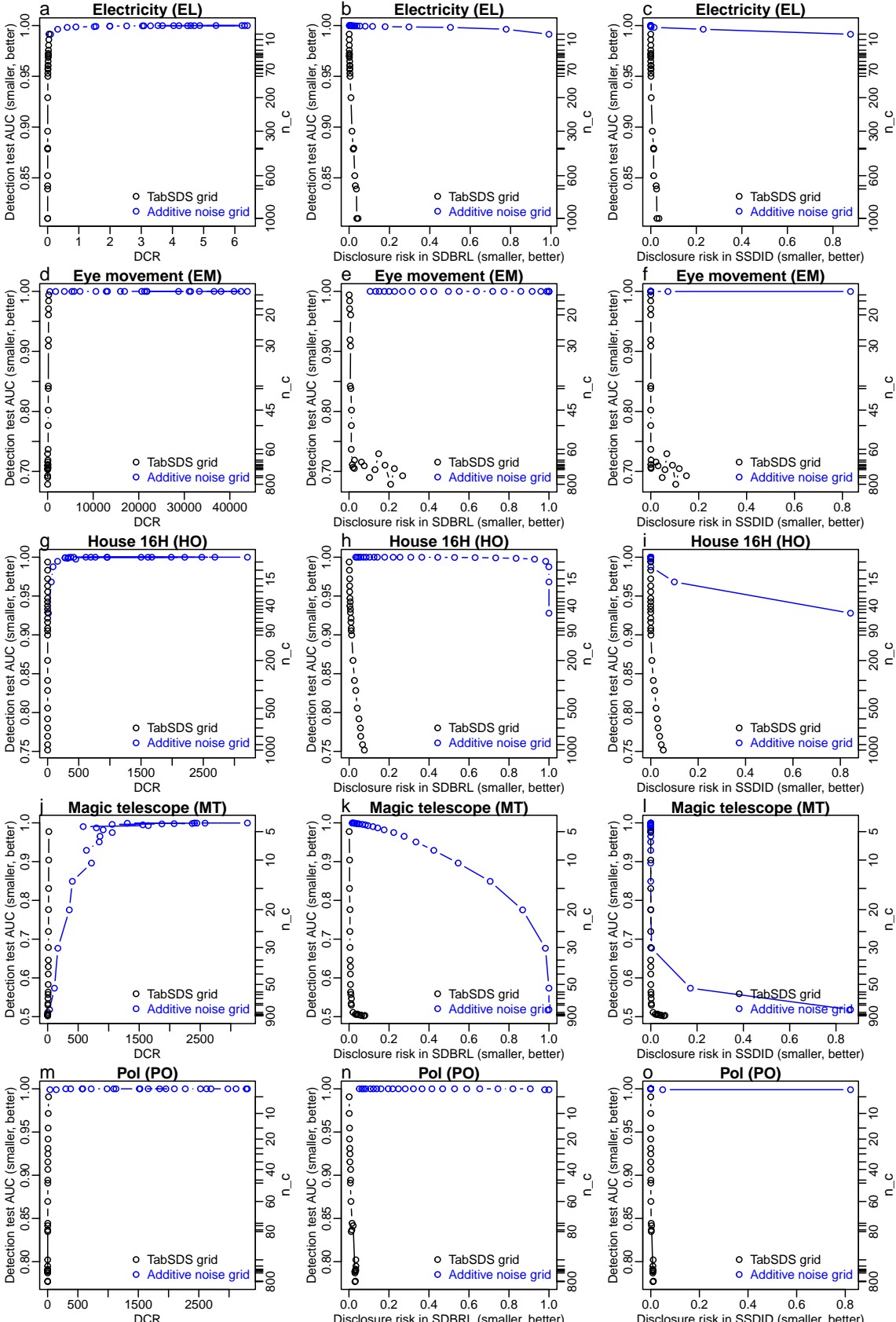

*Figure 27.* Privacy/utility tradeoff comparisons of TabSDS vs additive noise.

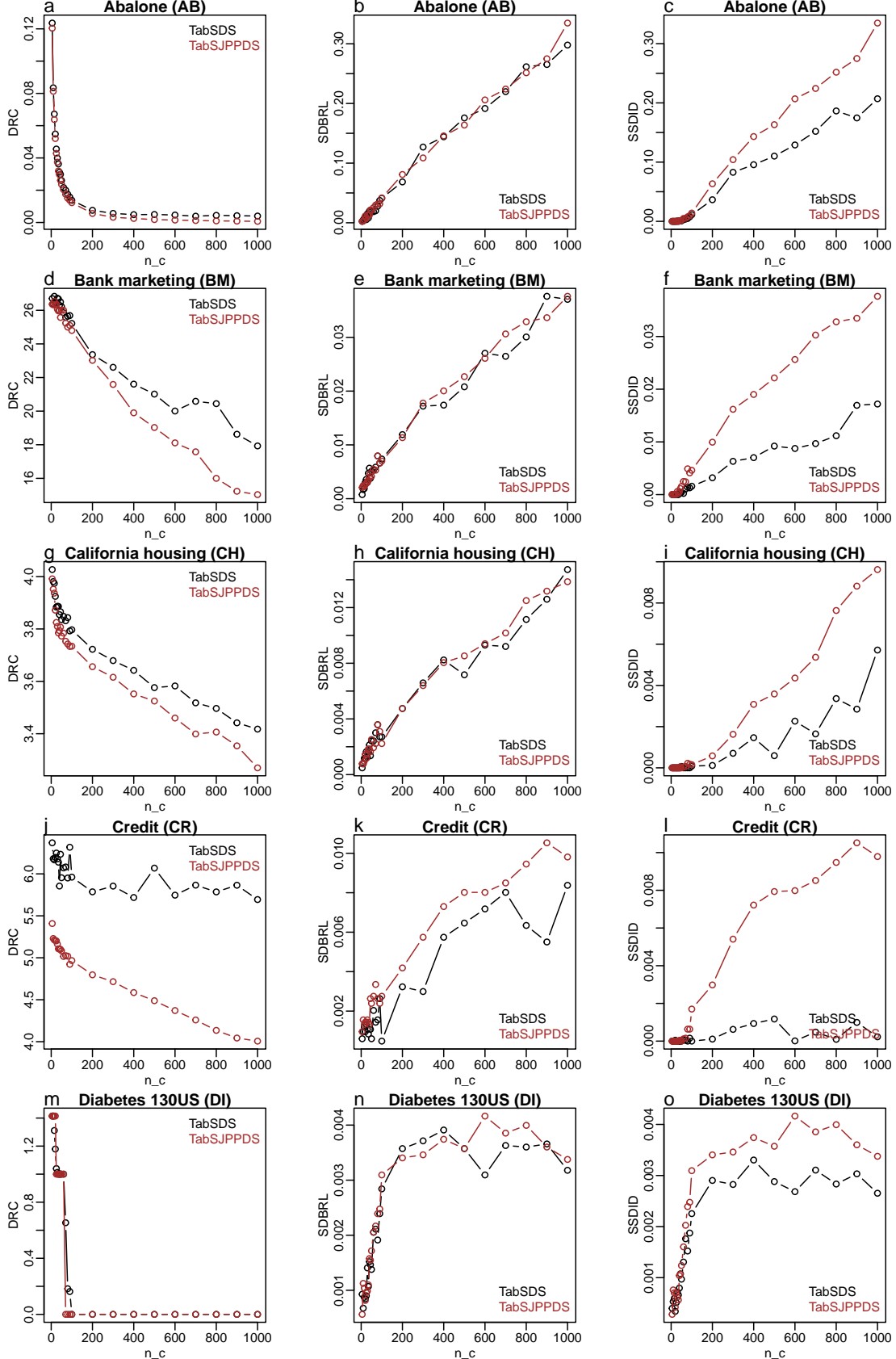

*Figure 28.* Privacy comparisons of TabSDS vs TabSJPPDS.

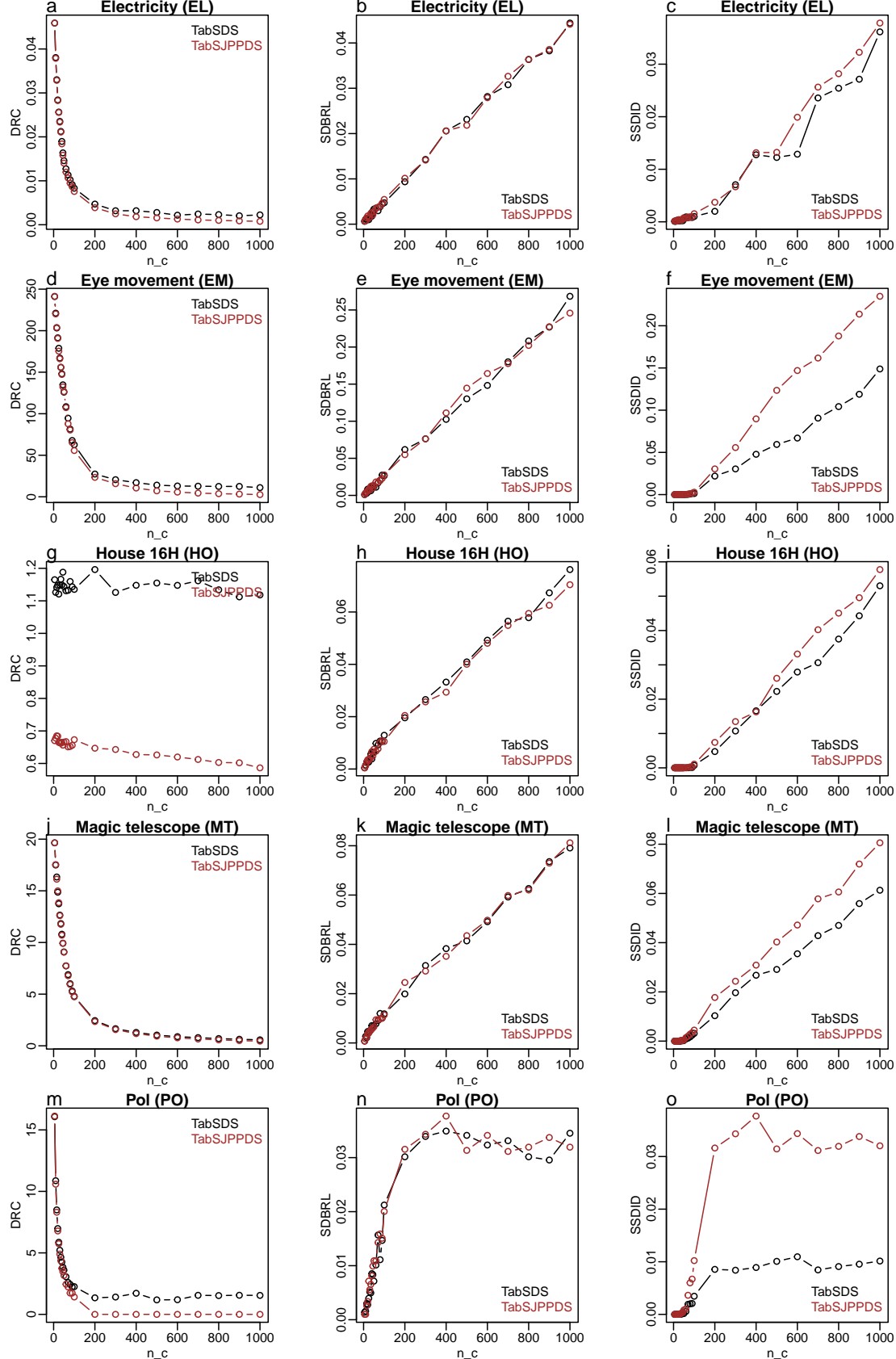

*Figure 29.* Privacy comparisons of TabSDS vs TabSJPPDS.

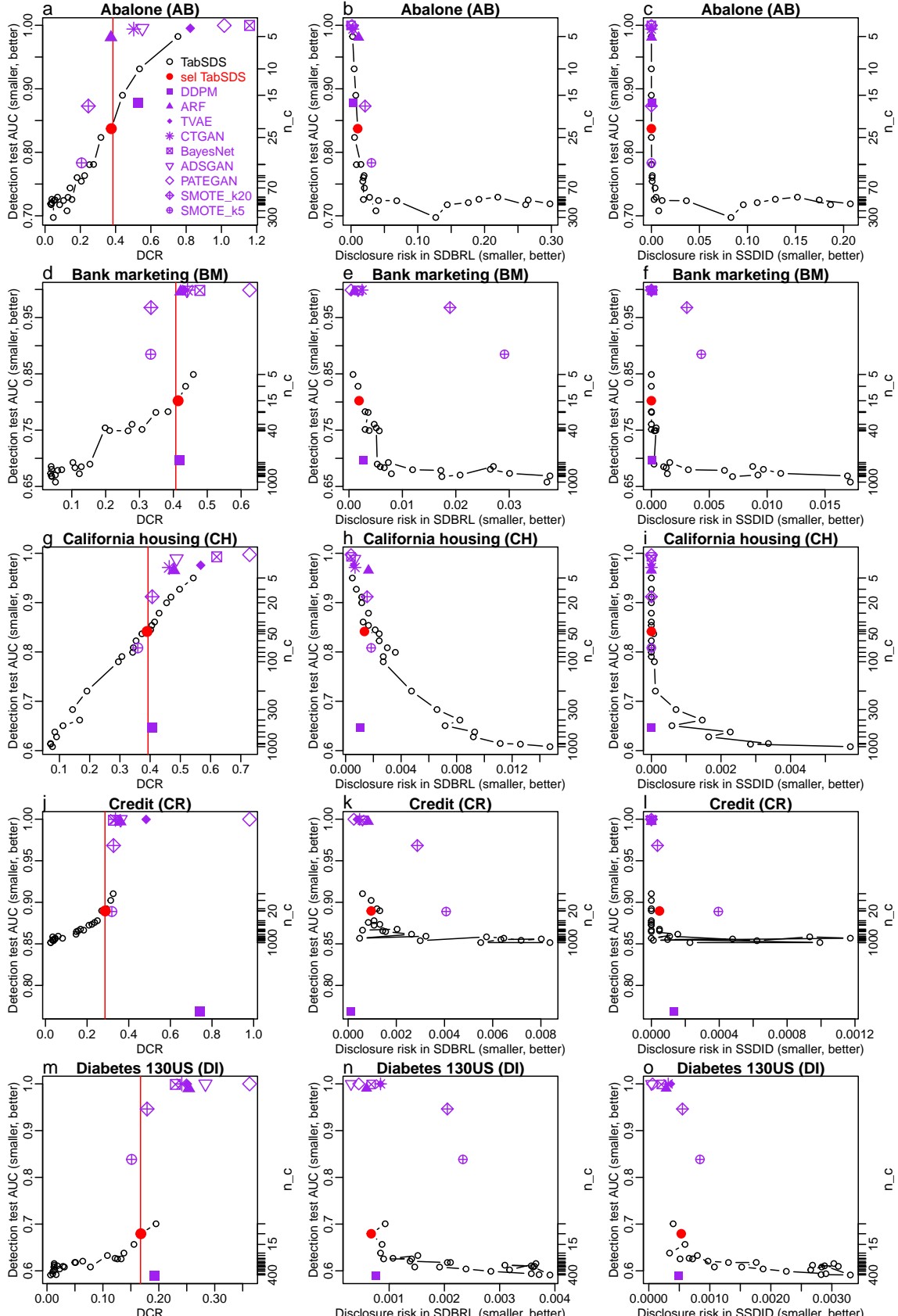

*Figure 30.* Privacy/utility tradeoff comparisons (with scaled DCR). See legend on panel a.

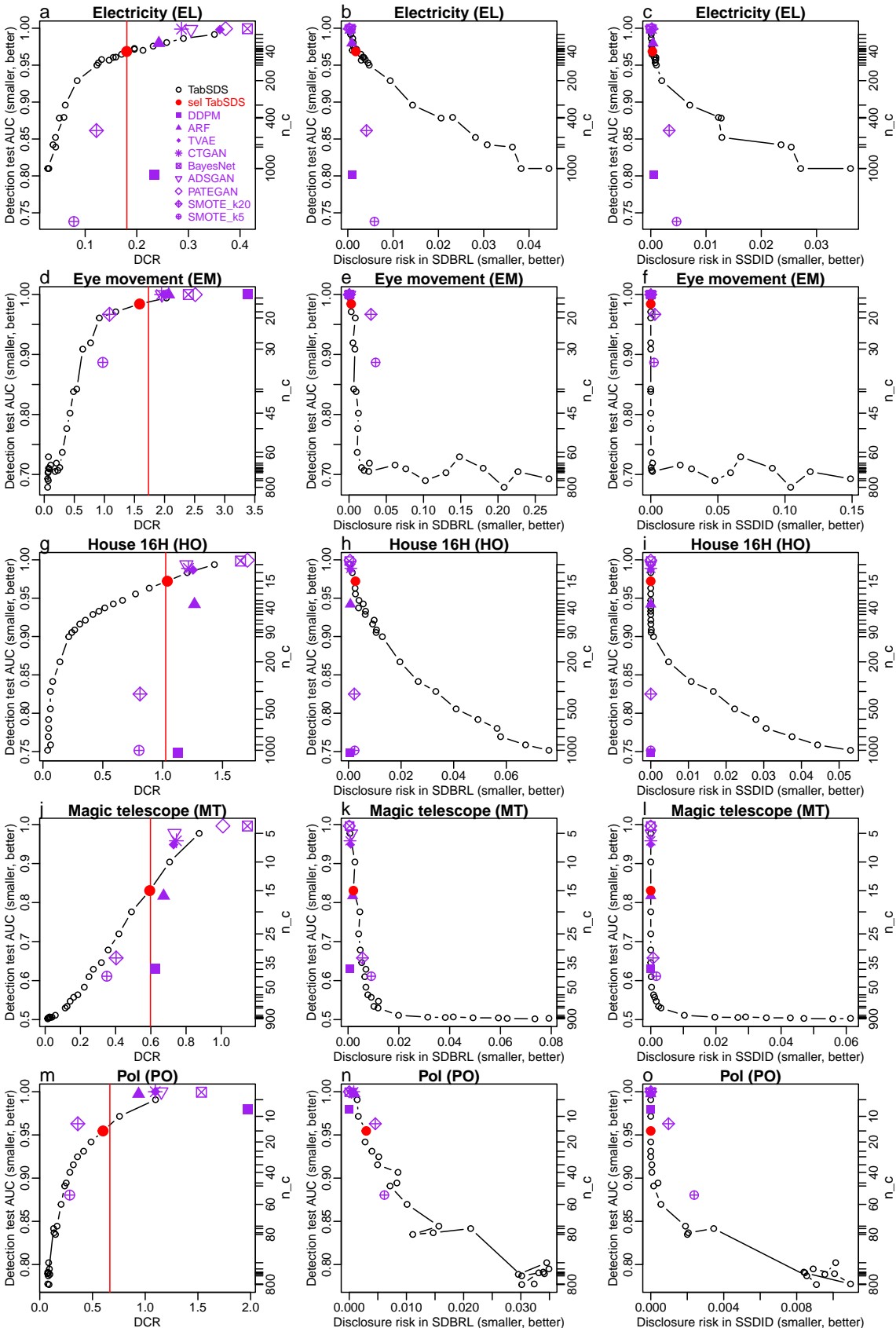

*Figure 31.* Privacy/utility tradeoff comparisons (with scaled DCR). See legend on panel a.

### G.7. Runtime Experiment Details

The runtime experiments report the average combined runtime for model training and data generation computed using the `timeit` Python module. (Results are reported in seconds and the averages were based on 5 replications.) For each dataset, models were benchmarked using the respective optimized hyperparameter values. Experiments were conducted on a C5.2xlarge compute optimized AWS instance equiped with 8 (Intel Xeon Scalable processors or AMD EPYC processors). Hyperparameter optimization takes considerably more time and is not reported in these experiments.

### G.8. Qualitative Comparisons

For the qualitative comparisons we investigate how well the synthetic data is able to recover the pairwise statistical associations observed in the real data, as well as, how well the synthetic data mimic the marginal distributions of the real data. This is done for generators in the experiment sets 1 and 2.

For each of the 12 datasets in experiment set 1, we produce synthetic data using the TabSDS, DDPM, ARF, TVAE, CTGAN, and BayesNet generators (based on their selected tuning parameter values) and:

1. Compute the pairwise association matrices for each of the generated synthetic datasets, as well as, for the real data. Given that these datasets contain both numeric and categorical variables, we compute the pairwise associations between numerical variables using Pearson correlation and between categorical variables using the Cramer-V statistic. For pairs of numeric and categorical variables we regress the numeric variable on the categorical one and use the square root of the $R^2$ statistic as our measure of association. (Note that this measure reduces to the absolute value of the correlation coefficient if we replace the categorical variable for a numeric one.)

2. For each synthetic dataset we compute the absolute value of the differences between the synthetic and real data association matrices. These absolute differences are visualized in Figure 32, where columns represent the generators and rows the datasets. In this plot, darker colors represent larger discrepancies between real and synthetic data. Additionally, we computed the L2 distance between the association matrices (l2d), which is also reported in Figure 32.

3. For each dataset, we visualize the marginal distributions of the first 7 variables in the data. These marginal distributions are reported in Figures 34 to 45.

For experiment set 2, we perform similar comparisons on the 10 datasets comparing TabSDS against the SMOTE (with $k = 5$ and $k = 20$ neighbors), ADSGAN, and the PATEGAN generators. The absolute differences between association matrices comparisons are presented in Figure 33. The marginal distribution comparisons are reported in Figures 46 to 55.

Overall, the results show that the TabSDS approach tends to generate more realistic data than baseline generators. In terms of pairwise associations, Figures 32 and 33 show that it tended to produce association matrices which where closer to the real data association matrix. (Note the overall lighter color and lower l2d scores for TabSDS in comparison to the other generators. Note as well that the SMOTE baseline (Figure 33) also tended to recover well the association structure of the data.) In terms of marginal distributions, inspection of Figures 34 to 55 also show that TabSDS tended to generate more realistic marginal distributions than the other baselines.

Finally, we point out that while the inspection of pairwise associations is useful, it is certainly not ideal as, sometimes, we can see similar correlations for fairly unrealistic data. As an illustration, consider Figure 56 which shows scatterplots of the Latitude versus Longitude data in the California housing (CH) dataset (alongside the boundaries if the CA state in black). In each panel, a random sample of 1000 points is shown in blue for the real data and in red for the synthetic data. The blue and red lines represent the least squares fits to the real and synthetic data, respectively. Note that the blue and red lines are not too far from each other for ARF, TVAE, and CTGAN generators, which indicates that the correlations computed on the synthetic data are not too far from the correlation observed in the real data. However, the scatterplots show that these generators are still producing fairly unrealistic data (i.e., outside the boundaries of the CA state). Only TabSDS and DDPM produce realistic looking data in this example.

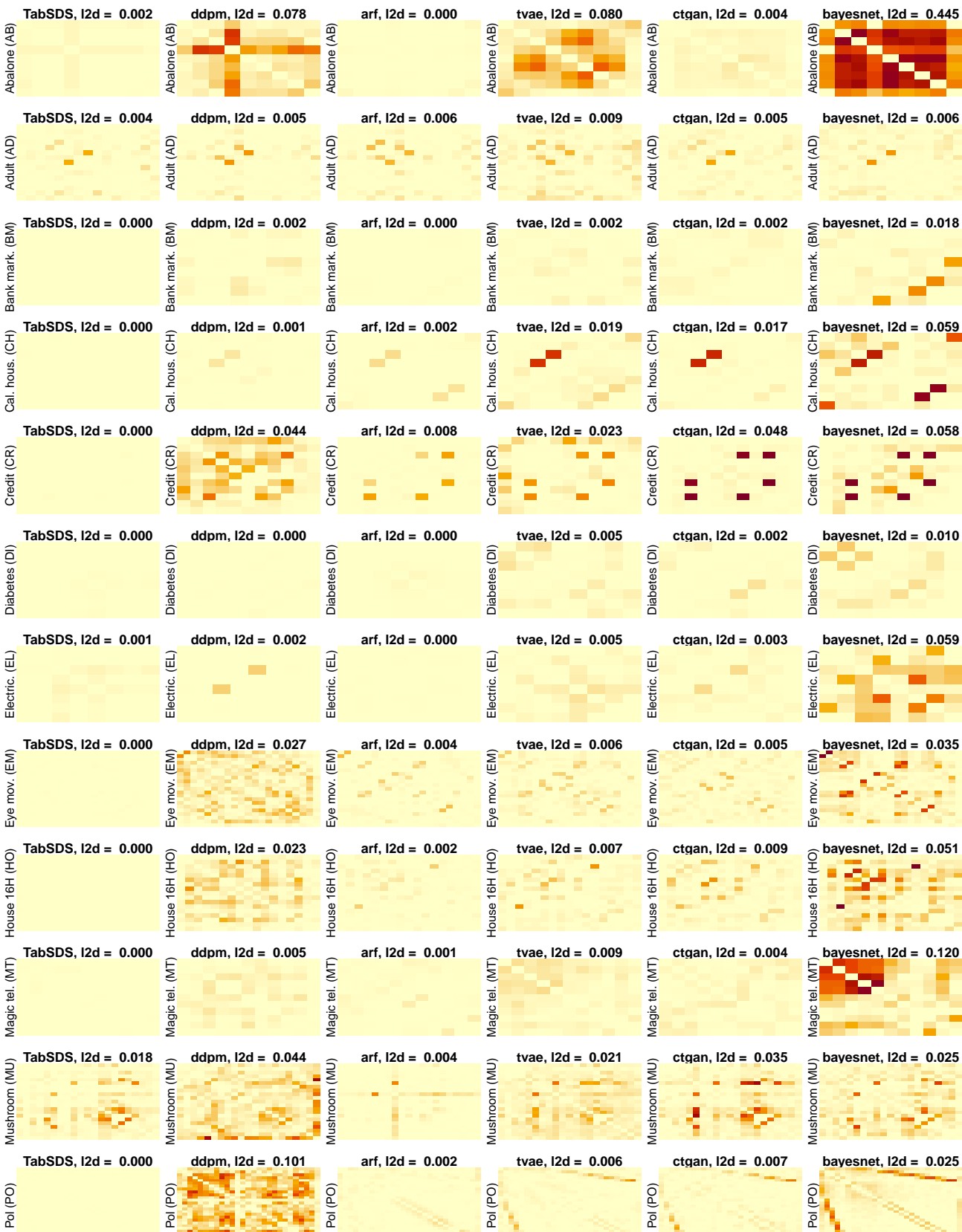

*Figure 32.* Absolute differences between association matrices for experiment set 1. Darker values indicate stronger discrepancy between the real and synthetic datasets. l2d indicates the L2 norm between real and synthetic association matrices. (l2d = 0 indicate a value lower than 0.001.)

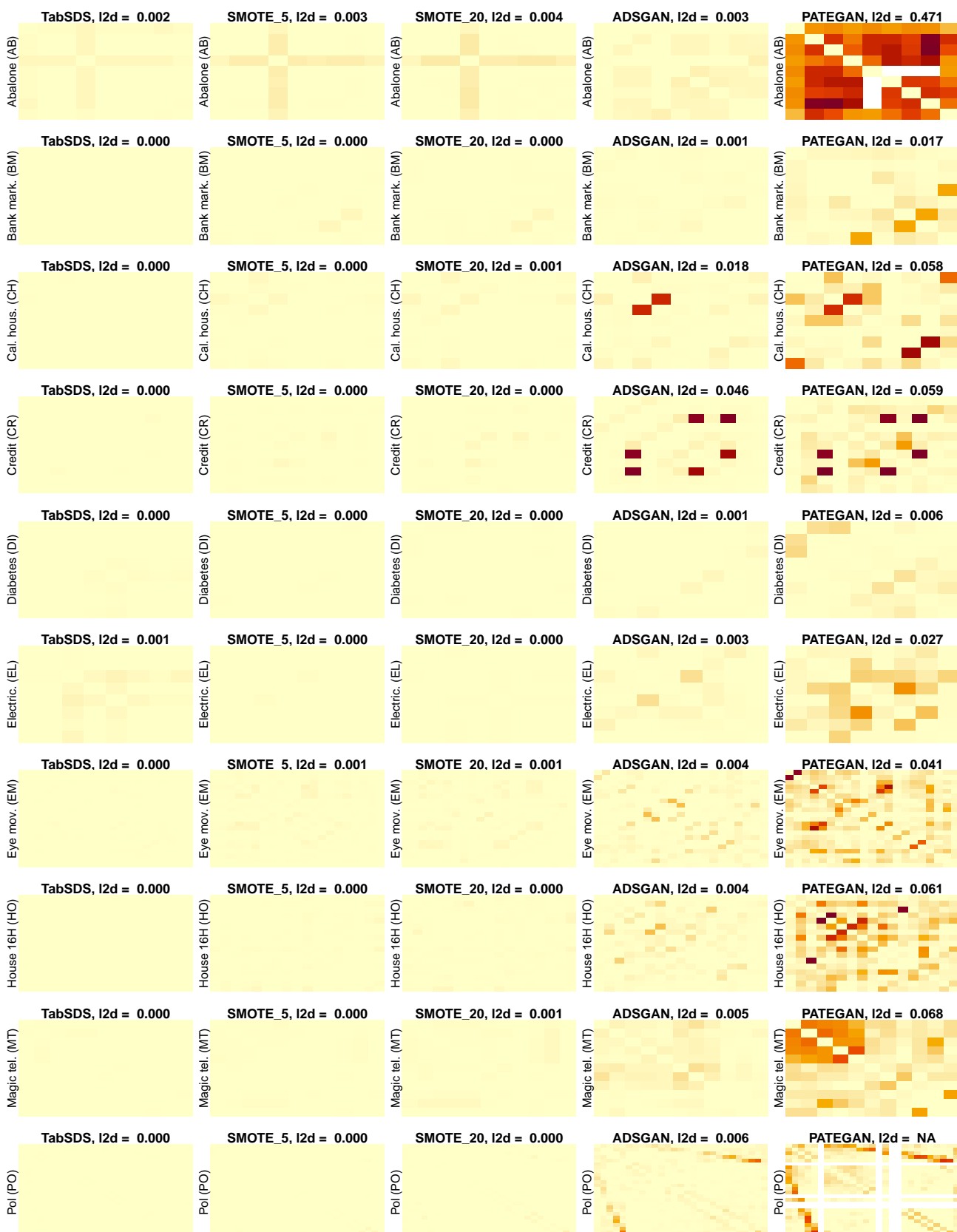

*Figure 33.* Absolute differences between association matrices for experiment set 2. Darker values indicate stronger discrepancy between the real and synthetic datasets. l2d indicates the L2 norm between real and synthetic association matrices. (l2d = 0 indicate a value lower than 0.001.)

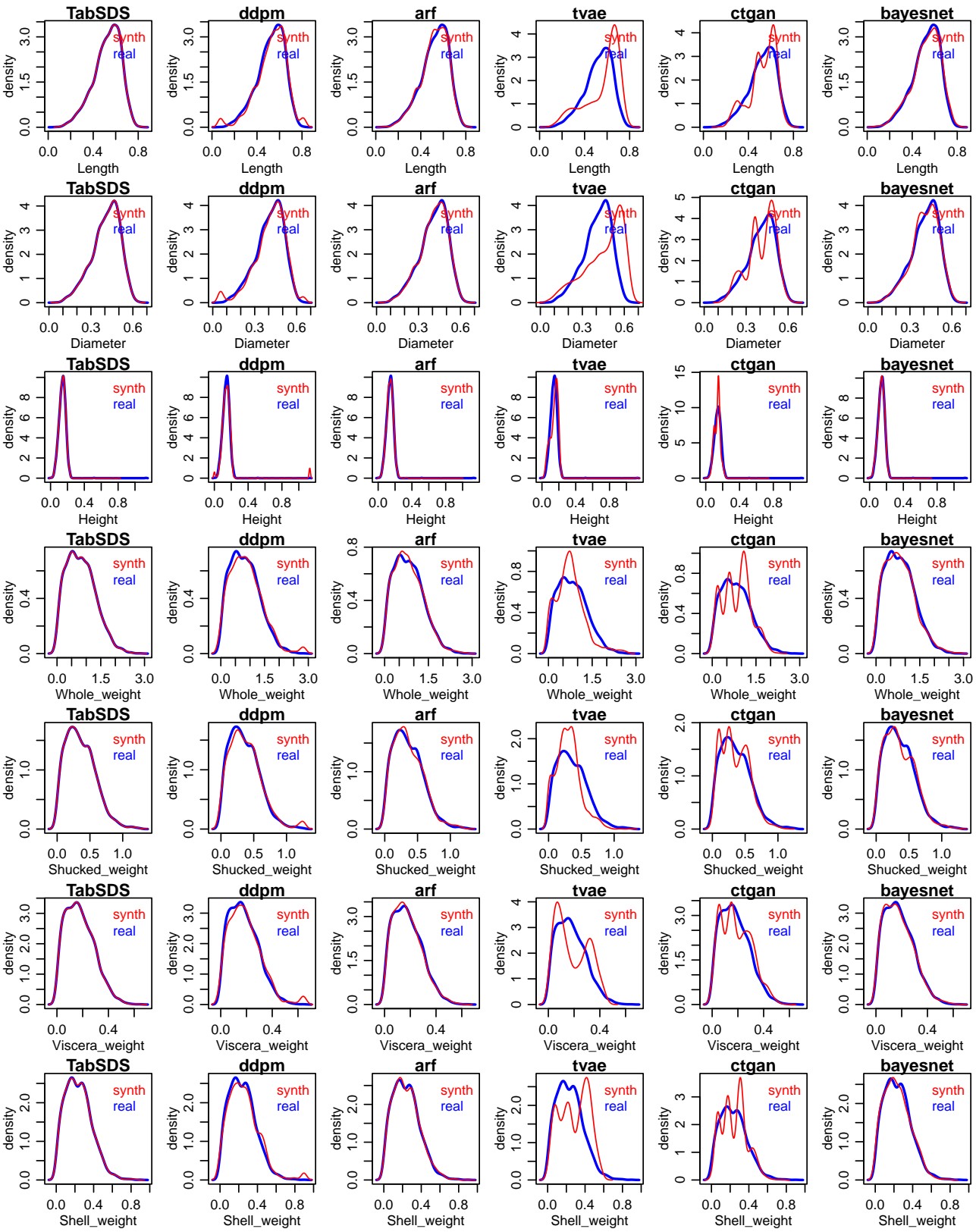

*Figure 34.* Subset of marginal distributions for the Abalone (AB) dataset. (Experiment set 1)

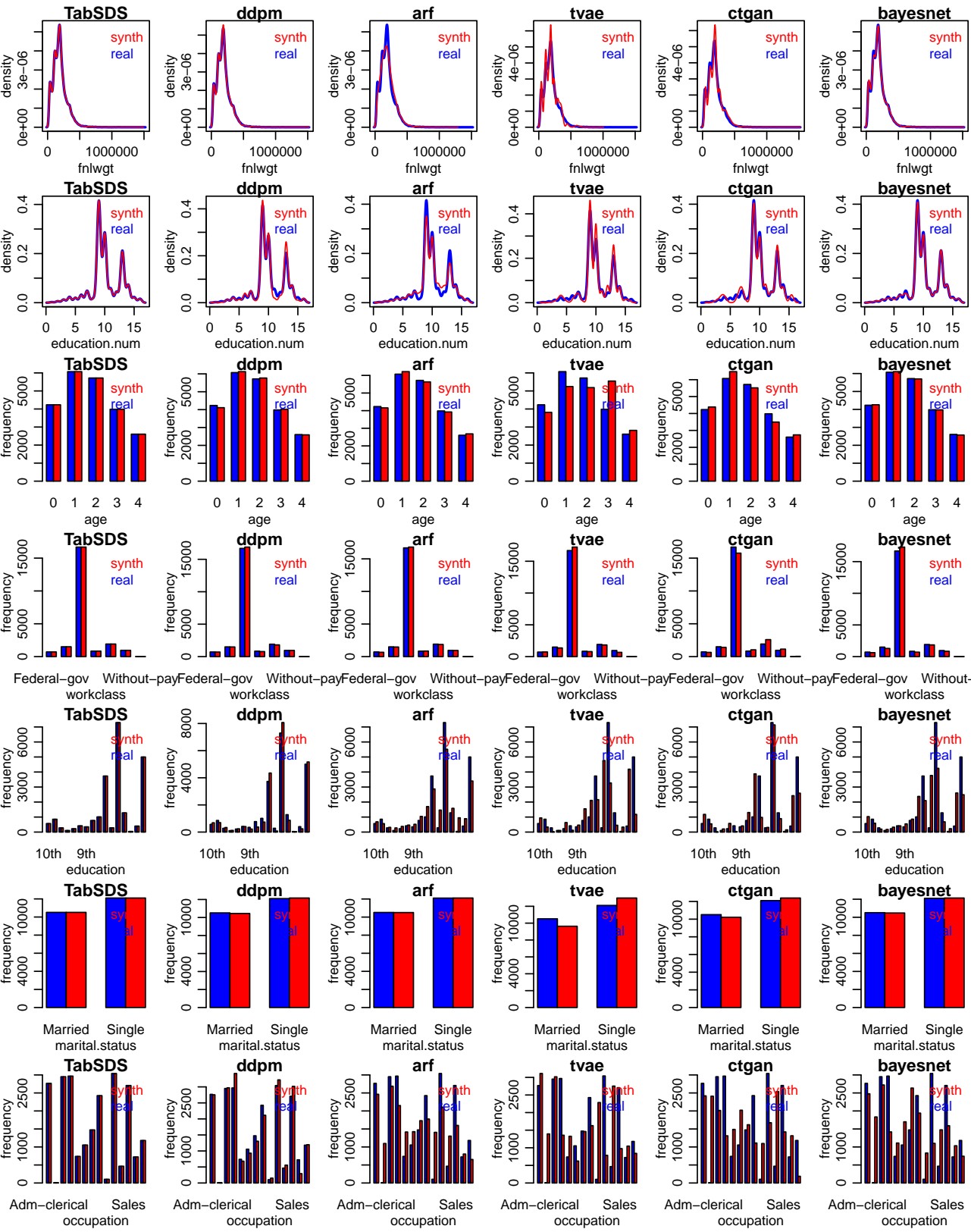

*Figure 35.* Subset of marginal distributions for the Adult (AD) dataset. (Experiment set 1)

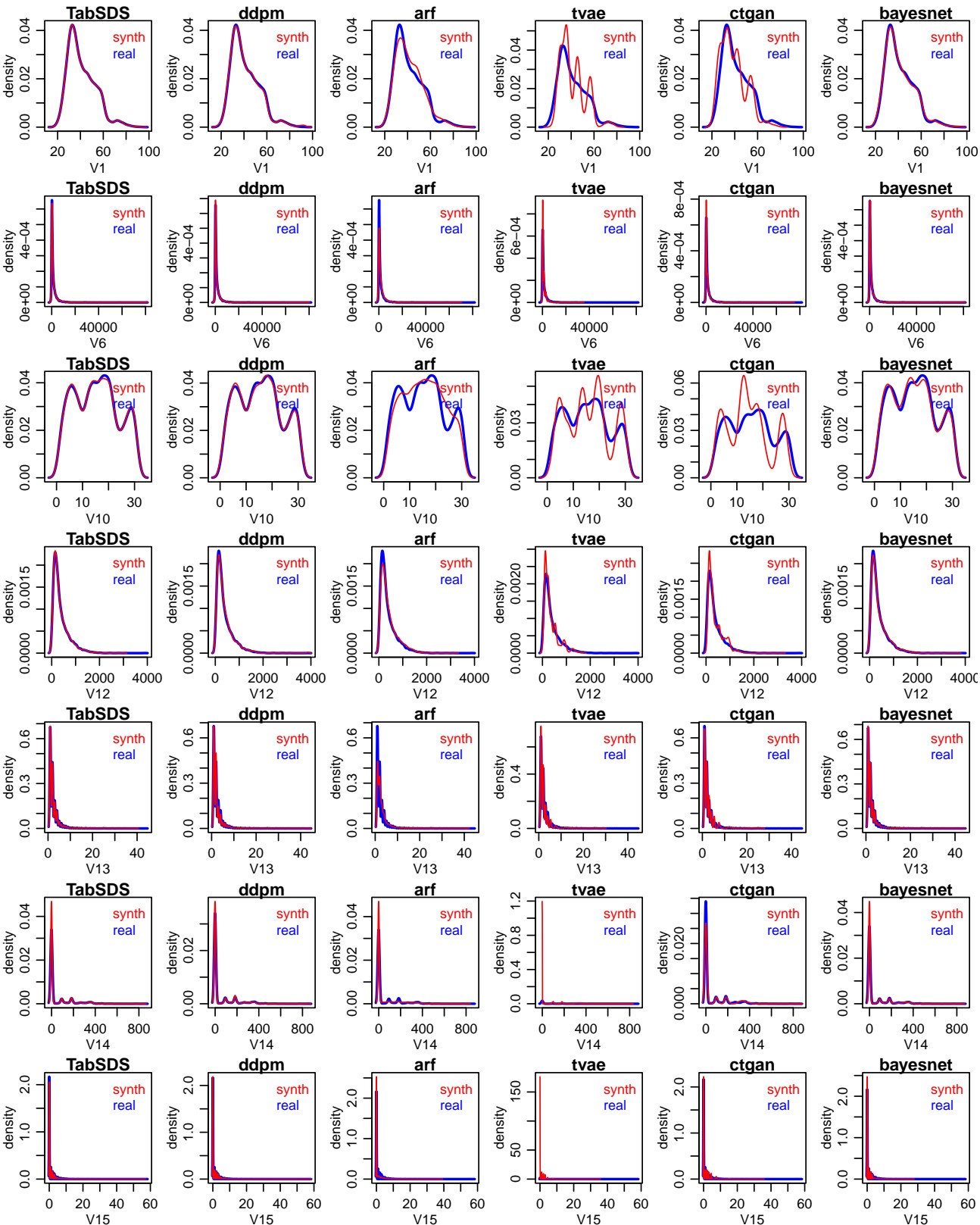

*Figure 36.* Subset of marginal distributions for the Bank marketing (BM) dataset. (Experiment set 1)

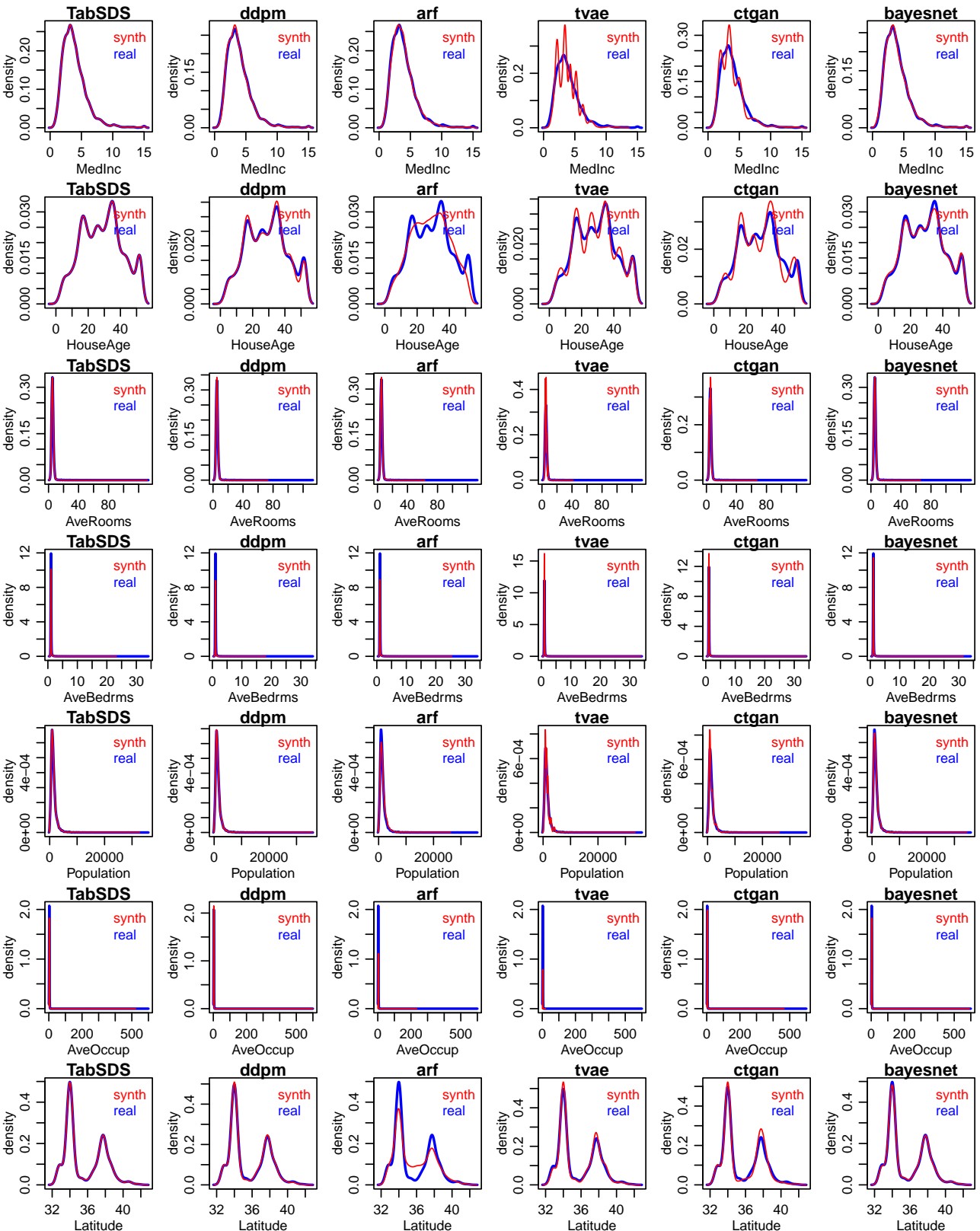

*Figure 37.* Subset of marginal distributions for the California housing (CH) dataset. (Experiment set 1)

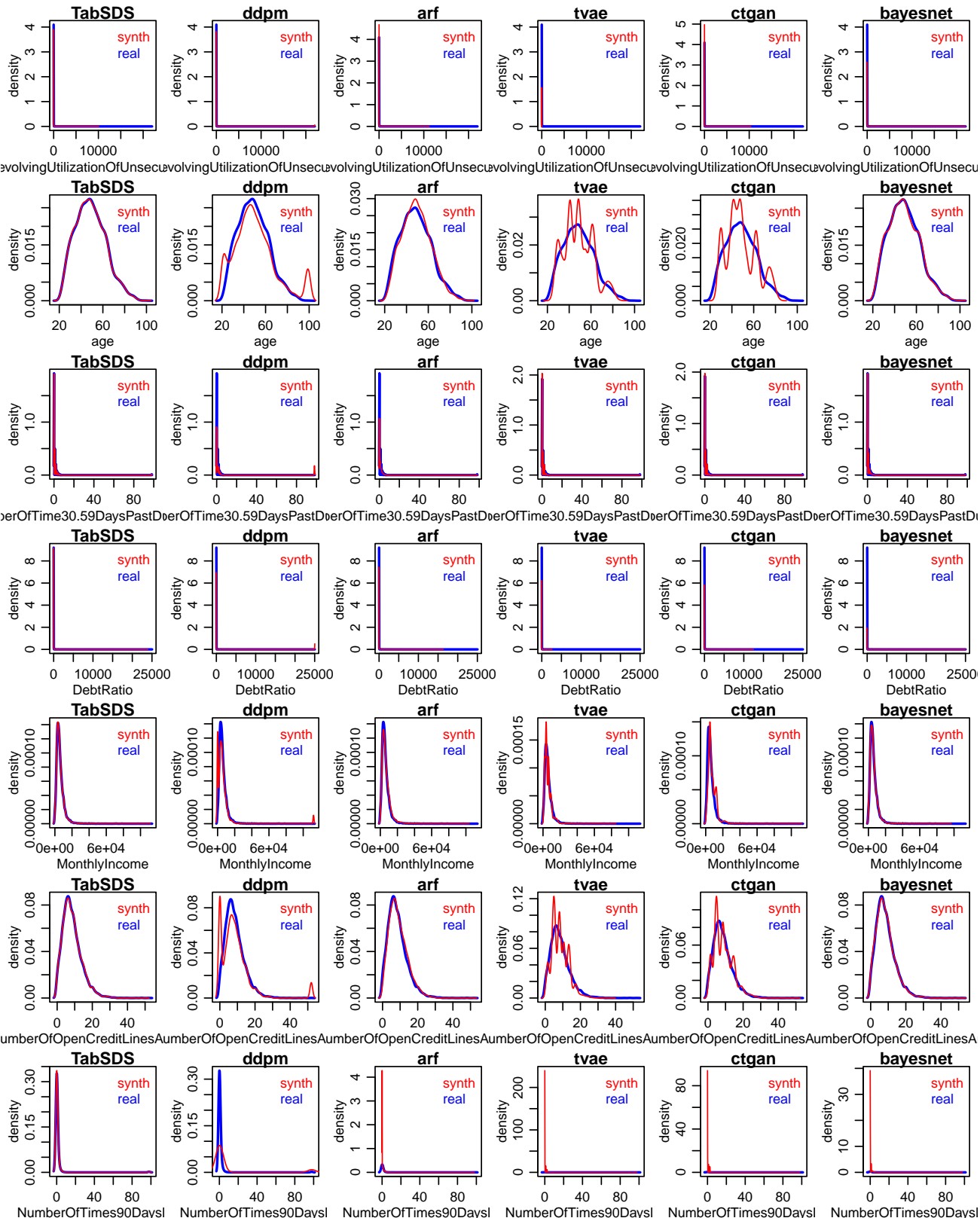

*Figure 38.* Subset of marginal distributions for the Credit (CR) dataset. (Experiment set 1)

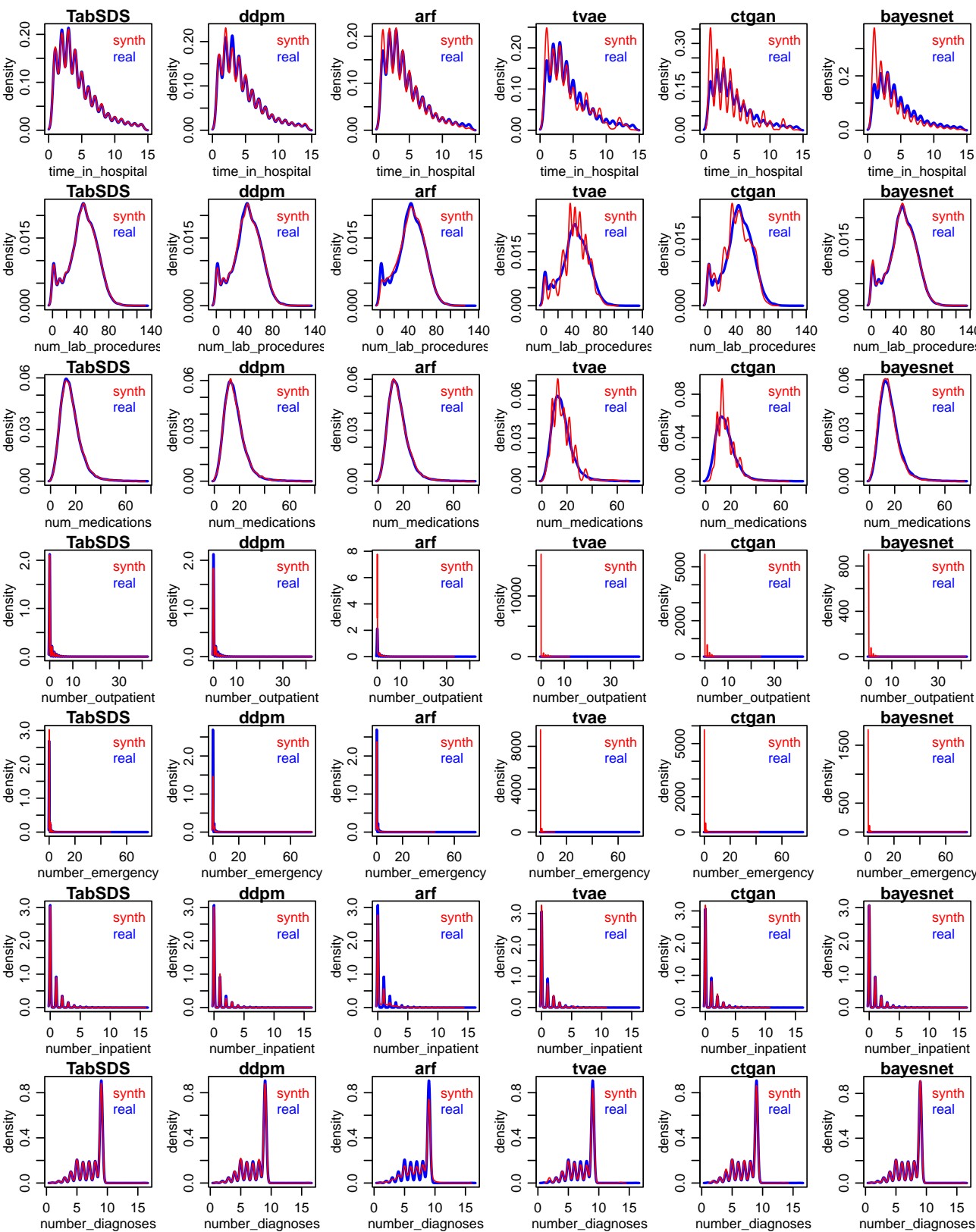

*Figure 39.* Subset of marginal distributions for the Diabetes 130US (DI) dataset. (Experiment set 1)

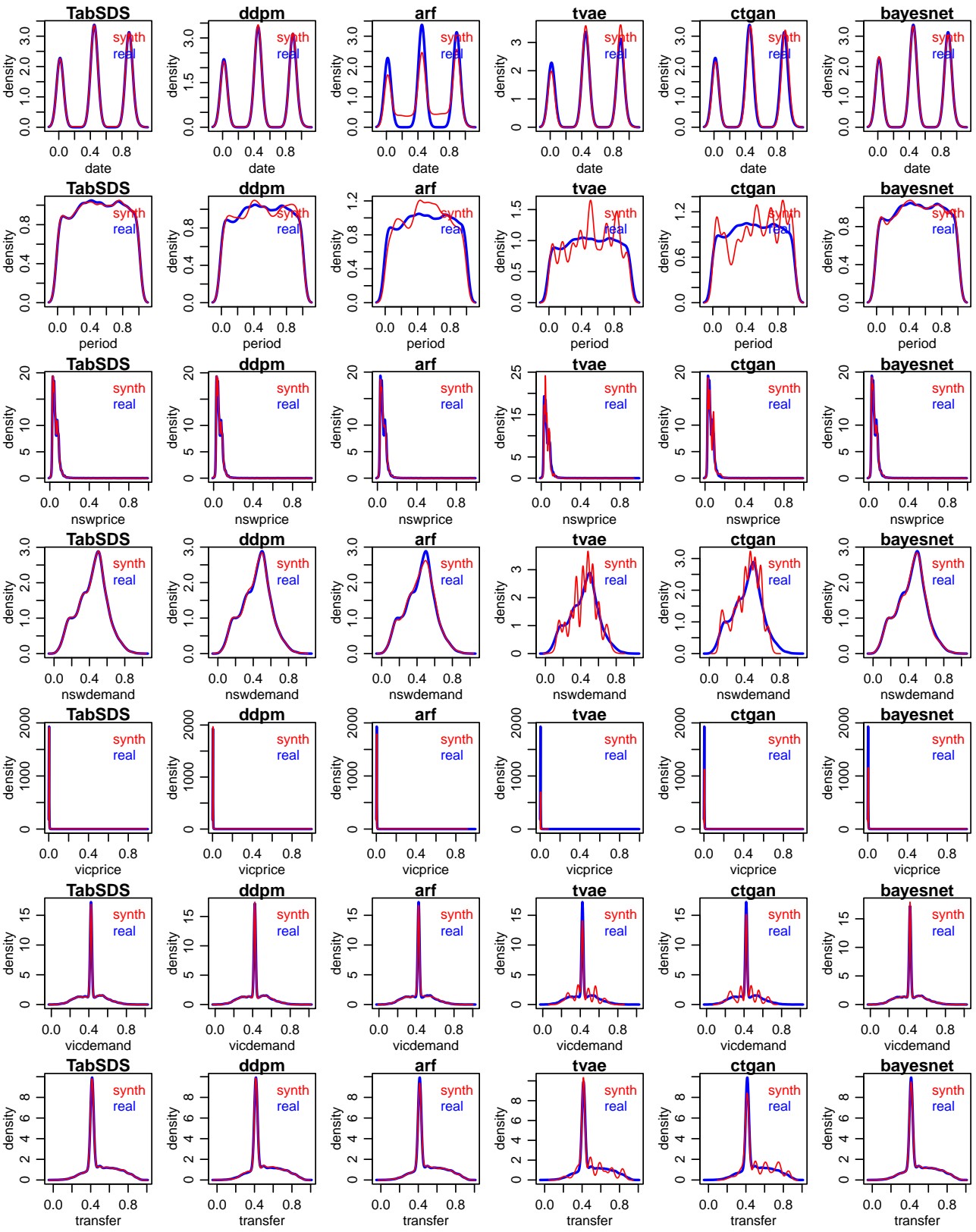

*Figure 40.* Subset of marginal distributions for the Electricity (EL) dataset. (Experiment set 1)

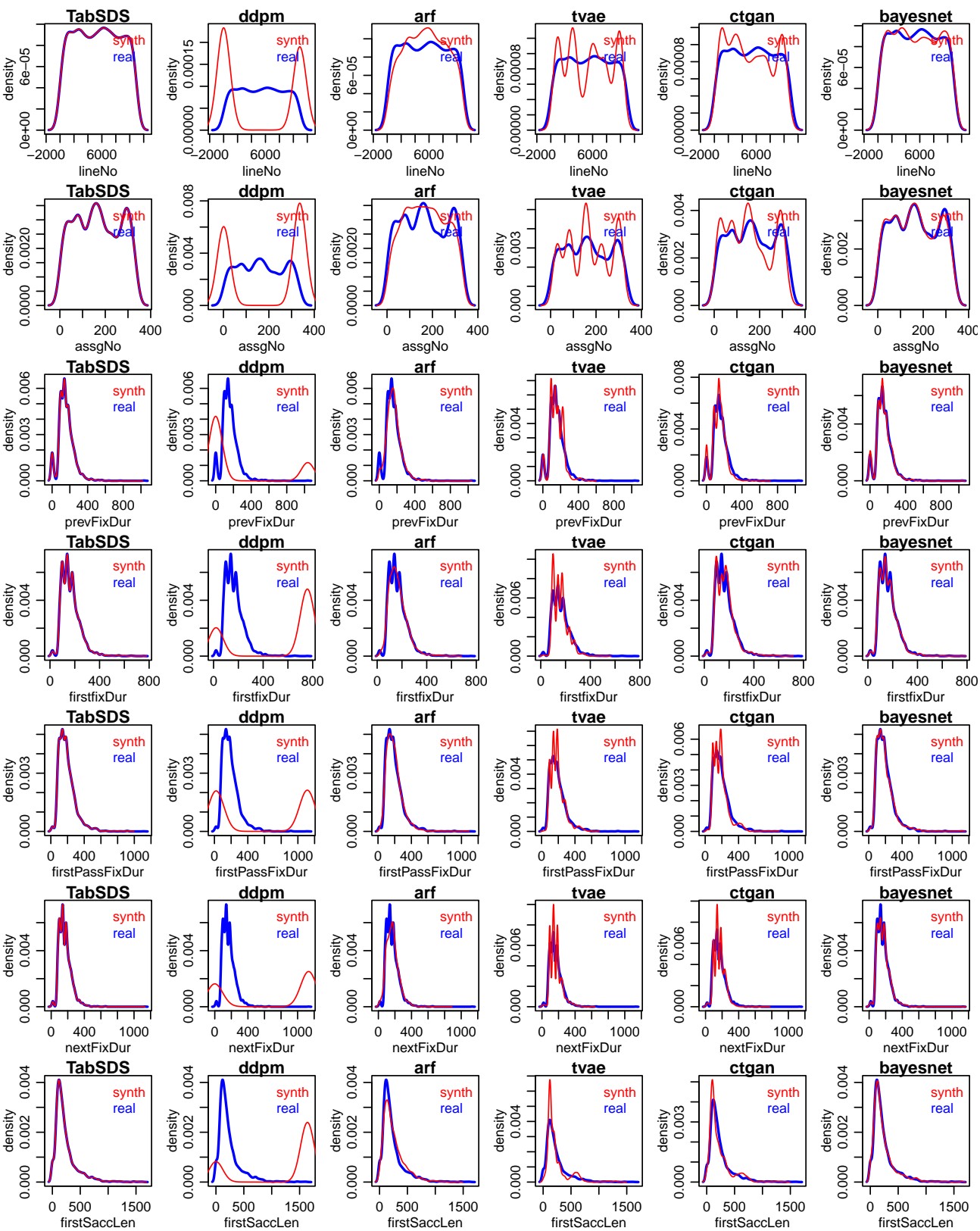

*Figure 41.* Subset of marginal distributions for the Eye movements (EM) dataset. (Experiment set 1)

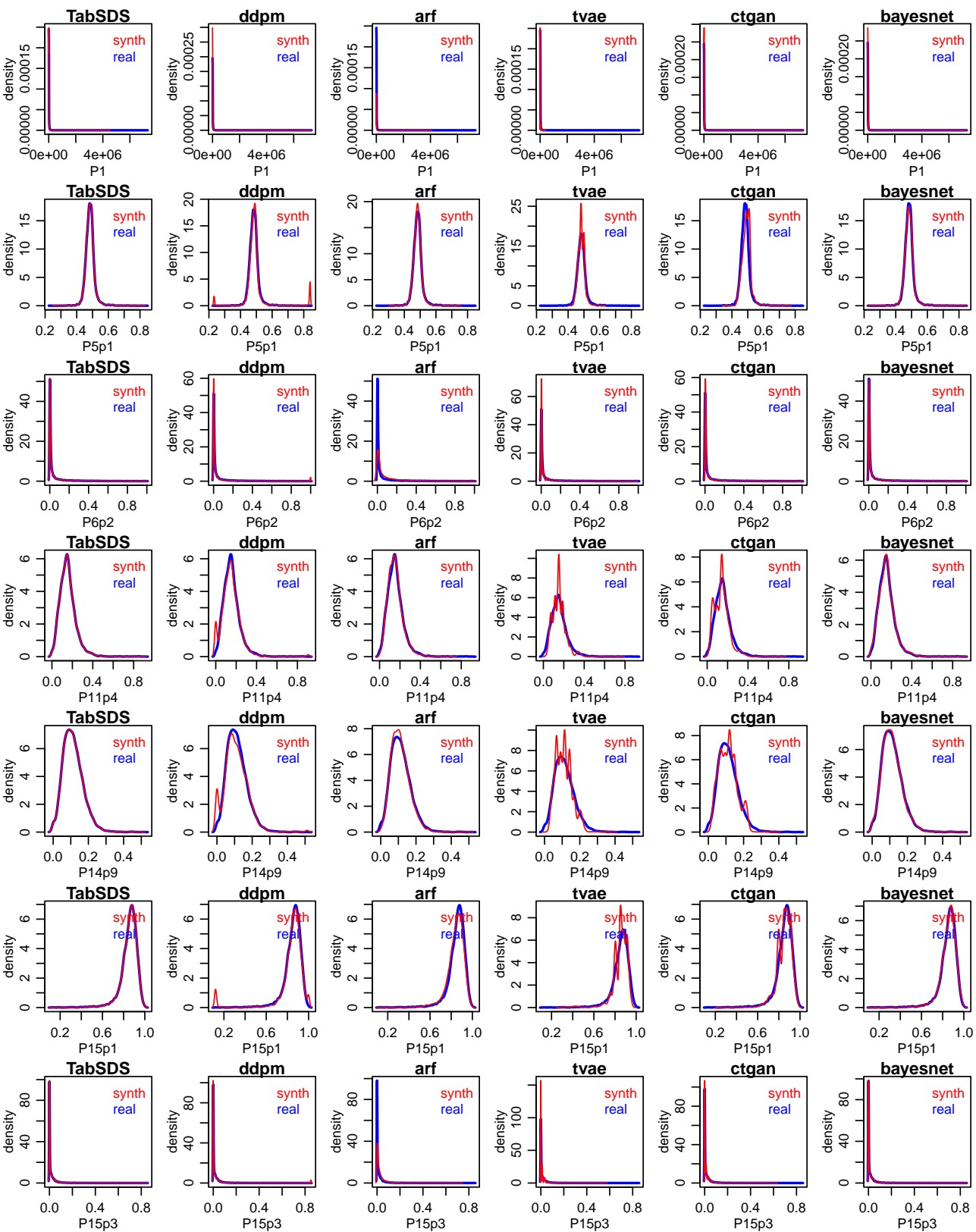

*Figure 42.* Subset of marginal distributions for the House 16H (HO) dataset. (Experiment set 1)

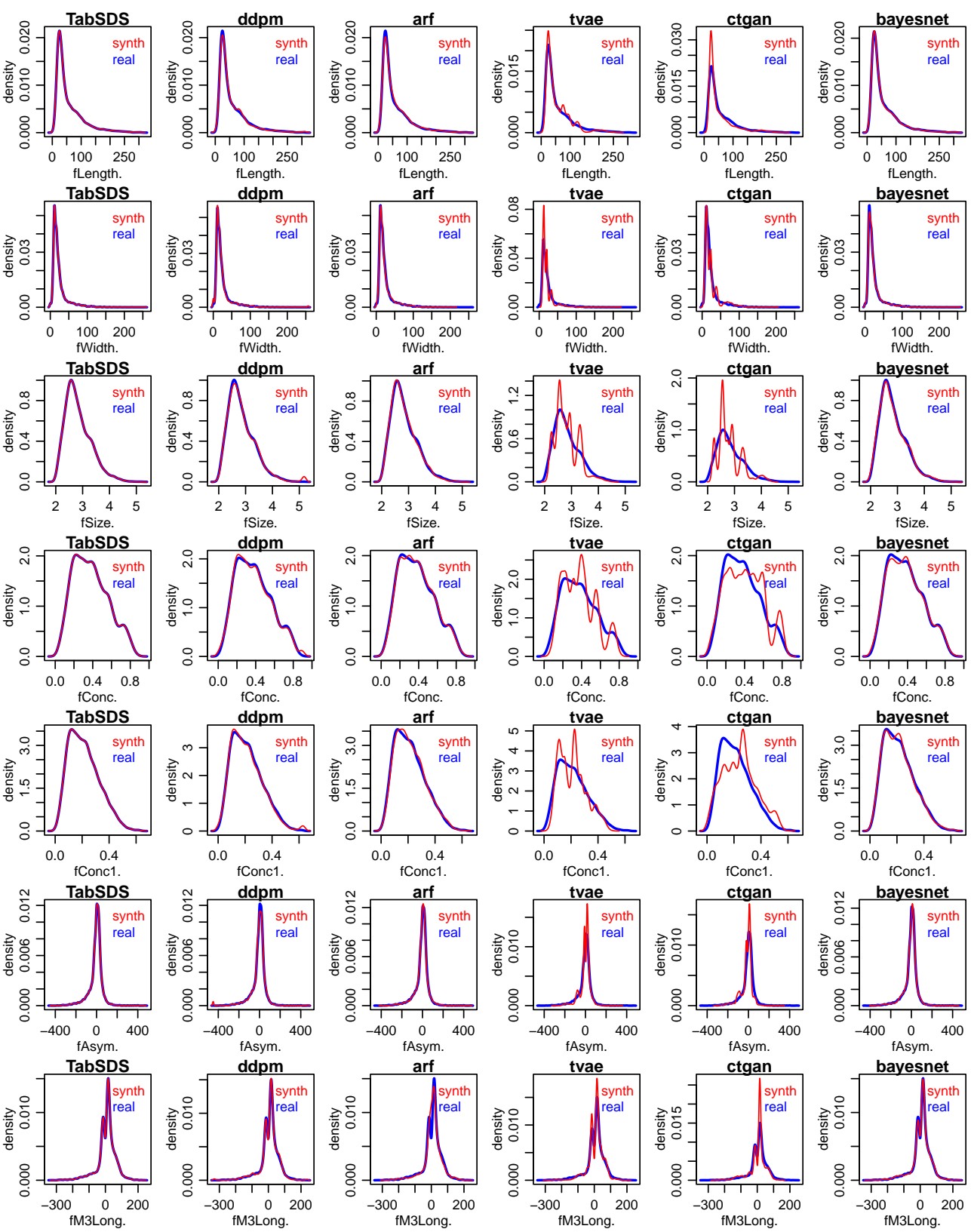

*Figure 43.* Subset of marginal distributions for the Magic telescope (MT) dataset. (Experiment set 1)

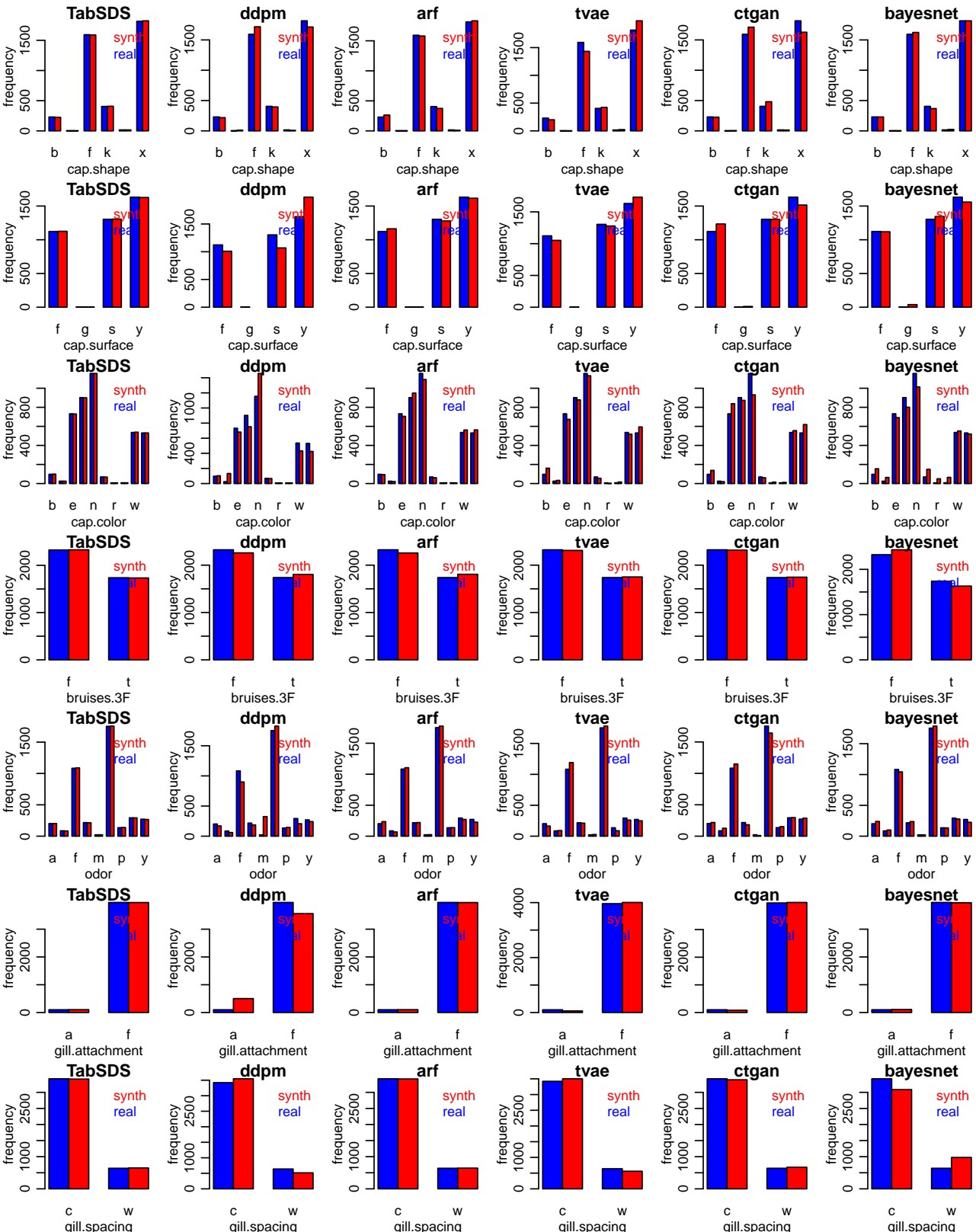

*Figure 44.* Subset of marginal distributions for the Mushroom (MU) dataset. (Experiment set 1)

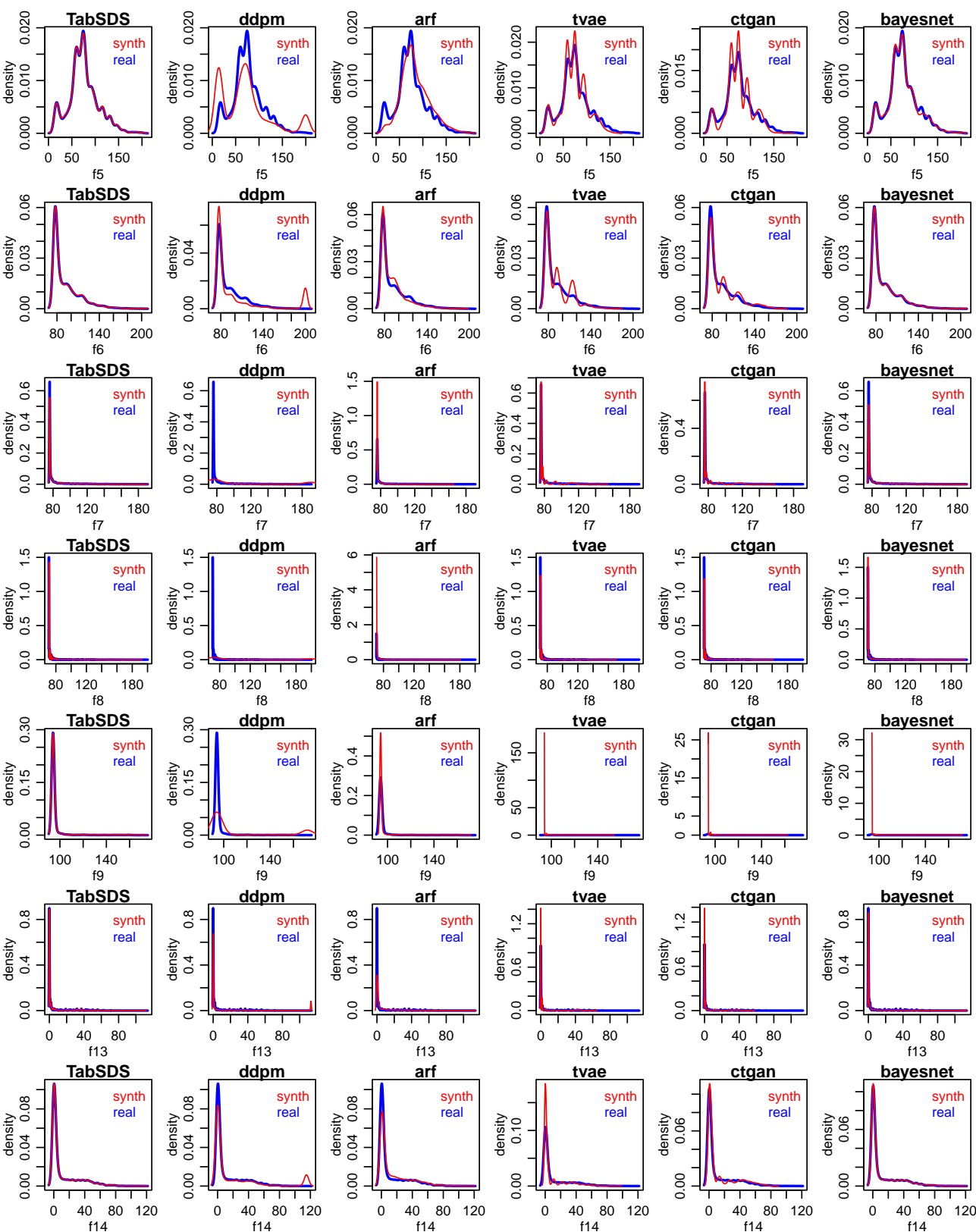

*Figure 45.* Subset of marginal distributions for the Pol (PO) dataset. (Experiment set 1)

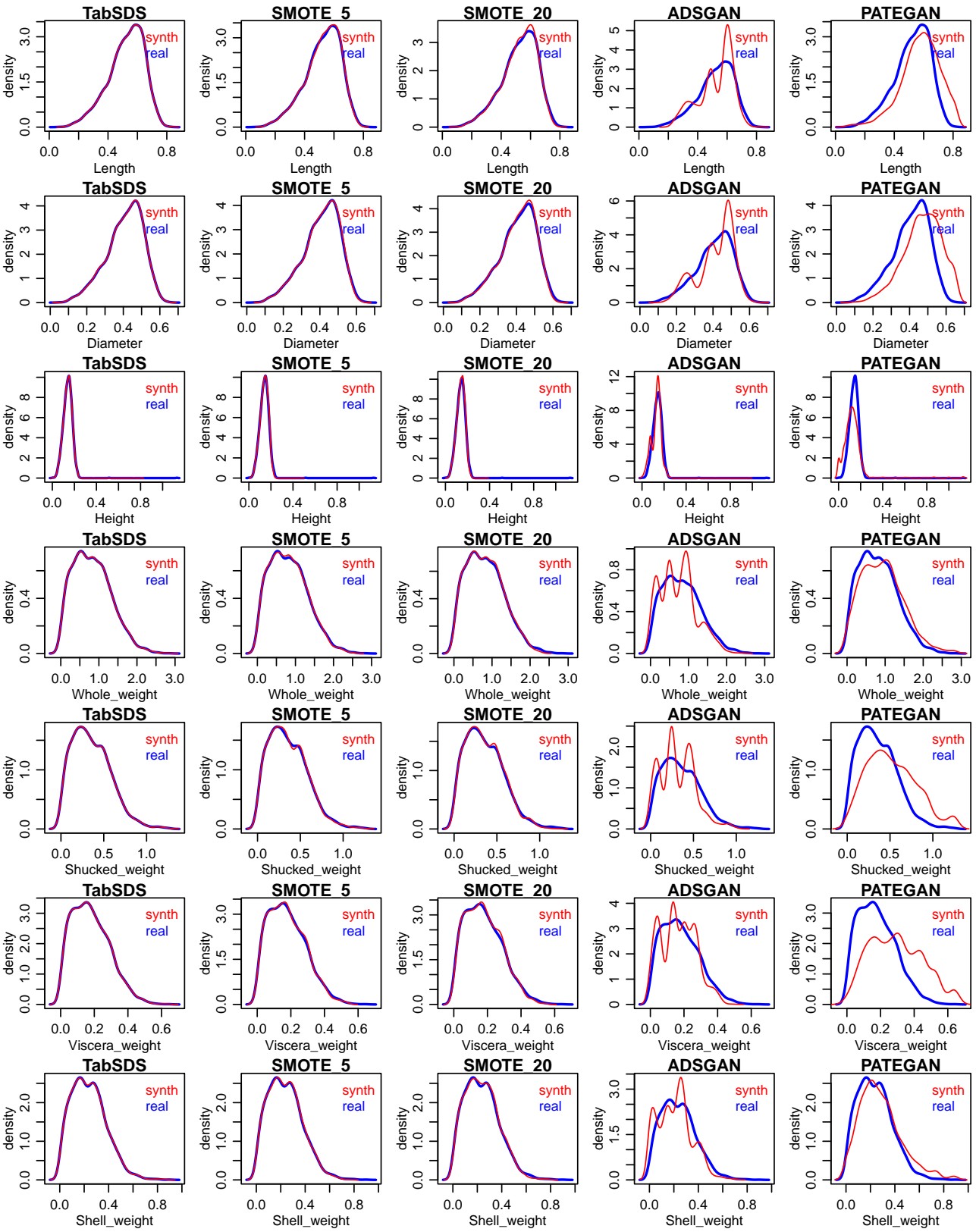

*Figure 46.* Subset of marginal distributions for the Abalone (AB) dataset. (Experiment set 2)

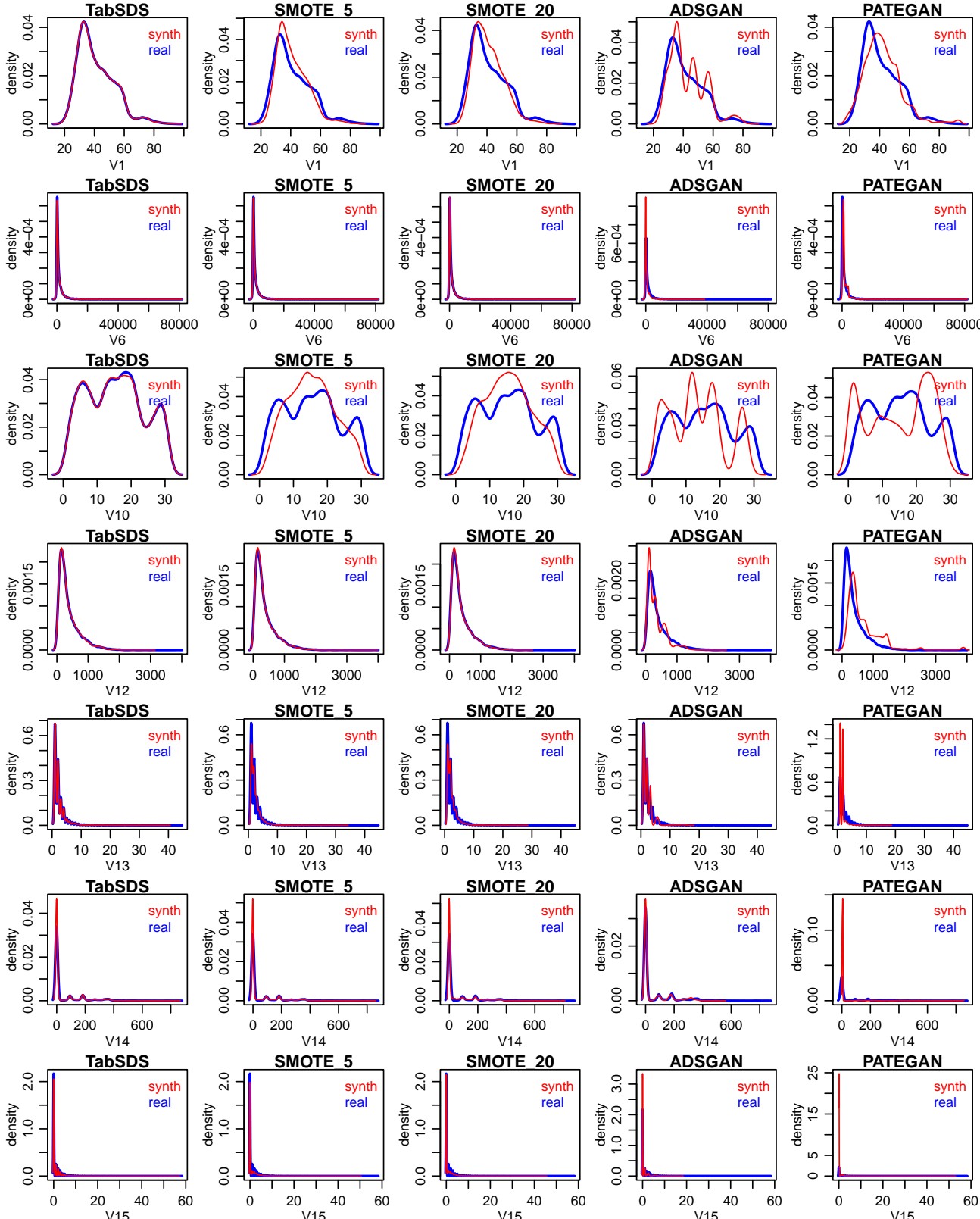

*Figure 47.* Subset of marginal distributions for the Bank marketing (BM) dataset. (Experiment set 2)

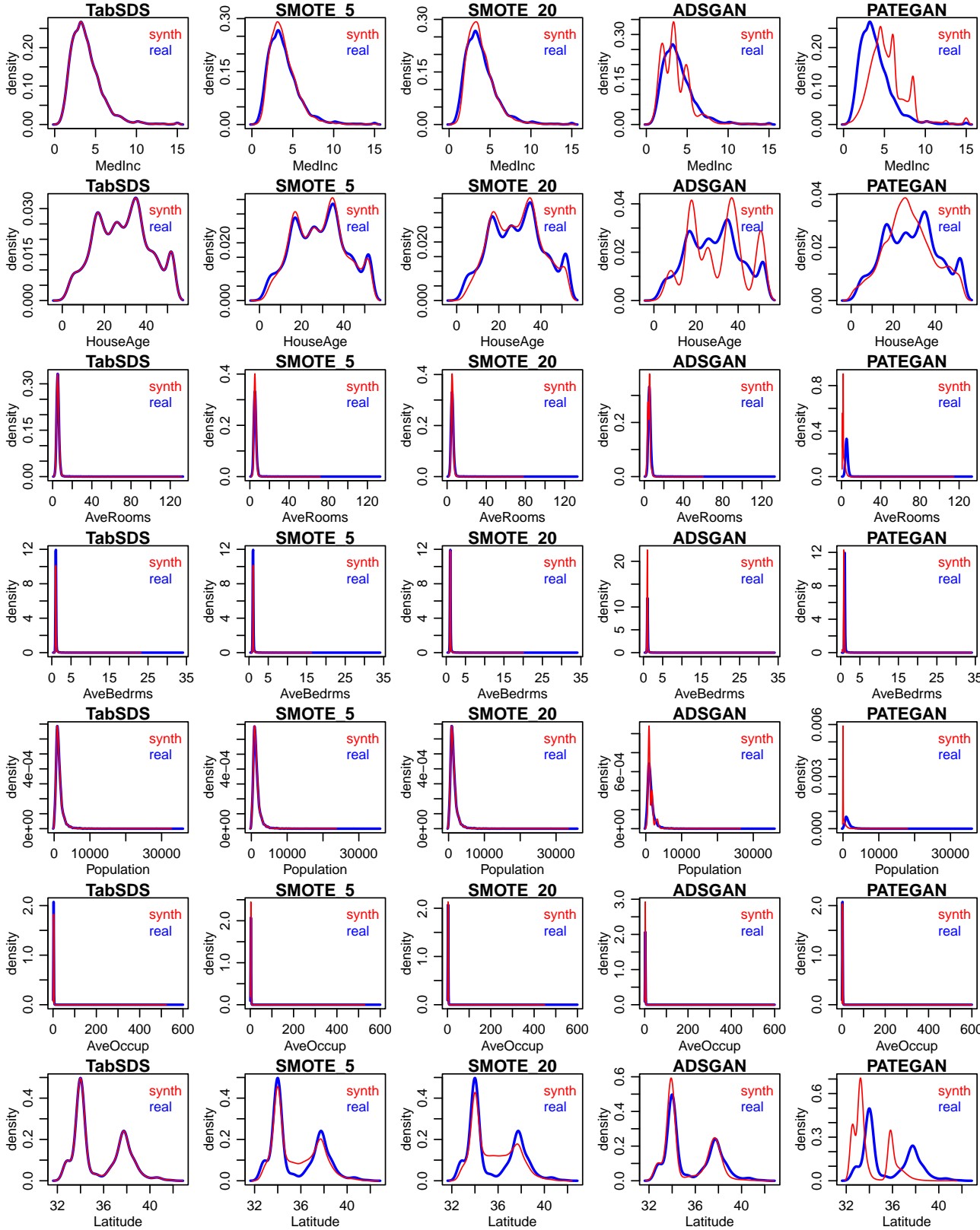

*Figure 48.* Subset of marginal distributions for the California housing (CH) dataset. (Experiment set 2)

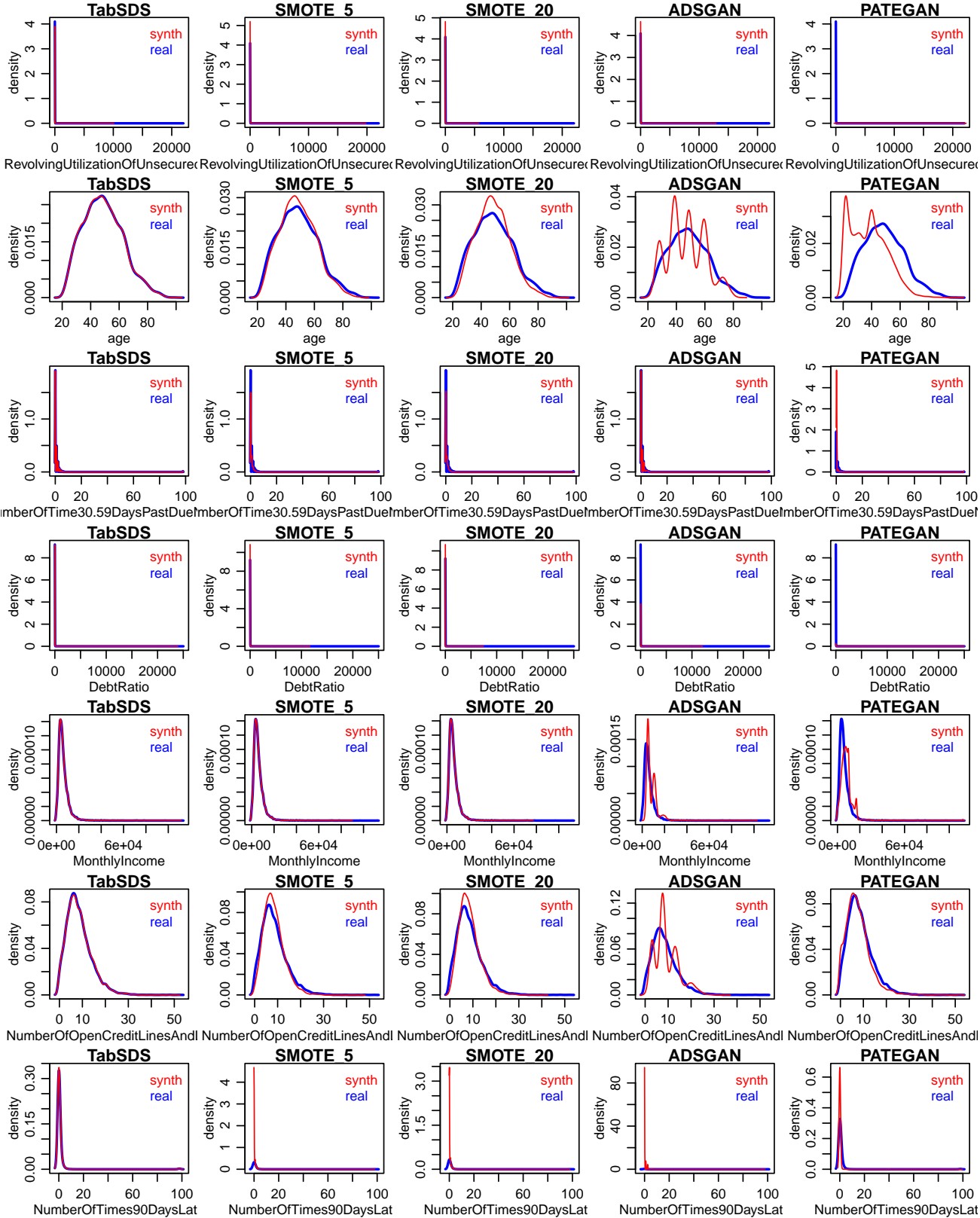

*Figure 49.* Subset of marginal distributions for the Credit (CR) dataset. (Experiment set 2)

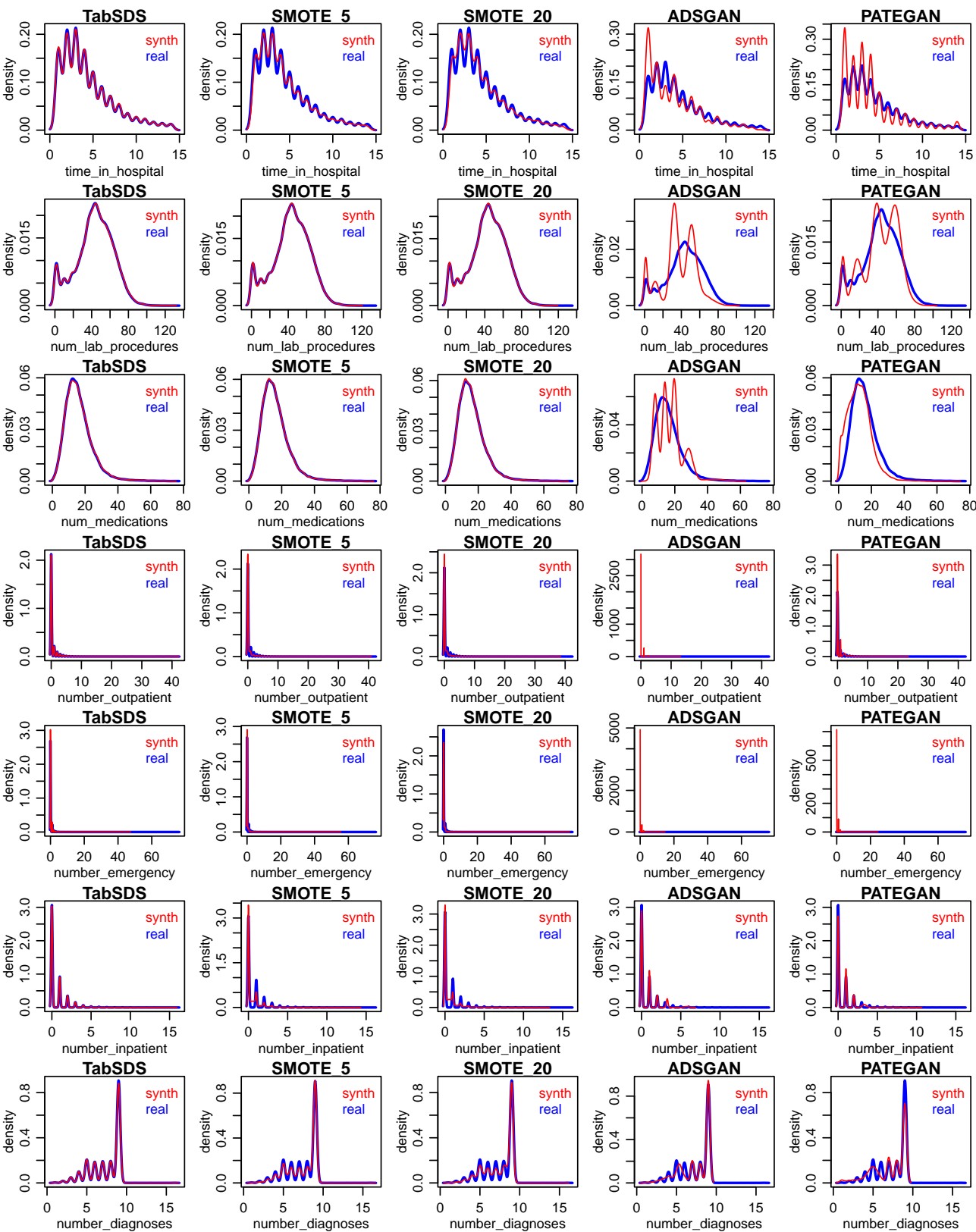

*Figure 50.* Subset of marginal distributions for the Diabetes 130US (DI) dataset. (Experiment set 2)

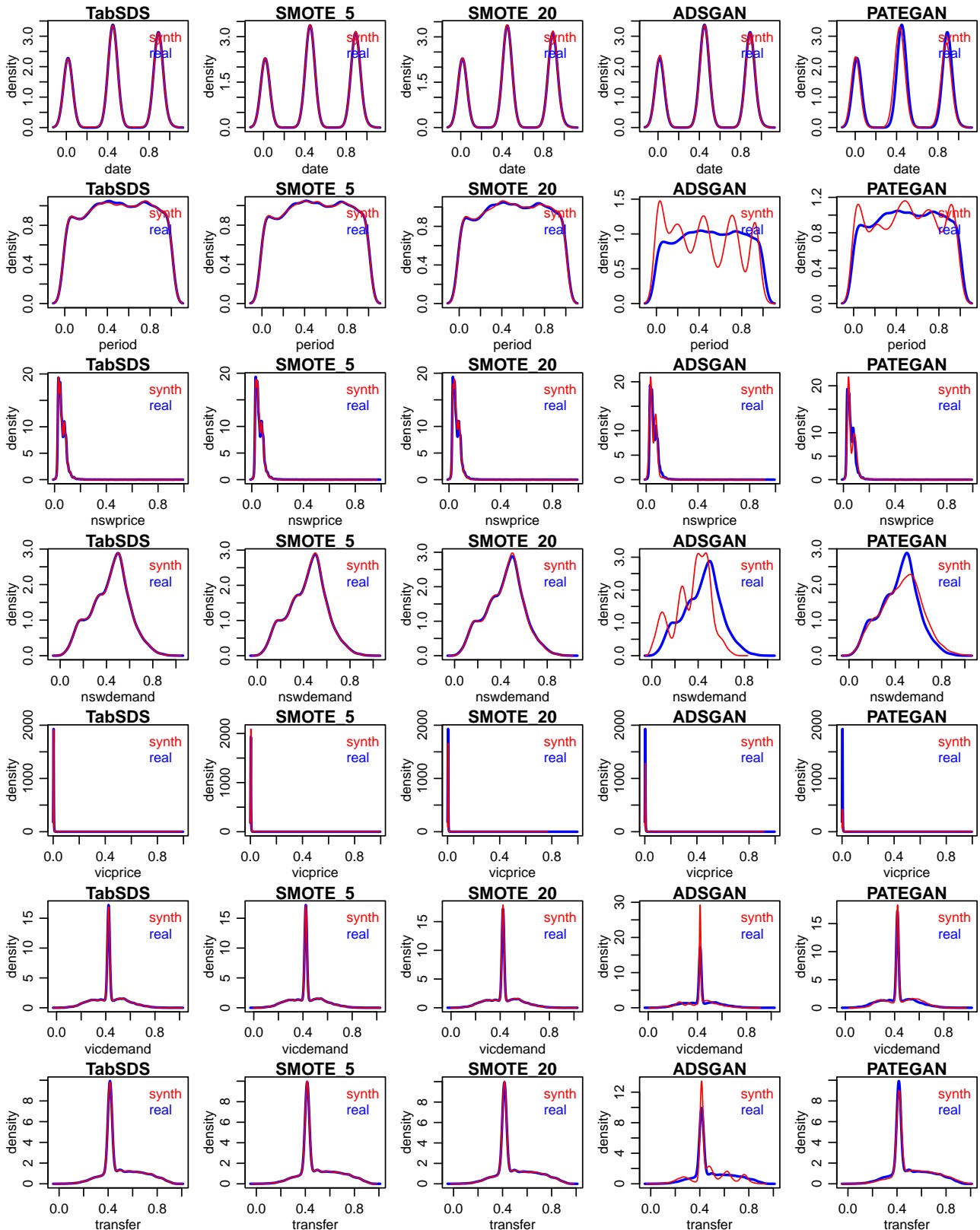

*Figure 51.* Subset of marginal distributions for the Electricity (EL) dataset. (Experiment set 2)

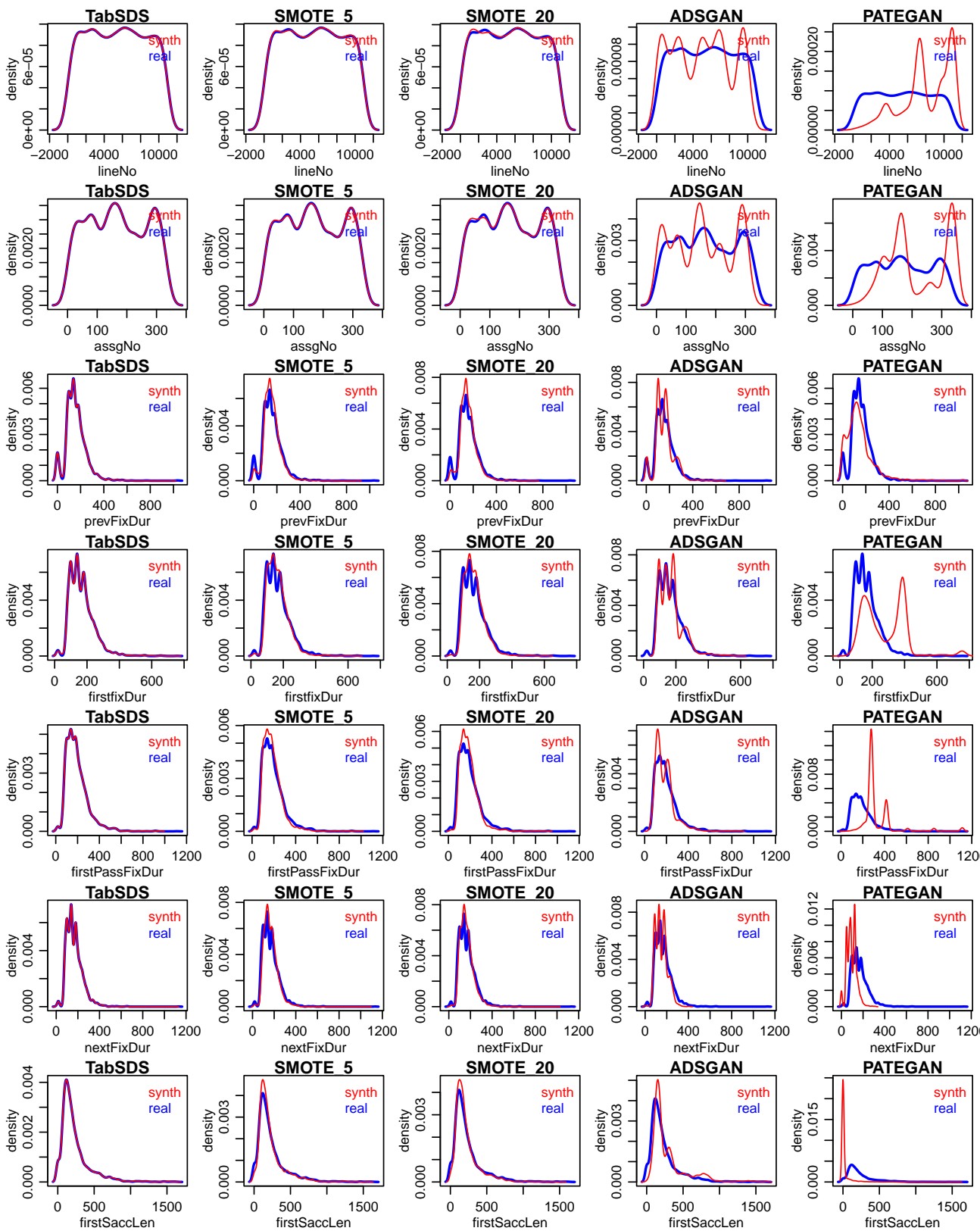

*Figure 52.* Subset of marginal distributions for the Eye movements (EM) dataset. (Experiment set 2)

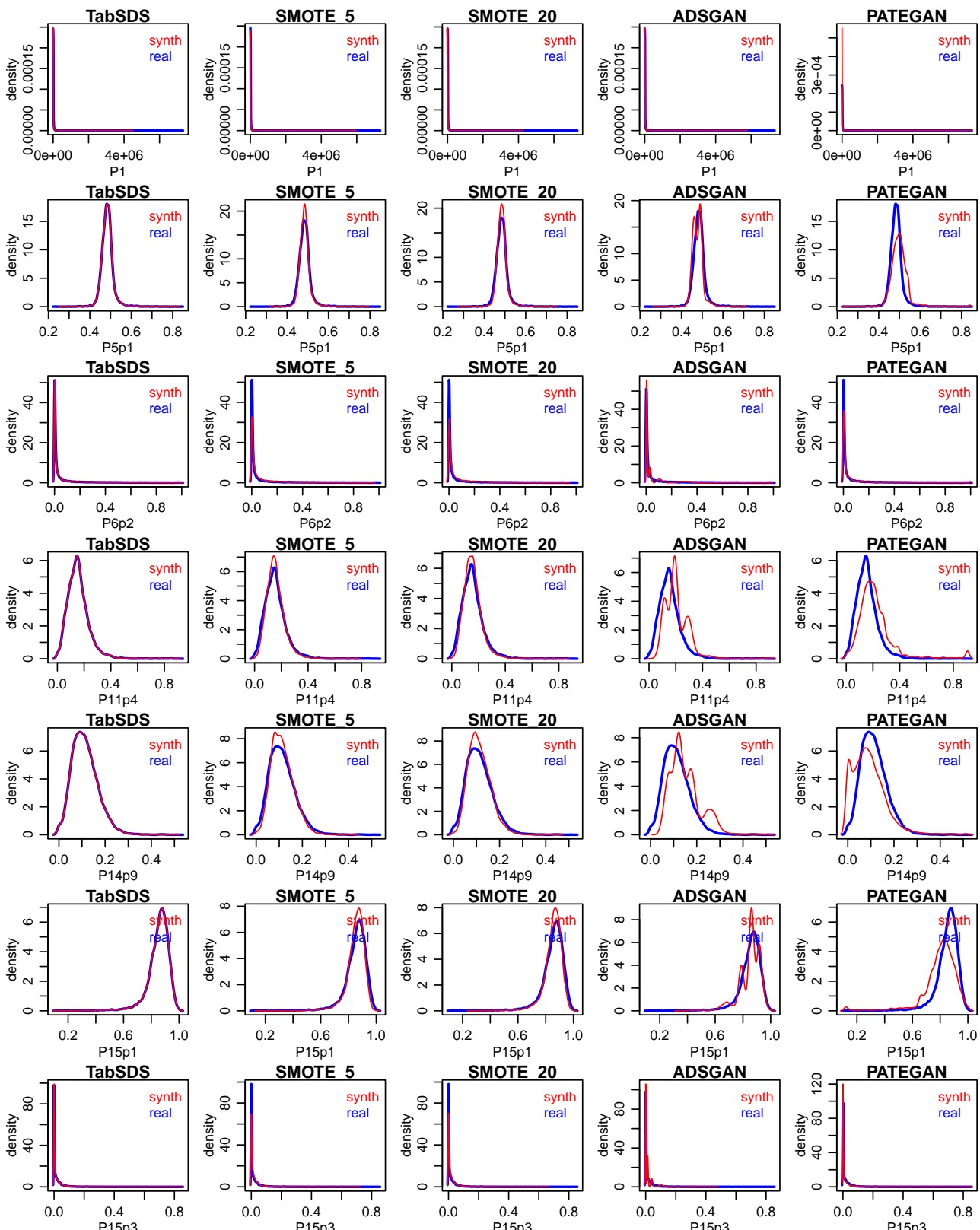

*Figure 53.* Subset of marginal distributions for the House 16H (HO) dataset. (Experiment set 2)

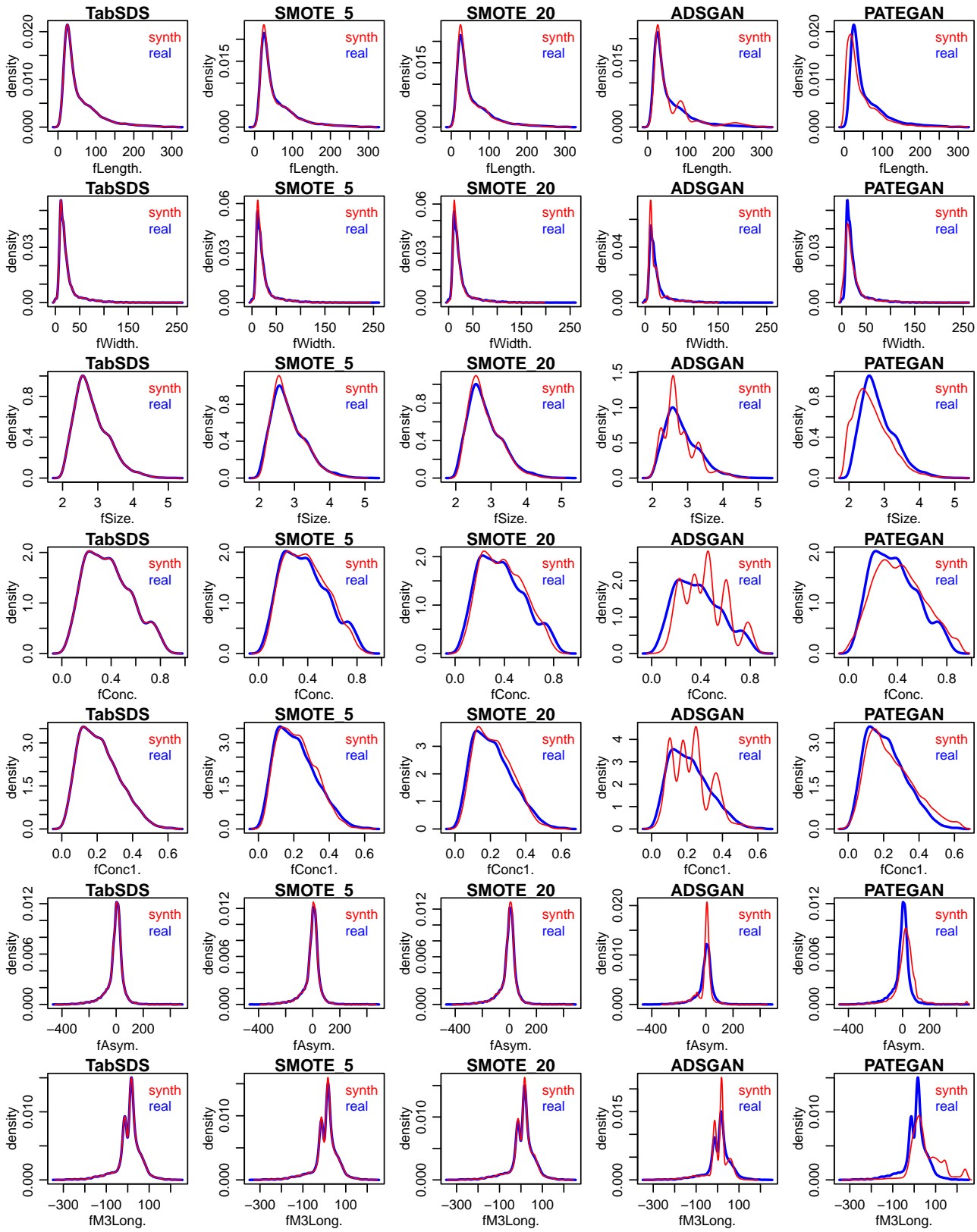

*Figure 54.* Subset of marginal distributions for the Magic telescope (MT) dataset. (Experiment set 2)

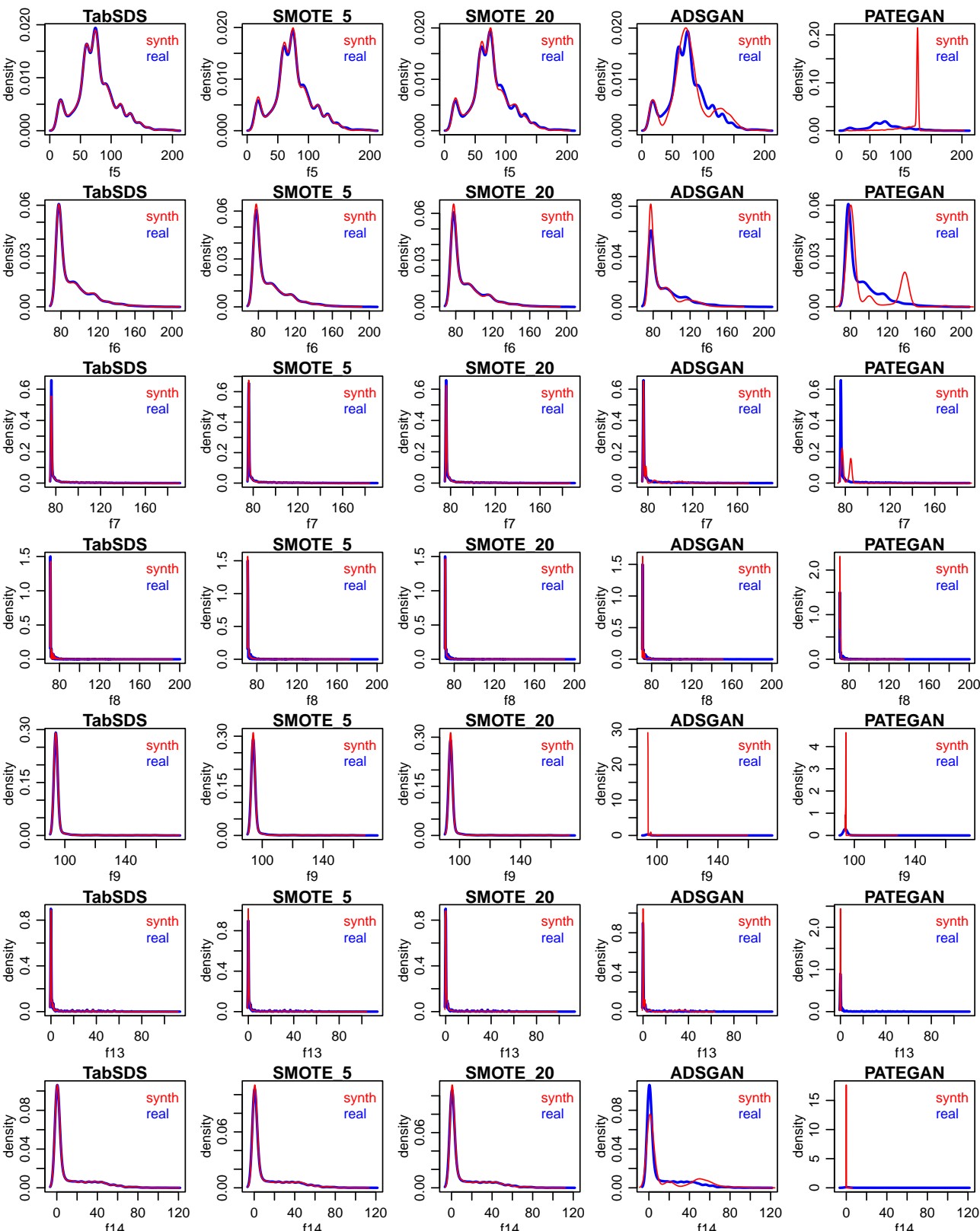

*Figure 55.* Subset of marginal distributions for the Pol (PO) dataset. (Experiment set 2)

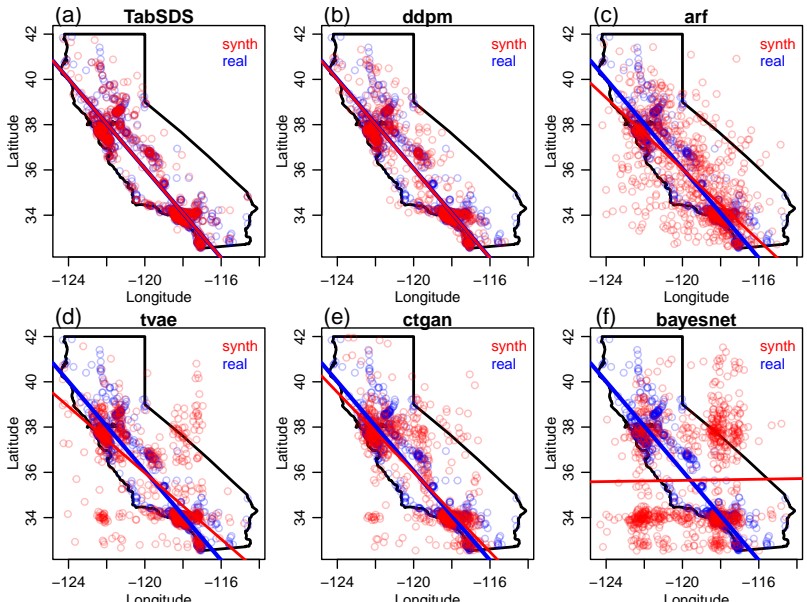

*Figure 56.* Scatter plots of Latitude versus Longitude in the California housing (CH) dataset.

### G.9. Evaluation of the Influence of the Sampling Proportion Parameter

All the TabSDS results presented in the main text were computed using $p = 0.5$. Here, we report and compare results based on the nosier choice of $p = 0.1$. Figures 57 to 68 report the detection test, ML efficiency, domias MIA, and DCR distributions over on the same grid of 24 $n_c$ values as before. Results based on $p = 0.1$ and $p = 0.5$ are shown in blue and red, respectively. Overall, the adoption of noisier marginal distributions (lower $p$) lead to an accentuated degradation of the detection test (note the considerably higher detection scores in panel a across most figures) without a corresponding improvement in privacy. (Overall, we do not see very accentuated differences in panels c and d across the figures.) For this reason, we adopted $p = 0.5$ in our evaluations.

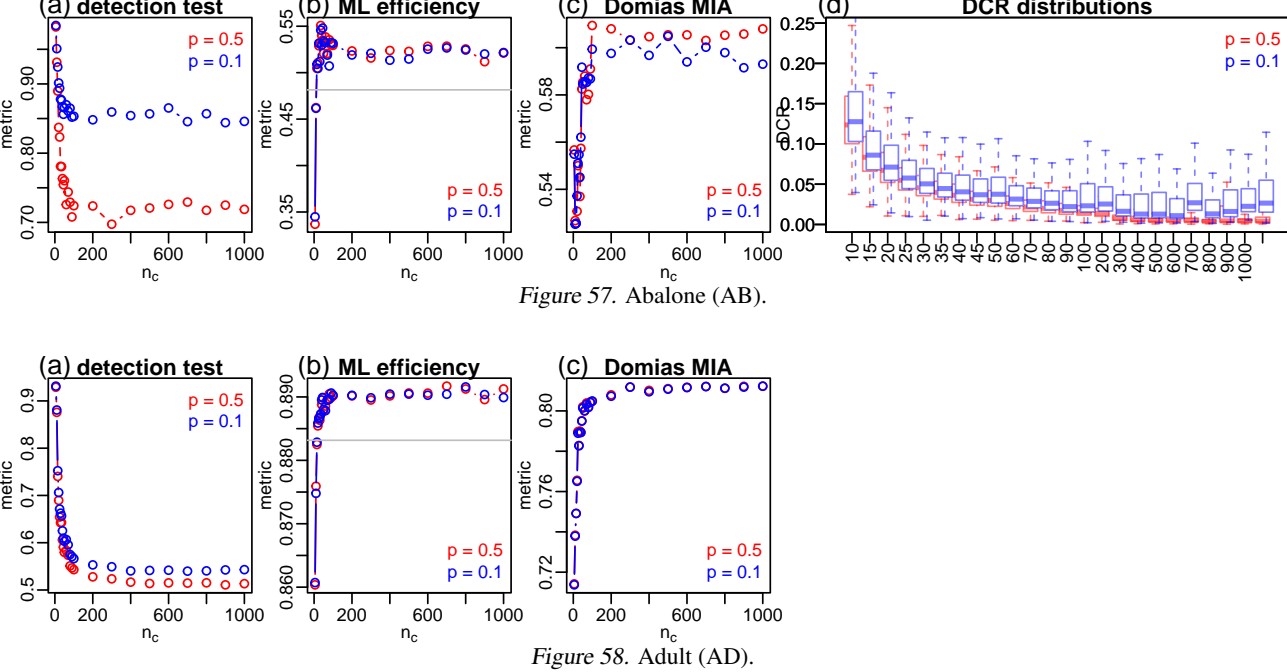

*Figure 57.* Abalone (AB).

*Figure 58.* Adult (AD).

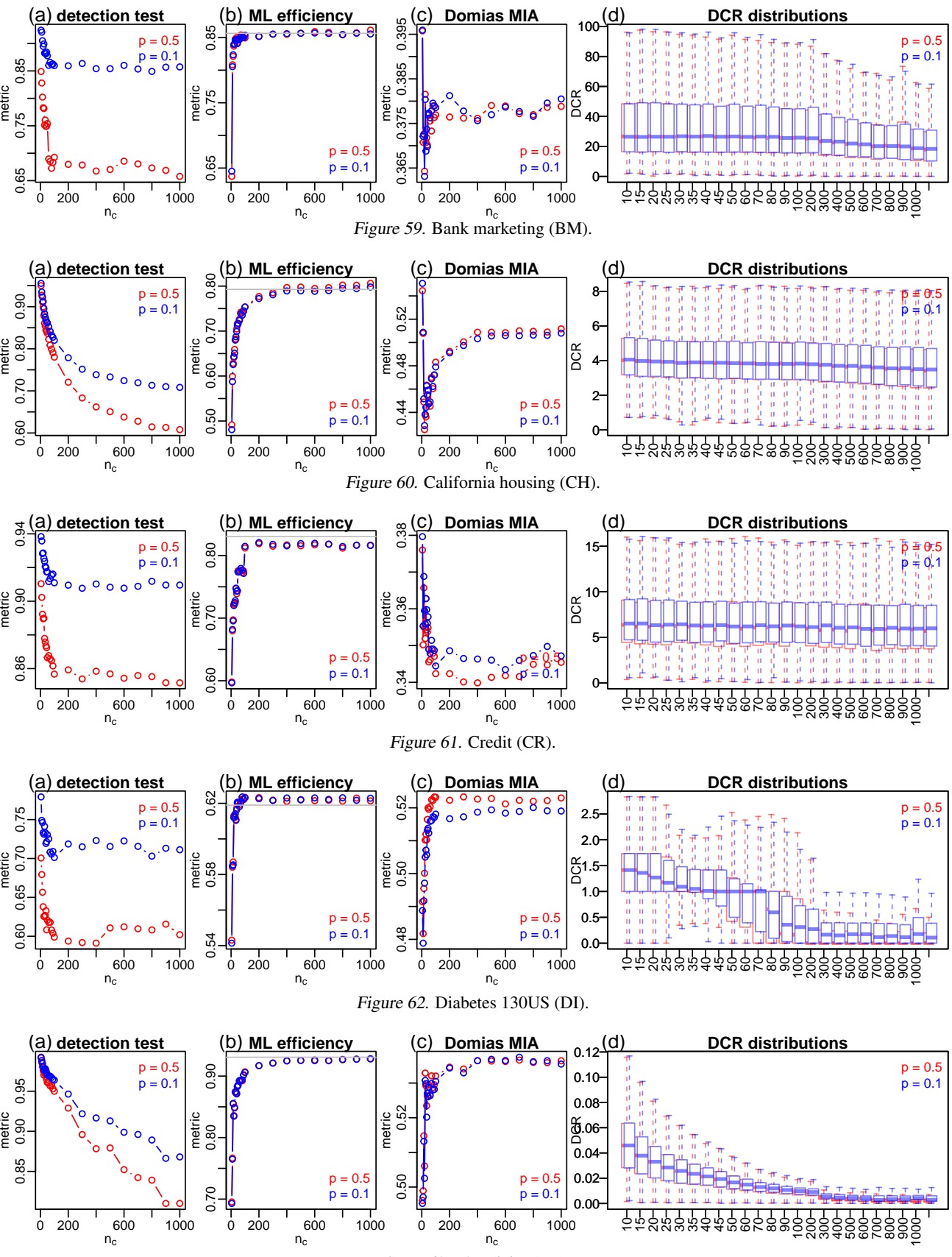

*Figure 59.* Bank marketing (BM).

*Figure 60.* California housing (CH).

*Figure 61.* Credit (CR).

*Figure 62.* Diabetes 130US (DI).

*Figure 63.* Electricity (EL).

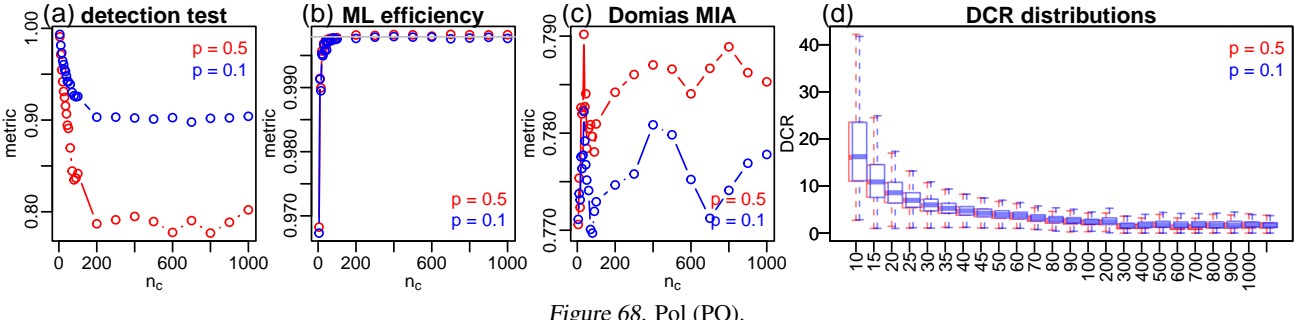

*Figure 64.* Eye movements (EM).

*Figure 65.* House 16H (HO).

*Figure 66.* Magic telescope (MT).

*Figure 67.* Mushroom (MU). (Note that for purely categorical datasets the choice of $p$ is immaterial since it is not used in the Categorical-SJPPDS algorithm (Algorithm 4).)

*Figure 68.* Pol (PO).

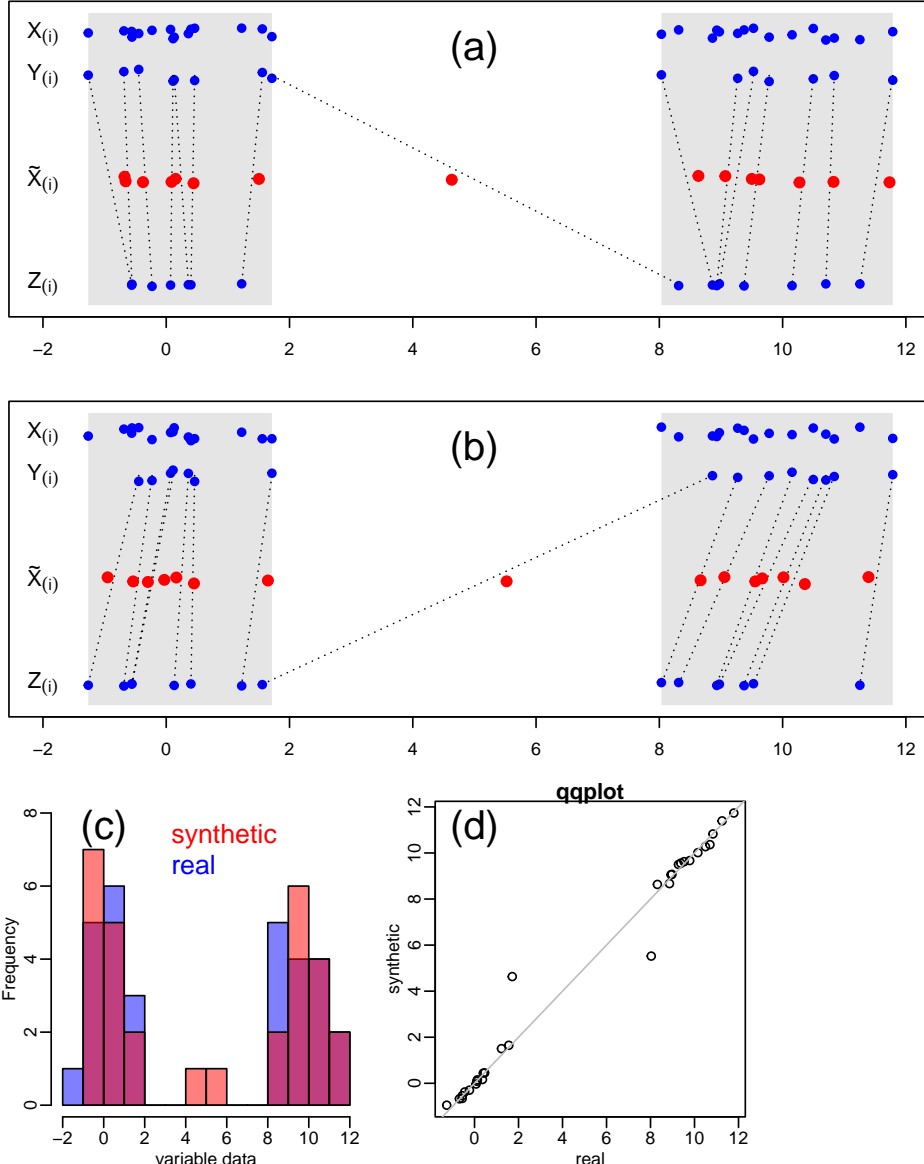

*Figure 69.* Problematic case for Algorithm 3: multimodal distributions with discontinuous support. To illustrate this problematic case, we consider a variable with a bimodal distribution with modes centered at 0 and 10 and with a discontinuous support between the two modes. The real data is shown in the $\boldsymbol{X}_{(i)} = (X_{(1)}, X_{(2)}, \ldots, X_{(30)})$ rows in panels a and b, and the blue histogram in panel c. (Note that there are no real data values (blue dots) between 2 and 8.) In this example, $n = 30$, with 15 values in the mode around 0 and 15 values in the mode round 10. Panels a and b show the synthetic data (red points) generated, respectively, by two calls to Algorithm 3. $\boldsymbol{Y}_{(i)} = (Y_{(1)}, Y_{(2)}, \ldots, Y_{(15)})$ and $\boldsymbol{Z}_{(i)} = (Z_{(1)}, Z_{(2)}, \ldots, Z_{(15)})$ represent the sorted subsamples of size $m = 15$ (randomly drawn from $\boldsymbol{X}$). The dotted lines connect the elements of same order in $\boldsymbol{Y}_{(i)}$ and $\boldsymbol{Z}_{(i)}$ (i.e., $Y_{(1)}$ is connected to $Z_{(1)}$, $Y_{(2)}$ to $Z_{(2)}$, and so on). Algorithm 3 may generate synthetic values outside the training data range (grey bands) whenever the number of points in $\boldsymbol{Y}_{(i)}$ and $\boldsymbol{Z}_{(i)}$ within a given mode are different. For instance, in panel a, the mode around 0 contain 8 points $(Y_{(1)}, Y_{(2)}, \ldots, Y_{(8)})$ belonging to $\boldsymbol{Y}_{(i)}$, and 7 points $(Z_{(1)}, Z_{(2)}, \ldots, Z_{(7)})$ belonging to $\boldsymbol{Z}_{(i)}$, whereas the mode around 10 contain 7 points $(Y_{(9)}, Y_{(10)}, \ldots, Y_{(15)})$ in $\boldsymbol{Y}_{(i)}$ and 8 points $(Z_{(8)}, Z_{(9)}, \ldots, Z_{(15)})$ in $\boldsymbol{Z}_{(i)}$. As a consequence, we have that $Y_{(8)}$ belongs to the 0 mode while $Z_{(8)}$ belongs to the 10 mode, and the algorithm ends generating a synthetic value $\tilde{X}_{(8)} = 4.63$ outside the range of $\boldsymbol{X}$. A similar issue happens in panel b because mode 0 contains 7 points in $\boldsymbol{Y}_{(i)}$ and 8 points in $\boldsymbol{Z}_{(i)}$, and mode 10 contains 8 points in $\boldsymbol{Y}_{(i)}$ and 7 points in $\boldsymbol{Z}_{(i)}$. This issue can, nonetheless, be easily resolved by generating additional synthetic samples and discarding samples outside the range of the training data.

