# OpenReview forum: "TabSDS: a Lightweight, Fully Non-Parametric, and Model Free Approach for Generating Synthetic Tabular Data"
_ICML.cc/2025/Conference — ICML 2025 poster_

### Official Review · Reviewer_r8L7 · 2025-03-08

**Overall Recommendation:** 4

**Summary:**

This paper introduces a model-free approach for synthetic data generation based on direct data perturbation. The method operates by sequentially shuffling feature values while conditioning on binned representations of other features. The authors compare their approach to generative models, evaluating its effectiveness in generating synthetic data.

**Claims And Evidence:**

>[...] it extends SJPPDS from a data perturbation approach (where the data is shuffled but no new values are generated) into a fully synthetic data method.

While the authors introduce an initial step aimed at producing new values for numerical features that did not exists in the training set, in my view, the proposed method should still be classified as a data perturbation approach as it is not unlike adding noise to an existing dataset.

The authors compare their method with generative models which are forced to compress a dataset into a set of model parameters which are then used to generate new data. To me, this seems like a fundamentally different approach.

> In addition to being easier to tune, this allows for a very precise control of the trade-off between data utility/fidelity and data privacy

[**Resolved Post-rebuttal**] While this may be true, I find the experimental setup fails to properly analyze this trade-off when comparing to the other methods (see **Methods And Evaluation Criteria**).

> [...] show that it consistently shows very competitive performance against the alternative approaches (including TabDDPM - the current state-of-the-art for data quality)

The authors miss more recent works that have claimed to outperform TabDDPM.
A non-exaustive list would be:

[1] Alexia Jolicoeur-Martineau et al. Generating and Imputing Tabular Data via Diffusion and Flow-based Gradient-Boosted Trees. AISTATS 2024

[2] Hengrui Zhang et al. Mixed-Type Tabular Data Synthesis with Score-based Diffusion in Latent Space. ICLR 2024

[3] Juntong Shi et al. TabDiff: a Mixed-type Diffusion Model for Tabular Data Generation. ICLR 2025

While I do not believe that comparison with these methods is necessary since TabDDPM's performance is, in my experience, comparable, this claim should probably be adjusted.

[**Post-rebuttal note:**] I considered this a minor issue and the authors added more relevant baselines like PATE-GAN and SMOTE

**Essential References Not Discussed:**

Not that I am aware.

**Experimental Designs Or Analyses:**

The overall experimental design is sensible in my opinion.
The hyperparameters of competing methods are tuned or taken from previous works.

My main concern is with the choice of metrics and how they are presented (see **Methods And Evaluation Criteria**).

**Methods And Evaluation Criteria:**

[**Resolved Post-rebuttal**] In my opinion the selection of benchmark datasets and generative baselines is adequate. I believe, however, that the authors should also compare to other data perturbation approaches, such as approaches that add noise to the data.

The proposed method generates data by directly manipulating an existing dataset, in contrast to the baselines which are forced to distill a training set down to a more compressed representation which is then used to generate new data.
The question of privacy if therefore of central importance and I have two main concerns in this regard:

1. [**Resolved Post-rebuttal**] The choice of the privacy metric. On some datasets we see that the membership inference score is worse than a random classifier. Furthermore, when looking at how this MIA score varies with $n_c$ in Figures 3 and Figures 12-22, we see that it sometimes has little effect and often is not monotonic and varies wildly. All of these observations cast doubts on how adequately this metric can be used to assess privacy.
2. [**Resolved Post-rebuttal**] Instead of picking a single value of $n_c$, comparison should be made considering both privacy and utility at the same time. This could be accomplished, for example, by making plots similar to those in [1], which show the full trade-off curve between privacy and utility.
Alternatively, a fair comparison would control for one aspect and measure the other. This approach, however, seems less viable given that the other generative baselines don't necessarily allow for this fine control using a single hyperparameter.
As it stands, we are presented with two separate tables where it looks like TabSDS is sometimes achieving higher utility at the cost of less privacy.


[1] van Breugel et al. Membership Inference Attacks against Synthetic Data through Overfitting Detection (2023)

**Other Comments Or Suggestions:**

N/A

**Other Strengths And Weaknesses:**

[**Resolved Post-rebuttal**] The SJPPDS approach is somewhat *ad-hoc* and does not come with any formal privacy guarantees, unlike other perturbation approaches.
It is unclear to me why one would opt to use it instead of simpler methods such as adding noise to the data.
In my opinion, some comparison with this type of approach is crucial to prove the value of the proposed method.

The Appendix includes additional reproducibility details and exhaustive analysis of the results on a dataset-by-dataset basis and is one of the strong points of the paper.

**Questions For Authors:**

- [**Resolved Post-rebuttal**] How would this method compare to other perturbtation approaches based on adding noise to data?
- [**Resolved Post-rebuttal**] What do the privacy-utility trade-off curves look like for both approaches?
- [**Resolved Post-rebuttal**] Why would I prefer TabSDS over the simpler option?

[**Post-rebuttal Note:**] The authors have provided convincing evidence that their method outperforms simpler approaches like noise addition. While the results against SMOTE and particularly TabDDPM are much more mixed and it seems to me that the latter generally achieves a better trade-off than those made possible by TabSDS, I can see the value in the alternative proposed approach.
As a result, I have raised my score from a 2 to a 4.

**Relation To Broader Scientific Literature:**

This work extends previous work [1] in two directions:

1. It adds a prior step aimed at generating new values for numerical features.
2. It extends the method (SJPPDS) to work with categorical features.

While these extensions are interesting, I still believe that the question of originality arises.
The second point seems rather trivial and the impact of the first is not analyzed regarding additional privacy.
The main draw of the paper is, in my opinion, the extensive experiments comparing the method to generative approaches

[1] Elias Chaibub Neto. Statistical disclosure control for numeric microdata via sequential joint probability preserving data shuffling

**Theoretical Claims:**

I checked the proof of Theorem 1 and found no issues.

---

> ### Author Rebuttal · Authors · 2025-04-01
>
> Thanks for your very thoughtful comments/suggestions. We address below your main concerns, and we would be happy to follow up on the remaining ones (or any additional questions) during the discussion period. Please, let us know.
>
> >Authors should compare to approaches based on adding noise data … My main concern is with the choice of privacy metric and how the comparisons are presented. Comparison should be made using plots which show the full tradeoff curve between privacy and utility.
>
> We now included new comparisons with the additive noise approach using 2 new privacy metrics (and display the results using privacy/utility tradeoff curves as suggested).
>
> Following the suggestion of Reviewer Y1nw, the new adopted metrics are the sorted versions of the Standard Deviation Interval Distance (SDID) metric (which measures robustness to attribute disclosure attacks), and the sorted version of the Distance Based Record Linkage (DBRL) metric (which measures robustness to re-identification attacks). Please, see our reply to Reviewer Ai12 for further details about these metrics.
>
> We also adopted Energy Distance (ED) as a fidelity metric. (ED is closely related to MMD and can be used to measure the distance between 2 multivariate distributions.)
>
> We implemented the noise addition comparison using the sdcMicro R package. Noise addition was evaluated over a grid of 13 increasing levels of additive noise ranging from 1% to 50% of the standard deviation of each variable.
>
> Figures 1-10 in the linked pdf report the results: https://drive.google.com/file/d/1PYa_pgyQqqWwPI0Dq0ichscMmYftoyfK/view?usp=sharing
>
> In each figure, panel a show the ED vs DBRL tradeoff curves for noise addition (blue) and TabSDS (black). The red dot represents the selected value of $n_c$ based on the DCR criterium, while the purple symbols represent the other baseline generators. To better compare the generators, panel b shows the same results without displaying the noise addition curve. Panels c and d show analogous results in terms of the ED vs SDID tradeoff. To further illustrate the influence of the amount of noise and of $n_c$ on these privacy and fidelity metrics, panels e-i show the metric values against the perturbation parameter values.
>
> Inspection of the figures shows that:
>
> 1. TabSDS usually achieves a better ED vs DBRL tradeoff than the additive noise approach (note how the black curves are usually closer to the bottom left corner than the blue curves in panel a).
>
> 2. TabSDS tends to achieve better fidelity (lower ED) than the other approaches (the red dot tends to be closer to 0).
>
> 3. In terms of re-identification risk (panel b), all methods generated low risks (below 2.5% across all datasets) and none of the methods stands out as systematically better or worse than the other methods.
>
> 4. In terms of attribute disclosure (panel d), again, the risks tended to be low for all methods.
>
> >While these extensions are interesting, I still believe that the question of originality arises. The second point [extension to categorical variables] seems rather trivial and the impact of the first [generating new values for numerical features] is not analyzed regarding additional privacy.
>
> Regarding the extension to categorical variables, while our approach is simple, alternative approaches based on the computation of ranks on categorical variables are considerably more involved and difficult to implement in practice. For instance, [1] proposes the use of an ontology-based semantic distance.
>
> Regarding the evaluation of additional privacy protection achieved by new values, in our new evaluations we also compare TabSDS against a simplified version of the algorithm (denoted TabSJPPDS), which discards the new value generation step. The results are presented in panels e, f, and g of Figures 1-10 on the pdf linked above. Overall, generating new values improves privacy protection, as shown by the lower disclosure risks of TabSDS (black) relative to TabSJPPDS (brown) in panels e and f. As expected, generating new values also lead to a decrease in fidelity, as illustrated by the higher ED values of TabSDS relative to TabSJPPDS (panel g).
>
> Finally, note that these extensions have important practical impacts. First, by handling categorical variables TabSDS can be much more widely applied than SJPPDS (since tabular datasets often contain mixed data). Second, even for exclusively numeric datasets, in addition to the better re-identification and attribute disclosure protection, TabSDS also offers better protection against a trivial form of membership inference attack for which SJPPDS is helpless. (Namely, for datasets containing unusual values, an attacker might immediately infer membership by recognizing the presence of unique values of a given record in the perturbed dataset.)
>
> [1] Domingo-Ferrer et al (2013). Information Sciences 242:35-48.
>
> Thanks again for your review, and let us know if you have any additional questions.

---

> > ### Comment · Reviewer_r8L7 · 2025-04-03
> >
> > Thank you for the detailed response. I find the additional results encouraging but I have a couple follow-up comments/questions:
> > - In the provided figures, the range of values used for the added noise make it so that one can't really properly see the remaining approaches. There is a zoomed in version of the pictures but which doesn't really show the trade-off curve for the proposed method. Could you zoom in to the area of interest and still show the curve for TabSDS?
> > - What was the reasoning behind the change to ED for the fidelity metric? I thought that the previous setup with the AUC of a classifier was easy to understand and, as far as I am aware, no reviewers raised any issues with it. In contrast, ED seems to assume the existence of a meaningful metric on the input space which I find questionable for tabular data. How did you choose this metric? How are categorical variables handled? And how are variables scaled so that the the distance between points is a sensible metric given the different semantic meanings of each variable.

---

> > > ### Author Response · Authors · 2025-04-07
> > >
> > > Sorry for the late reply. (We ended up implementing the SMOTE baseline, as suggested by Reviewer 9VCx, and wanted to include it in these comparisons.)
> > >
> > > The reviewer raises good reservations wrt the use of ED as a fidelity metric. We regenerated the tradeoff plots replacing ED by the detection test AUC for discriminating synthetic and real data. (We adopted ED because it has an efficient R implementation and is analogous to the Wasserstein distance, the fidelity metric in the tradeoff plot of van Breugel et al 2023, pointed by the reviewer.)
> > >
> > > As for the privacy metrics, we now include additional comparisons against DCR (in addition to the DBRL and SDID metrics).
> > >
> > > Figures 1 and 2 of the linked PDF https://drive.google.com/file/d/1MGP02tn1rmiZansoQXP27EbXhwrtnC-_/view?usp=sharing show tradeoff plots for all datasets (now including the tradeoff curve for the TabSDS method). In addition to the original baselines, they now also include comparisons vs SMOTE (based on 5 and 20 nearest neighbors), and vs ADSGAN and PATEGAN (as requested by Reviewer Ai12). The results for these DP-based models were, nonetheless, based on default hyperparam. choices and should be taken with a grain of salt.
> > >
> > > The left panels of Figs 1 and 2 show DCR vs AUC plots. The red line represents the DCR score comparing the training and test sets and provides an estimate of the DCR value we would expect to see for an ideal generator able to draw i.i.d. data from the same distribution as the training data (as described in Section G5).  The red dot represents the selected value of $n_c$ based on the DCR criterium (i.e., the DCR value closest to the test set DCR). The middle and right panels show the tradeoff plots comparing DBRL vs AUC and SDID vs AUC, respectively. (Note the DBRL and SDID values differ slightly from the values reported previously due to a small bug in our code, which is now fixed.)
> > >
> > > Overall, DDPM, TabSDS, and SMOTE tended to outperform the other methods in terms of fidelity. These 3 methods tended to show somewhat balanced performances with none of the methods consistently outperforming the others. However, in terms of privacy, SMOTE tended to be considerably worse than DDPM and TabSDS with respect to DCR. It also tended to be worse than DDPM (and of TabSDS to a lesser extent) in terms of DBRL and SDID.
> > >
> > > ADSGAN, PATEGAN, TVAE, CTGAN, and BayesNet tended to trade high data privacy by low data fidelity. In all datasets, these methods showed AUCs close to 1, low DBRL and SDID, and high DCR. (These high DCR values are likely a consequence that these models fail to approximate well the distribution of the training data.) ARF tended to do slightly better than these models in a few datasets.
> > >
> > > Figs 2 to 12, report additional comparisons for each of the datasets. Panels a, b, and c present the tradeoff curves comparing TabSDS (black) vs additive noise (blue). The additive noise approach showed high AUC values across all noise levels across most datasets and is not competitive against TabSDS. Panels g, h, and i compare TabSDS against TabSJPPDS, and illustrate the additional privacy protection achieved by generating new values. (Note the higher DCR and lower SDID scores. The differences were less clear cut for DBRL). Panels j, k, and l illustrate that higher noise levels lead to increased data privacy.
> > >
> > > The results described above (and in the paper) were based on DCR values computed on the original data scales. Categorical variables were one-hot-encoded prior to DCR computation. (Note this should not cause issues given that the datasets contained only a few binary categorical variables. Datasets containing only or mostly categorical variables such as Mushroom and Adult datasets were not evaluated given that noise addition can only be applied to numeric data.)
> > >
> > > As pointed by the reviewer, one potential caveat when variables have different scales is that distance-based metrics (such as DCR) might be dominated by the wider range variables. To evaluate this potential issue, we also performed comparisons based on DCR values computed on scaled data. (Reported here: https://drive.google.com/file/d/1Kw1x3pDmiLT-ApTFR8vB5fNxjKAON2B5/view?usp=sharing) These results show the same qualitative conclusions as before. The main quantitative difference was that the selected $n_c$ values tended to be lower, leading to a decrease in fidelity and increase in privacy of TabSDS relative to the previous results.
> > >
> > > Finally, the linked PDF shows the qualitative comparisons of TabSDS against the newly included baselines: https://drive.google.com/file/d/1r1k8prAyA-8AUXJYysPiXeQtk5-IlqbK/view?usp=sharing As before, TabSDS tended to generate more realistically looking marginal distributions than the other methods (Figs 2 to 11). SMOTE, however, tended to recapitulate better the correlations from the real data (Fig 1). The DP based methods generated considerably lower quality data.
> > >
> > > Please, let us know if you have any other questions, and sorry again for the late reply.

---

### Official Review · Reviewer_Y1nw · 2025-03-13

**Overall Recommendation:** 3

**Summary:**

The paper proposes a novel method for synthetic data generation. This new method is based on two basic actions: generating new values for each of the features by sampling through from its "interpolated marginal", then shuffling the original data following the algorithm SJPPD and then matching the ranking of the shuffled data with the synthetic ones.

**Claims And Evidence:**

One of the worries I have with this paper is what happens when the number of features increases. As the number of features increases, we keep shuffling. Wouldn't this make the distribution of the features become more and more similar to the distribution obtained with random shuffling?

In your experimental analysis the authors tested the method with datasets with up to 27 features. What happens if we take datasets with 100 or more features?

Finally, it would be interesting to have a discussion/analysis of the relationship between $n_c$ and the privacy metrics. If $n_c$ grows, can we still retain good privacy values?

**Essential References Not Discussed:**

All the essential references are discussed.

**Experimental Designs Or Analyses:**

Given that the model is not always the best performing model it would be useful to perform a Friedman statistical test (once you have included also Great and WGAN) together with a Nemenyi test to check the statistical significance of your results. For more info see (Statistical Comparisons of Classifiers over Multiple Data Sets, Demsar, 2006).

**Methods And Evaluation Criteria:**

In the methods used for comparison some key methods are missing. In particular: WGAN [Arjovsky et al., 2017] and GReaT [Borisov et al., 2023].

In the metrics used for privacy, the authors used assessed the robustness of the generated data wrt membership attacks.
However, it would be interesting to also evaluate its robustness wrt attribute disclosure attacks, (i.e., try to gain access to sensitive attributes of an entity from the real dataset) and re-identification attacks (i.e., attacks try to map a synthetic data point back to the original dataset)

**Other Comments Or Suggestions:**

The paper contains a lot of algorithms, which often are not even needed and risk to simply confuse the reader. For example, algorithm 8 is simply the application to every column of the matrix $\mathbf{X}$ of the algorithm InterpolatedOrderStatsSampling.

Figures are also not aligned well (see, e.g., Figure 5c and 5d).

Finally the na,e of the other methods (e.g., TVAE) should be capitalized following the capilazion given by the authors.

**Other Strengths And Weaknesses:**

No other strengths and weaknesses need to be discussed.

**Questions For Authors:**

Please see the boxes above.

**Relation To Broader Scientific Literature:**

The authors have covered good part of the literature

**Theoretical Claims:**

See claims and evidence box

---

> ### Author Rebuttal · Authors · 2025-04-01
>
> Thanks for your comments and thoughtful suggestions. While we could only address your main concerns here, we would be happy to follow up on the remaining ones (or any additional questions) during the discussion period. Please, let us know.
>
> >One of the worries I have is what happens when the number of features increases. Wouldn't the distribution become more and more similar to the distribution obtained with random shuffling?
>
> No. The sequential joint probability preserving data shuffling approach employed to shuffle the data is designed to preserve the association structure of the data columns, irrespective of the number of columns.
>
> To illustrate this point, we:
>
> 1. Simulated a dataset $X$ from a multivariate normal distribution of dimension 200 (with a highly structured correlations).  In our experiment, this simulated data plays the role of the “real” data.
>
> 2. Applied TabSDS to subsets of $X$ with increasing numbers of features.
>
> 3. Compared how synthetic datasets generated by TabSDS recapitulated the real data's correlation structure as the number of features increased.
>
> Explicitly, in our experiment, we considered 10 subsets of $X$, composed, respectively by the first 20, 40, …, 180, and 200 features of $X$.
>
> For each of the 10 subsets, we generate synthetic data (using the same $n_c$ value) and computed the correlation matrices of the real and the synthetic data subsets. To measure how well the synthetic data recapitulated the correlations of the real data we computed the L2 distance (L2d) between the correlation matrices (define as average($(r_j – s_j)^2$), where $r_j$ and $s_j$ represent entries of the real and synthetic data cor. matrices and the average is taken over the upper (or lower) diagonal entries of the matrices).
>
> The above procedure was repeated using $n_c$ values set to 5, 10, 15, 20, 25, and 30.
>
> The results are reported in Figures 1 to 5 on the linked pdf file: https://drive.google.com/file/d/1dB9Vsb0IucFZBCft_k13Ljb82FXr0zE1/view?usp=sharing
>
> Figure 1a reports the values of the L2d for each of the 10 subsets. (Each boxplot reports the L2d values across the 6 distinct $n_c$ values.) Figure 1b shows the L2d values across the 10 subsets for each of the $n_c$ values. As expected, larger $n_c$ values lead to higher fidelity synthetic data and lower L2d values. This figure shows that the ability to recover the correlation structure of the real data remains largely constant as we increase the number of features from 20 to 200. (We would expect to see an upward trend if the method’s performance worsened with larger number of features.)
>
> Finally, Figures 2 to 5 compare the correlation matrices of the real and synthetic data for subsets of 20, 40, 100, and 200 features (based on $n_c = 30$). They illustrate how TabSDS is able to recover very well the correlation structure of the real data across all these subsets.
>
> >In the comparison some key methods are missing. In particular: WGAN and GReaT
>
> We restricted our comparisons to generators implemented in Synthcity. While GReaT is available, preliminary checks showed it was computationally unfeasible to include it in our comparisons (we run our experiments in CPUs). (Also, recent work by Zhang et al. (2024) [1] has shown that GReaT tends to be outperformed by TabDDPM - which is included in our comparisons.) WGAN is not available in Synthcity.
>
> [1] Zhang et al (2024) Mixed-type tabular data synthesis with score-based diffusion in latent space. ICLR 2024.
>
> >the authors assessed robustness of the generated data wrt membership attacks ... it would be interesting to also evaluate robustness wrt attribute disclosure attacks and re-identification attacks
>
> This is an good suggestion. We have now included new evaluations based on the sorted versions of the Standard Deviation Interval Distance (SDID) and Distance Based Record Linkage (DBRL) metrics (which measure robustness to attribute disclosure and re-identification attacks, respectively). Please, see our reply to Reviewer Ai12 for further details about these metrics.
>
> As suggested by Reviewer r8L7, we now report these new results using tradeoff curves comparing these privacy metrics against the Energy Distance (ED) fidelity metric. Results are presented in the linked pdf file: https://drive.google.com/file/d/1PYa_pgyQqqWwPI0Dq0ichscMmYftoyfK/view?usp=sharing Please, see our reply to Reviewer r8L7 for the interpretation of these results.
>
> >it would be interesting to have a discussion/analysis of the relationship between n_c and the privacy metrics. If n_c grows, can we still retain good privacy values?
>
> As $n_c$ grows, TabSDS generates less private data. This point is discussed in lines 376 to 380 of the manuscript. Also, Figure 4 (main text) and Figures 11d - 22d (Appendix) illustrate this by showing decreasing DCR privacy metric values as $n_c$ increases. Similarly, panels e and f of Figures 1-10 (see the linked pdf file) show rising disclosure risks in sorted DBRL and SDID metrics as $n_c$ grows.

---

### Official Review · Reviewer_Ai12 · 2025-03-13

**Overall Recommendation:** 4

**Summary:**

The paper introduces TabSDS, a non-parametric and model-free method for generating synthetic tabular data. Unlike deep generative models (DGMs), which are computationally expensive and require extensive hyperparameter tuning, TabSDS leverages rank-based transformations and data shuffling to approximate the joint probability distribution of real data. The method extends sequential joint probability preserving data shuffling (SJPPDS) by incorporating categorical features and generating entirely new data points rather than merely shuffling existing data.

## update after rebuttal

I thank the authors for their rebuttal and remain positive about the work.

**Claims And Evidence:**

Claim 1: TabSDS offers competitive fidelity and utility compared to state-of-the-art models. -- Supported by benchmarking against TabDDPM, ARF, TVAE, CTGAN, and Bayesian networks using the Synthcity library. Results demonstrate strong performance in fidelity and downstream ML utility.

Claim 2: TabSDS is significantly faster than both deep generative models and adversarial random forests (ARF). -- Experimental results show orders-of-magnitude improvements in runtime across multiple datasets.

Claim 3: TabSDS improves privacy compared to deep generative models. -- Measured via membership inference attack (MIA) success rates, showing reduced susceptibility compared to deep models and ARF.

**Essential References Not Discussed:**

The paper could benefit from a comparison with differential privacy-based synthetic data generation methods.

**Experimental Designs Or Analyses:**

Yes. The evaluation looks sound to me (cf. metric and evaluation criteria section). Experiments were repeated 10 times to account for random factors.

**Methods And Evaluation Criteria:**

The evaluation settings and criteria are appropriate to support the claims.

The evaluation assess fidelity (ROC AUC on synthetic vs. real detection), utility (ROC AUC on downstream ML tasks trained with synthetic data) and privacy (Membership inference attack (MIA) success rate). These characteristics are suitable and the metrics used appropriate.

The comparisons baselines are TabDDPM, ARF, TVAE, CTGAN, and Bayesian networks, which is a good representative set of the main synthetic data generation methods for tabular data.

Evaluation includes 12 datasets containing mixed numerical and categorical data. These datasets are often used in the literature.

**Other Comments Or Suggestions:**

None

**Other Strengths And Weaknesses:**

Efficiency: Faster than generative models, with minimal parameter tuning required.

Flexibility: Handles both categorical and numerical data, unlike SJPPDS.

Privacy-Utility Tradeoff: Provides explicit control over fidelity and privacy through the nc parameter.

**Questions For Authors:**

How does TabSDS compare to differentially private synthetic data methods?
How does it handle imbalanced categorical variables?

**Relation To Broader Scientific Literature:**

The key contribution is to propose a new tabular data generation method that is more efficient than state of the art approaches (like deep generative models) while remaining of quality in terms of fidelity, utility and privacy. Tabular data generation remains an unsolved problem (or at least, existing solutions are not entirely satisfactory) so I believe there is value in pursuing research in this area. The proposed approach rely on SJPPDS, which is novel compared to the usual practice of the literature.

**Theoretical Claims:**

I checked the proof of Theorem 4.1 (Rank-matching step: Maintains the joint probability distribution of the original data). They look senseful to me but I cannot guarantee it.

---

> ### Author Rebuttal · Authors · 2025-04-01
>
> Thanks for your thoughtful comments and questions.
>
> >How does TabSDS compare to differentially private synthetic data methods?
>
> We now included comparisons with two additional DP based methods available in Synthcity: ADSGAN [1] and PATEGAN [2]. (We also evaluated the DPGAN [3] but didn’t include it in the comparisons because it was considerably slower and tended to generate considerably lower fidelity data.)
>
> Due to time constraints, we were unable to optimize the hyperparameters of these two DP-based generators, and the results presented here are based on the default hyperparameter values in Synthcity. (Hence, they should be taken with a grain of salt and are aimed for the appendix.)
>
> Following a suggestion by Reviewer Y1nw we now evaluate the methods wrt 2 additional privacy metrics, the sorted version of Distance Based Record Linkage (DBRL) metric, which measures robustness to re-identification attacks; and the sorted version of the Standard Deviation Interval Distance (SDID) metric, which measures robustness to attribute disclosure attacks. The standard versions of the DBRL [4] and SDID [5] metrics are traditionally used in the Statistical Disclosure Control field to evaluate perturbation methods.  Application to synthetic data requires a prior sorting step as described in [6]. (The basic idea is to sort the rows of both the original and synthetic datasets according to the values of a given column of the data prior to the computation of the DBRL and SDID metrics.)
>
> After the sorting step, the DBRL metric is implemented by computing the Euclidean distances between each record in the synthetic dataset against all records in the real dataset. A synthetic record is considered “linked” when the nearest record in the real data turns out to be the corresponding real record. The metric is defined as the proportion of synthetic records linked to real records. After the sorting step, the standard SDID metric corresponds to the proportion of real records inside a standard deviation interval whose center is the corresponding synthetic record.
>
> Following the suggestion of Reviewer r8L7, to better facilitate the visualization of the tradeoffs between privacy and fidelity we now report plots comparing these two new privacy metrics against the Energy Distance (ED) fidelity metric. (ED is closely related to maximum mean discrepancy and can be used to measure the distance between 2 multivariate distributions.)
>
> Figures 1 to 4 in the linked pdf report these comparisons: https://drive.google.com/file/d/1M93WfhXtWQVRLXsvsgoac0S1O_kAqjt_/view?usp=sharing Figure 1 shows scatterplots of ED vs DBRL for the original methods and ADSGAN. Overall, TabSDS tended to generate higher fidelity data than the other methods (the red dot tends to be closer to 0), while ADSGAN tends to generate lower fidelity data (the inverted purple triangle tends to be farther from 0 than most methods on most of the datasets). In terms of re-identification risk, all methods generated low risks (below 2.5% across all datasets) and none of the methods stood out as systematically better or worse than the other methods. Not even ADSGAN tended to do systematically better on this metric. This might, however, be due to the fact that the default hyperparameter value in Synthcity is set for moderate privacy protection (or due to suboptimal model training).
>
> Figure 2 shows analogous results for the ED vs SDID comparison. Again, the SDID risk tended to be low for all methods (and was 0 for ADSGAN across all datasets). Figures 3 and 4 add the comparisons against PATEGAN (open purple diamond). Overall, PATEGAN tended to achieve considerably lower data fidelity than ADSGAN (note the extended x-axis).
>
> [1] Yoon et al (2019) Anonymization through data synthesis using generative adversarial networks (ADS-GAN): a harmonizing advancement for AI in medicine.
>
> [2] Jordon et al (2019) PATE-GAN: generating synthetic data with differential privacy guarantees.
>
> [3] Xie et al (2018) Differentially private generative adversarial network.
>
> [4] Domingo-Ferrer and Torra (2001) A quantitative comparison of disclosure control methods for microdata.
>
> [5] Mateo-Sanz et al (2004) Outlier protection in continuous microdata masking.
>
> [6] Chaibub Neto (2024) Statistical disclosure control for numeric microdata via sequential joint probability preserving data shuffling.
>
> >How does TabSDS handle imbalanced categorical variables?
>
> When categorical variables are imbalanced, synthetic data generators can struggle to preserve the frequencies of rare categories observed in the real data.
>
> This is not an issue for TabSDS which, by construction, preserves the marginal distribution of the categorical variables (since the data from each categorical variable is simply shuffled around, so that the marginal frequencies of the categories are preserved).
>
> Thanks again, and please let us know if you have any further questions.

---

### Official Review · Reviewer_9VCx · 2025-03-14

**Overall Recommendation:** 2

**Summary:**

The paper introduces TabSDS, a non-parametric, lightweight framework for generating synthetic tabular data. The approach is based on the Sequential Joint Probability Preserving Data Shuffling (SJPPDS) algorithm, a perturbation method that relies on restricted feature permutations to preserve the joint distribution of the data. Specifically, given T columns or features, the algorithm performs multiple restricted permutations of T-1 columns with respect to the remaining column. Each column's data is shuffled, ensuring that the marginal distributions are preserved.

The authors extend this approach by generating new data for each feature, effectively providing synthetic data while enhancing privacy. This generation process is carried out using a new algorithm called IOSSampler, which samples new data uniformly from the values of ordered sub-samples of the input feature.

The proposed method is significantly faster than existing deep learning and machine learning approaches, while it trades off privacy for better realism.

## update after rebuttal
The authors addressed some of my concerns, but I believe the generation of new data --- in practical settings --- could be improved. I keep the original score.

**Claims And Evidence:**

Yes, claims are supported by theorems/proofs and experiments.

**Essential References Not Discussed:**

None

**Experimental Designs Or Analyses:**

The datasets are not explicitly linked, and in Table 4 the number of features (num and cat) appears to differ from the cited paper where these datasets were originally used, as well as from other related studies (TabDDPM). To improve reproducibility, I suggest adding the specific URL or reference for each dataset, at least in the appendix.

Additionally, the authors mention that all models were tested on CPU instance (with 8 cores). Given the computational demands of deep learning models like TabDDPM, I would like to ask about the number of training epochs used. If the models were not trained sufficiently, it could explain potential performance limitations.

**Methods And Evaluation Criteria:**

The proposed method and evaluation is reasonable. The selected benchmarks could be extended, including for example a shallow interpolation-based methods, e.g., SMOTE (Chawla et al., 2002).

**Other Comments Or Suggestions:**

- Acronyms in Section 2 maybe should be capitalized, e.g., *DDPM* instead of *ddpm*.

- The datasets used in the paper (table 4) seem to have a different number of columns compared to the original paper and TabDDPM.  For exampel, the columns *#num* and *#cat* list feature for numerical and categorical. But House_16H appears to have 17 features instead of the 16 features in the cited Hansen et al. (2023).

- I suggest that to improve reproducibility, consider providing a URL where the exact datasets can be downloaded. And also add details on training epochs for benchmarks

- In large datasets with many columns and values, category granularity can be an issue—some categories may be too broad or too fine-grained, leading to sparsity or excessive heterogeneity. How to handle that and select the correct hyper-parameters?

- Theorem 4.1 is cited ad Theorem 1.

**Other Strengths And Weaknesses:**

Strengths:
- The paper addresses an important topic, proposing a new non-parametric lightweight method .
- The method has very good performance in terms of utility and realism.
- The method could be a good alternative solution to more expensive and classic DL/ ML approaches.

Weaknesses:
- The novelty of the proposed approach compared to existing work should be better explained. In particular, compared to Chaibub Neto (2024) and Domingo-Ferrer et al. (2025), the main contribution appears to be a lighter approach to fit marginal distributions and generating synthetic samples.
- I have some concerns about the evalution. The method has very good performance in utility and realism, but privacy may be a problem. Also the real and syn marginal distributions seem really close. Some benchmarks are missing (e.g., SMOTE, CTABGAN+), while some details are missing.

See other comments and questions.

**Questions For Authors:**

Q1: In the examples of Table 2 and Table 3, does the method requires for the same category to have different ranking? Is possible to ahve the same ranking for the category A, and what's the impact of doing that? Different ranks, could "break" the join distributions in the synthetic data?

Q2: In algorithm 3: while the distribution is expected to converge as *m* → ∞, in practice, will the input and real distributions actually match for like bimodal distributions? How to choose the correct value for m? For example a bimodal distribution with values clustering around 0 and 10? When we sample: uniform(0, 10) we can get values not in the real/training set?

Q3: The authors mention that all models were tested on a CPU instance (with 8 cores). Given the computational demands of deep learning models like TabDDPM, I would like to ask about the number of training epochs used. If the models were not trained sufficiently, it could explain potential performance limitations. How they handle this possible limitation?

**Relation To Broader Scientific Literature:**

The paper addresses an important topic: synthetic tabular data generation. While it differs from existing deep learning and machine learning models, it builds and extends an interesting line of research, particularly relevant are the work of Chaibub Neto, 2024 (SJPPDS), and the work of Domingo-Ferrer et al. (2025).

**Theoretical Claims:**

Theorem 4.1 seems correct to me, but I didn't check the formal proof.

---

> ### Author Rebuttal · Authors · 2025-04-01
>
> Thanks for the very thoughtful comments and questions.
>
> >Q1. In Tables 2 and 3, does the method requires for the same category to have different ranking?
>
> Yes, it requires different ranks for each category. The problem is that if you assign the same rank value for each category the restricted permutation generates an identical copy of the real data (except for the order of the rows). The linked table https://drive.google.com/file/d/12OXRKwkT06lokekRKe42VOjANX0RJytV/view?usp=sharing shows an illustrative toy example, where the original categorical data is recoded as $X_1=[C \equiv 1, D \equiv 2]$ and $X_2=[A \equiv 1, B \equiv 2]$, and perm. 1 to 4 represent restricted permutations of $X_1$ relative to $X_2$ (where we shuffle the values of $X_1$ within each level of $X_2$). Clearly, this approach cannot be used for datasets containing only categorical variables. Having different ranks for the same category is necessary for generating distinct datasets, while still preserving the joint distribution to a good extent. (Also, it helps unify the treatment of categorical and numeric variables in mixed datasets.)
>
> >Q2. In algorithm 3 will the input and real distributions match for bimodal distributions? How to choose the value for m?
>
> Yes, the synthetic and real distributions will match if the bimodal distribution has continuous support (e.g., bottom left panel of Fig. 29). However, for a discontinuous bimodal distribution—where values cluster near 0 and 10 with no training examples in between—the method may generate a small fraction of intermediate values outside the training range. Thanks for noting this edge case; we will mention it in the Limitations section and indicate that some post processing might be necessary in this case.
>
> For the choice of m, note that m = floor(n * p), where n is the number of samples and 1/n <= p <= 0.5.  Hence, m can vary from 1 to floor(n/2). As described in page 6 (lines 292 to 300 in the 2nd column) we found that lower values of p can lead to considerable decreases in data fidelity without comparable increases in privacy. Hence, in practice, we recommend taking m = floor(n/2).
>
> >Q3. models were tested on a CPU instance. Given the computational demands of deep models like TabDDPM, how many training epochs were used?
>
> The number of training epochs was one of the parameters evaluated during hyperparameter optimization (or taken from other works) using the hyperpar. search spaces defined in Synthcity. We will add tables of hyperpar. values to the final version of the paper. For TabDDPM, the number of epochs ranged from 1051 to 8300, and the experiments run for long times on the CPU instance.
>
> >benchmarks could include SMOTE and CTABGAN+
>
> We restricted our comparisons to models available in Synthcity. Since Kotelnikov et al (2023) performed extensive comparisons of these weaker baselines against TabDDPM, and we performed comparisons against TabDDPM, we feel these additional comparisons might not be essential. But, please, let us know if you disagree (as we might be able to implement SMOTE and report results over the discussion period).
>
> >In Table 4 the # of features appears to differ from the cited paper
>
> Table 4 reports # of variables (i.e., # of features plus the target) while the Table 4 in Hansen et al 2023 reports only # of features. (Also, for the California housing we used the sklearn data rather than Hansen’s.) We will clarify these points and add specific URL/references for each dataset in the appendix.
>
> >novelty of the proposed approach should be better explained
>
> Compared to Chaibub Neto (2024), we extend SJPPDS to categorical data while also generating synthetic marginal distributions. This improves practicality: first, handling categorical variables makes our approach more widely applicable (as tabular data is often mixed). Second, even for purely numeric data, our method offers better protection against trivial membership inference attacks (which SJPPDS cannot prevent) when datasets contain unusual attribute values which can be easily spotted by an attacker.
>
> Our approach differs from Domingo-Ferrer et al (2025) in 3 key ways. First, we generate marginals via order statistics interpolation, whereas they require the user to choose predefined parametric distributions (e.g., Gaussian, Gamma). Second, we extend SJPPDS to categorical variables, while they employ a more complex ontology-based semantic ranking [1] (what is also more involved in practice). Third, we use SJPPDS for rank data shuffling, whereas they rely on different algorithms.
>
> [1] Domingo-Ferrer et al (2013). Information Sciences 242:35-48.
>
> >In large datasets, category granularity can be an issue. How to select the correct hyper-parameters?
>
> For deep learning and ML baselines, we optimized hyperparameters using Optuna within Synthcity's defined search spaces or adopted values from Hansen et al (2023), which were also optimized with Optuna in Synthcity.
>
> Please, let us know if you have any further questions.

---

### Decision · Program_Chairs · 2025-05-01

**Decision:**

Accept (poster)

**Comment:**

This paper performs synthetic data generation by sequentially shuffling features, not training a generative model. The authors provide a proof on how the joint probability distribution is captured by this process, and compare their approach to generative models in terms of fidelity and privacy.

Reviewers noted interest in the non-parametric approach, but asked for more complete benchmarking against prior work with expanded metrics that focus on privacy. These aspects were provided in the rebuttals. A majority of deficiencies raised by reviewers were addressed satisfactorily, and reviewers found the theoretical aspects sound. The addition of existing non-parametric baselines like SMOTE helps to ground the work. Some remaining weaknesses are that the main baseline method TabDDPM is likely out of date and no longer SOTA, as was originally claimed in the abstract and in the paper. I concur with Reviewer r8L7 that these claims should be toned down, and that work in generative modelling since the introduction of TabDDPM should be cited and discussed, and baselined against if possible. Given that the proposed method is non-parametric, notably different from deep generative models like TabDDPM, it may serve a very different role. Hence, it does not necessarily need to be SOTA compared to DGMs, and I can overlook this weakness to recommend acceptance.